# HIMMELI v1.0: HelsinkI Model of MEthane buiLd-up and emIssion for peatlands

Maarit Raivonen[1], Sampo Smolander[1,2], Leif Backman[3], Jouni Susiluoto[3,4], Tuula Aalto[3], Tiina Markkanen[3], Jarmo Mäkelä[3], Janne Rinne[5], Olli Peltola[1], Mika Aurela[3], Annalea Lohila[3], Marin Tomasic[1], Xuefei Li[1], Tuula Larmola[6], Sari Juutinen[7], Eeva-Stiina Tuittila[8], Martin Heimann[1,9], Sanna Sevanto[10], Thomas Kleinen[11], Victor Brovkin[11], Timo Vesala[1,12]

[1]Division of Atmospheric Sciences, Department of Physics, University of Helsinki, P.O.Box 68, 00014 Helsinki, Finland
[2]Princeton Environmental Institute, Guyot Hall, Princeton University, Princeton, NJ 08544, USA
[3]Climate research, Finnish Meteorological Institute, P.O. Box 503, 00101 Helsinki, Finland
[4]Lappeenranta University of Technology, School of Science, Finland
[5]Department of Physical Geography and Ecosystem Science, Lund University, Sölvegatan 12, 22362 Lund, Sweden
[6]Natural Resources Institute Finland (Luke), Latokartanonkaari 9, 00790 Helsinki, Finland
[7]Department of Environmental Sciences, University of Helsinki, Viikinkaari 1, 00790 Helsinki, Finland
[8]School of Forest Sciences, University of Eastern Finland, P.O. Box 111, 80770 Joensuu, Finland
[9]Max Planck Institute for Biogeochemistry, 07745 Jena, Germany
[10]Earth and Environmental Sciences Division, Los Alamos National Laboratory, Bikini Atoll Rd. MS J535, Los Alamos, NM 87545, USA
[11]Max Planck Institute for Meteorology, Bundesstr, 53, 20146, Hamburg, Germany
[12]Department of Forest Sciences, University of Helsinki, P.O.Box 27, 00014 Helsinki, Finland

*Correspondence to*: Maarit Raivonen (maarit.raivonen@helsinki.fi)

**Abstract.** Wetlands are one of the most significant natural sources of methane ($CH_4$) to the atmosphere. They emit $CH_4$ because decomposition of soil organic matter in waterlogged anoxic conditions produces $CH_4$, in addition to carbon dioxide ($CO_2$). Production of $CH_4$ and how much of it escapes to the atmosphere depend on a multitude of environmental drivers. Models simulating the processes leading to $CH_4$ emissions are thus needed for upscaling observations to estimate present $CH_4$ emissions and for producing scenarios of future atmospheric $CH_4$ concentrations. Aiming at a $CH_4$ model that can be added to models describing peatland carbon cycling, we composed a model called HIMMELI that describes $CH_4$ build-up in and emissions from peatland soils. It is not a full peatland carbon cycle model but it requires the rate of anoxic soil respiration as input. Driven by soil temperature, leaf area index (LAI) of aerenchymatous peatland vegetation and water table depth (WTD), it simulates the concentrations and transport of $CH_4$, $CO_2$ and oxygen ($O_2$) in a layered one-dimensional peat column. Here, we present the HIMMELI model structure and results of tests on the model sensitivity to the input data and to the description of the peat column (peat depth and layer thickness), and demonstrate that HIMMELI outputs realistic fluxes by comparing modelled and measured fluxes at two peatland sites. As HIMMELI describes only the $CH_4$-related processes, not the full carbon cycle, our analysis revealed mechanisms and dependencies that may remain hidden when testing $CH_4$ models connected to complete peatland carbon models, which is usually the case. Our results indicated that 1) the model is flexible and robust and thus suitable for different environments; 2) the simulated $CH_4$ emissions largely depend on the prescribed rate of anoxic

respiration; 3) the sensitivity of the total $CH_4$ emission to other input variables is mainly mediated via the concentrations of dissolved gases, in particular, the $O_2$ concentrations that affect the $CH_4$ production and oxidation rates; 4) with given input respiration, the peat column description does not significantly affect the simulated $CH_4$ emissions in this model version.

## 1 Introduction

Methane ($CH_4$) is an important greenhouse gas, atmospheric concentrations of which have increased by more than 250% since preindustrial times, inducing the second largest radiative forcing among well-mixed greenhouse gases (Myhre et al., 2013). Wetlands are the largest single natural $CH_4$ source to the atmosphere and their $CH_4$ emissions respond to changes in climatic conditions, which can be seen at global level (Bridgham et al., 2013; Turetsky et al., 2014). In order to upscale observed $CH_4$ fluxes and to produce realistic scenarios for the future atmospheric greenhouse gas concentrations, it is thus essential to know

how wetland $CH_4$ emissions respond to climatic variables. Modelling these responses has been active in recent years (e.g. Wania et al., 2010; Riley et al., 2011; Melton et al., 2013; Schuldt et al., 2013; Grant et al., 2015).

Freshwater wetlands emit $CH_4$ from decomposition of soil organic matter because oxygen ($O_2$) concentrations in their water-saturated soils are low. Anoxic decomposition of soil organic matter is partly carried out by methanogenic microbes that

produce $CH_4$, so the decomposition process releases both $CH_4$ and carbon dioxide ($CO_2$) (Nilsson and Öquist, 2009). Anoxia has also forced vascular wetland plants to develop techniques to get $O_2$ to their roots that extend to the inundated soil layers. For example, sedge species from genera *Carex* and *Eriophorum*, common in northern fen-type peatlands, have aerenchyma, special tissue with air-filled spaces that allows diffusion of $O_2$ from the atmosphere to the roots (Moog and Brüggemann, 1998). Some aquatic plants transport $O_2$ actively through the aerenchyma with pressurized through-flow (Brix et al., 1996).

As a by-product, these mechanisms also transport $CH_4$ to the atmosphere (Morrissey et al., 1993; Brix et al. 1996). In addition to transfer via plants, $CH_4$ is known to be emitted from peatlands as ebullition, i.e. release of $CH_4$ bubbles into the atmosphere, and by diffusion through the peat column. $CH_4$ can also be consumed in the soil by methanotrophic bacteria that derive their energy by oxidizing $CH_4$ to $CO_2$.

The three transport mechanisms and the $CH_4$ oxidation have been implemented in many peatland models in which the peat column is divided into layers and physically based formulations simulate the carbon processes in them (see a review in Xu et al. 2016). Many of them have features adopted from previous models – for instance, the Walter and Heimann (1996, 2000) model of $CH_4$ production and emission is frequently utilized — but often the implementations include specific modifications. Some of the models also simulate the $O_2$ transport and the simulated $O_2$ concentrations affect the $CH_4$ processes. These models

have been used in multiple studies (e.g., Berrittella and van Huissteden, 2009, 2011; Khvorostianov et al., 2008; Ringeval et al., 2011; Melton et al., 2013; Budishchev et al., 2014; Cresto Aleina et al., 2015; Grant et al., 2015), and some are referred to in the Assessment Report of the Intergovernmental Panel on Climate Change (IPCC; Ciais et al., 2013). These models have

different approaches in simulating the production of $CH_4$, ranging from separating distinct heterotrophic microbial communities (Grant and Roulet, 2002) to taking a constant fraction of the simulated heterotrophic soil respiration (Riley et al., 2011). After that, the transport models essentially take care of determining which portion of the $CH_4$ is oxidized, and which is released to the atmosphere.

As $CH_4$ transport and oxidation can be simulated separately from other soil carbon processes, without the need to feed back to the main soil model, they can form a separate module. There are soil models that simulate anoxic respiration (e.g., Clark et al., 2011; Schuldt et al., 2013) and so this would be their interface to a $CH_4$ module. For this kind of uses, we composed HIMMELI, HelsinkI Model of MEthane buiLd-up and emIssion, which is a module that simulates only the processes related to transport
and oxidation of $CH_4$. It takes the rate of anoxic peat respiration as input, defined here as the rate of anoxic decomposition of organic compounds in peatland soil, and computes the subsequent $CH_4$ emission by simulating the transport and build-up of $CH_4$, $O_2$ and $CO_2$ in the soil, as well as the $CH_4$ oxidation rate that depends on the prevailing $O_2$ concentrations. HIMMELI is driven with soil temperature, water table depth and the leaf area index of the gas-transporting plant canopy.

HIMMELI does not bring any new processes as such into the $CH_4$ model world and it utilizes process descriptions largely adopted from earlier models (e.g., Arah & Stephen, 1998; Tang et al., 2010; Wania et al., 2010). However, it is among the most complete models considering the transport of compounds. According to Xu et al. (2016), there are only 5 models that simulate all vertically resolved biogeochemistry, $O_2$ availability to $CH_4$ oxidation, and three pathways of $CH_4$ transport. Of these, the Xu model (Xu et al., 2007), CLM-Microbe (Xu et al., 2014) and VISIT (Ito and Inatomi, 2012) do not explicitly
simulate $O_2$ transport between the atmosphere and peat. On the other hand, LPJ-WhyMe (Wania et al., 2010), a revised multi-substance version of TEM (Tang et al., 2010), *ecosys* (version in Grant and Roulet, 2002) and a recent model by Kaiser et al. (2017) – not included in the list by Xu et al. (2016) – do simulate all these. HIMMELI also simulates $CO_2$ transport via all three transport pathways. This is not a common feature in $CH_4$ models: to our knowledge, only the multi-substance version of TEM (Tang et al., 2010), *ecosys* (Grant and Roulet, 2002) and the Segers model (Segers and Leffelaar, 2001) included that.
The novelty of HIMMELI is that it has been developed independent of a full peatland carbon model, with the ambition to obtain a robust and flexible model that can be easily used as a tool within different environments as, for instance, its peat column structure is not fixed.

Sensitivity analyses on the complete peatland models have been presented, mostly concentrating on the sensitivity to model
parameters (e.g. Berrittella and Huissteden, 2009, 2011; Tang et al. 2010; Wania et al., 2010; Zhu et al., 2014), but we are not aware of any studies which would have analyzed the sensitivity of the $CH_4$ models as such to driving variables. This kind of analysis is, however, important because a $CH_4$ module can form a considerable part of a peatland carbon model and studying it alone may reveal dependencies that affect the output $CH_4$ emissions but are not seen in sensitivity tests on full carbon models.

Because HIMMELI includes components similar to earlier $CH_4$ models, the results of the sensitivity analysis should be interesting for the modeling community.

In the present work, we a) define key factors for $CH_4$ transport and oxidation, b) describe the model, c) analyze its dynamics and sensitivity of output fluxes to input data in steady-state tests, d) analyze the model sensitivity to the description of the peat column by running the model for a Finnish peatland flux measurement site Siikaneva (Rinne et al., 2007), and e) demonstrate with data from Siikaneva and another site Lompolojänkkä (Aurela et al., 2009) that combined with realistic input, HIMMELI output $CH_4$ fluxes are realistic compared to measurements, which is not so evident if looking only at the mechanistic sensitivity tests.

## 2 Key factors for $CH_4$ transport and oxidation

The rate of $CH_4$ production in peat has been found to be controlled by peat and substrate quality, temperature and pH (Valentine et al., 1994; Bergman et al., 1999; Reiche, 2010). However, the final emissions depend on how much $CH_4$ is consumed by methanotrophic bacteria. This can be up to 100% of the $CH_4$ produced (Whalen, 2005; Fritz et al., 2011). The probability of a $CH_4$ molecule to get oxidized is thought to depend on which pathway it takes to escape from the soil since the conditions are suitable for methanotrophy mostly in oxic peat layers. Ebullition may bypass this oxidative zone (Coulthard et al., 2009) and although methanotrophs are also found in some wetland plant roots (King, 1994), oxidation can largely be avoided by moving through the plants. Several studies have shown that the $CH_4$ emissions decrease clearly when the gas-transporting plants are removed from a site, indicating that aerenchymatous vegetation is an effective transport route for $CH_4$ (Waddington et al., 1996; King et al., 1998; Green and Baird, 2012).

Roots of sedges, particularly those of *Carex* species, extend deep to the soil (Shaver and Cutler, 1979; Saarinen, 1996). Therefore they have a large contact surface with the anoxic peat. The area of root surface permeable to gases was the most important factor controlling the $CH_4$ flux in *Juncus effusus*, another aerenchymatous species, and this permeable surface is concentrated in fine roots and the tips of coarser roots (Hennenberg et al., 2012). According to Reid et al. (2015), the rate for root-mediated gas transport in *P. australis* and *Spartina patens* increased during the growing season, indicating increase of permeable root surface area or aerenchyma along the summer. Thus, the growth of the plants seems to affect their gas transport capacity. Isotopic studies have shown that passive diffusion down the concentration gradient dominates the $CH_4$ transport in sedges (Chanton and Whiting, 1993; Popp et al., 1999), and Moog and Brüggemann (1998) also demonstrated that diffusion is a sufficient explanation for the supply of $O_2$ to the roots of *Carex* species. There are, however, contrasting findings about where the main resistance for the diffusive $CH_4$ flux lies. Kelker and Chanton (1997) suggested it is belowground, at the soil-root or root-shoot boundary, and that *Carex* releases $CH_4$ not through the leaf blades (and stomata) but from the point where the leaves bundle. This would be similar to rice (*Oryza sativa*), *Menyanthes trifoliata* and *J. effusus* that release $CH_4$ from the

stem or leaf sheath, possibly through micropores, not stomata (Nouchi et al., 1990; Macdonald et al., 1998; Hennenberg et al., 2012). However, in the studies by Schimel (1995) and Morrissey et al. (1993), $CH_4$ seemed to exit the sedges through the leaf blades and stomata and this would thus form the main resistance for the flux in the plant. Diurnal variation of the $CH_4$ emissions could indicate stomatal control but clear diurnal patterns have not been observed (Rinne et al., 2007; Jackowicz-Korczyński et al., 2010), the maximum emissions may even occur at night (Mikkelä et al., 1995; Waddington et al., 1996; Juutinen et al., 2004). On the other hand, possible diurnal changes in $O_2$ diffusion to the rhizosphere may be reflected in the $CH_4$ fluxes since $O_2$ concentration affects the rate of $CH_4$ oxidation (Thomas et al., 1996), as well as diurnal changes in the $CH_4$ substrate input from the photosynthesizing vegetation may affect $CH_4$ production (Mikkelä et al., 1995).

Gas ebullition occurs, in principle, when the concentration of a dissolved gas reaches saturation, but in practice, $CH_4$ ebullition has been observed in wetlands already with concentrations below saturation (Baird et al., 2004; Kellner et al., 2006; Waddington et al., 2009; Bon et al., 2014). Other gases increase the gas pressure and soil particles and impurities lower the energy barrier for gas nucleation. The $CH_4$ content in ebullitive gas fluxes has been estimated to be 45 to 60% (Glaser et al., 2004; Tokida et al., 2005; Kellner et al., 2006) and the rest consists mainly of $O_2$, $CO_2$ and nitrogen ($N_2$) (Tokida et al., 2005). The volumetric gas content (VGC) in the peat has been observed to be approximately 10 to 15% (Kellner et al., 2006; Tokida et al., 2007; Waddington et al., 2009) indicating that all the formed gas does not escape the soil. Ebullition events seem to be affected by atmospheric pressure. When the pressure declines, bubble volume increases and the solubility of gases decreases allowing more gases to accumulate in the bubbles, consequently, their buoyancy may overcome the forces that resist their movement and ebullition occurs (Tokida et al., 2007; Waddington et al., 2009). Increasing pressure, by contrast, may enhance the bubble mobility through the peat by causing bubble size to decrease (Comas et al., 2011). Movement of bubbles also depends on the peat structure that varies along the peat column as well as within and between peatlands, due to differences in peat composition and decomposition status (Rezanezhad et al., 2016). The shallow, less decomposed peat has more space for the bubbles, while the more decomposed deeper peat layers are tighter packed (Comas et al., 2011).

Properties of the peat column also affect the diffusion of $CH_4$ and $O_2$ in the air- and water-filled peat pores. Porosity of the soil, i.e., the fraction of the soil volume that is taken up by the pore space, as well as interconnectivity, pore shape and size distribution determine the rate of diffusion. Different descriptions of the dependency of diffusion coefficient on the soil porosity or tortuosity have been presented (Millington, 1959; Collin and Rasmuson, 1988; Staunton, 2008). The porosity of peat soils is generally high, at least 80% (Mitsch and Gosselink, 2007). Therefore, peat does not hinder the diffusion as much as many other soil types. In models the peat column is commonly considered in a simplified way, assuming that the water table depth (WTD) forms a border below which the peat is saturated with water and above which peat pores are air-filled. However, in reality the division is not this strict as VGC can be a considerable fraction of the total volume below the WTD for instance, due to the gas production in the peat (Waddington et al., 2009), and the peat can be wet above the WTD if the peat pores retain water when the WTD drops (Estop-Aragonés et al., 2012; Fan et al., 2014). Diffusion through the peat column is thought to

be a minor component in the total $CH_4$ emissions of a peatland when gas-transporting vegetation is present at the site (Walter et al., 1996; Lai, 2009), because the diffusion coefficient in water is approximately 4 orders of magnitude lower than in gas (Staunton, 2008) and because the probability of $CH_4$ being consumed by methanotrophs is higher in the peat, especially when the WTD is low (Estop-Aragonés et al., 2012).

Methanotrophic bacteria occur in all soils, not only wetlands, and methanotrophy in upland soils is the largest biogenic sink of atmospheric $CH_4$ (Ciais et al., 2013). Rate of the $CH_4$ oxidation reaction depends on the concentrations of both $CH_4$ and $O_2$ (Watson et al., 1997) and since $CH_4$ oxidation is a biochemical reaction, the rate is also limited by factors that affect the microbial activity, such as temperature (Whalen and Reeburgh, 1996). When the WTD is low, the $O_2$ concentrations in the top peat layers are high favoring $CH_4$ oxidation (Moore et al., 2011; Estop-Aragonés et al., 2012). However, there can be anoxic areas above the WTD (Silins and Rothwell, 1999; Fan et al., 2014) and the $O_2$ transported down by plant roots provides conditions suitable for methanotrophy also in the inundated peat layers (Fritz et al., 2011).

## 3 Model and methods

### 3.1 Model description

#### 3.1.1 General

The model (Fig. 1) simulates microbial and transport processes that take place in a one-dimensional peat column, keeping track on the concentration profiles of $CH_4$, $O_2$ and $CO_2$. The output is fluxes of $CH_4$, $O_2$ and $CO_2$ between the soil and the atmosphere, with the possibility to separate the contributions of the three different transport routes, as well as to extract the amount of oxidized $CH_4$. The required input and the model output is explained in more detail within the model code package that is provided as a Supplement of this article. So far the model does not consider freezing and ice, but it is valid when peat water is not frozen. Parameter values used in the present study (Table 1) were based on literature values (see Section 3.2) and the aim was to have physically sound parameter values. However, if using HIMMELI in large-scale $CH_4$ modeling, the model possibly needs to be re-calibrated.

The model is driven with:

- peat temperature, T (K)
- leaf area index of aerenchymatous gas-transporting vegetation, LAI ($m^2$ $m^{-2}$)
- water table depth, WTD (m)
- anaerobic carbon decomposition rate, i.e., the rate of anoxic respiration for the area of the peatland, $V_{anR}$ (mol $m^{-2}$ $s^{-1}$).

The reaction-diffusion equations governing the concentrations of the three compounds $CH_4$, $O_2$ and $CO_2$ at depth $z$ are (Eq. 1-3):

$$\frac{\partial}{\partial t} C_{CH_4}(t, z) = -\frac{\partial}{\partial z} F_{CH_4} - Q_{plt,CH_4} - Q_{ebu,CH_4} + R_{CH_4} - R_O \tag{1}$$

$$\frac{\partial}{\partial t} C_{O_2}(t, z) = -\frac{\partial}{\partial z} F_{O_2} - Q_{plt,O_2} - Q_{ebu,O_2} - R_{aR} - 2R_O \tag{2}$$

$$\frac{\partial}{\partial t} C_{CO_2}(t, z) = -\frac{\partial}{\partial z} F_{CO_2} - Q_{plt,CO_2} - Q_{ebu,CO_2} + (R_{anR} - R_{CH_4}) + R_O + R_{aR} \tag{3}$$

Here, $F_{CH4}$, $F_{O2}$, and $F_{CO2}$ are the diffusive fluxes in the peat (in water below the WTD and in air above it; see Sect. 3.1.8), $Q_{plt,X}$ and $Q_{ebu,X}$ are the transport rates of gas $X$ between peat and atmosphere via plant roots and by ebullition, respectively, $R_{CH4}$ is $CH_4$ production rate, $R_{anR}$ is the rate of anaerobic respiration, $R_{aR}$ is the rate of aerobic respiration and $R_O$ is the $CH_4$ oxidation rate.

The model has been developed principally using a daily timestep for input and output, as our main target has been to use it with models that provide daily input. However, we also tested running HIMMELI on a shorter timestep (Sect. 3.3.2). The internal time step is determined by the turnover time of $CH_4$ and $O_2$ concentrations in the peat. It is assumed that the longest usable time step is half of the turnover time. The differential equations are solved simultaneously using the fourth order Runge-Kutta method.

### 3.1.2 Peat geometry, root distribution and movement of water

The model basically describes a one-dimensional vertically layered peat column. Peat depth and layer thicknesses are not fixed but different set-ups can be used. The only limitation for the layer structure is that if the peat thickness exceeds 2 m, there has to be a layer border exactly at the 2 m depth, because of how the roots are treated in the model. The layering below 2 m must start from that depth.

In the model, WTD is a strict divider of the peat into water-filled and air-filled parts. This has been implemented by adding an extra layer in the pre-described layer composition (Fig. 1). Its thickness is adjusted so that the water surface is always exactly at the interface between the two layers. This approach enables using the exact given WTD as input. Only in the case that the boundary of the extra layer would be closer than 1 cm to a boundary of the background layering, the WTD is rounded to this nearest permanent layer boundary. Strict division of the peat to air-filled and water-filled parts is a simplification since anoxic sites can occur above the WTD (Estop-Aragonés et al., 2012). However, as in site-level and larger scale simulations even an observation-based WTD is an approximate value over peatland areas, we consider the strict division to anoxic and oxic parts a robust approach.

In HIMMELI, the water level can also be above the peat surface and in this case an extra water layer is located above the peat surface. In nature, windmixing can affect the concentrations of different compounds in free water but this is not considered in the model. This simplification is justified as there often is vegetation that decreases the windmixing via affecting wind speed.

Changing WTD essentially means addition or removal of water to/from the peat column. At the same time, the masses of $CH_4$, $O_2$ and $CO_2$ need to be conserved. In the case of rising WTD, the $CH_4$, $O_2$ and $CO_2$ that were in the air-filled layers are dissolved in the water until the concentrations in the newly water-filled layers reach the solubility limit with the previous air concentrations. The excess gas is pushed upwards to the lowest air-filled layer (or to the atmosphere). In the case of lowering WTD, the $CH_4$, $O_2$ and $CO_2$ of the previously water-filled layers are introduced into the air-filled layers replacing them. This
can cause exceptionally high or low fluxes and concentrations in some layers, but these even out fast in relation to the daily time step, mainly through diffusion.

An essential role is played by the vertical distribution of plant roots since that determines how the input anoxic respiration and the gas-transporting root mass is distributed vertically. The formulation has been adopted from Wania et al. (2010):

$$f_{root}(z) = Ce^{-z/\lambda} \tag{4}$$

where $f_{root}(z)$ is the fraction of roots at depth $z$, $\lambda$ is a root depth distribution decay parameter and $C$ is a normalizing constant defined so that the sum of root fractions equals 1 (Eq. 5):

$$\int_0^{z_{max}} f_{root}(z)dz = 1. \tag{5}$$

The maximum depth that the roots are assumed to reach is 2 m (Saarinen, 1996). If the peat depth exceeds 2 m there is a
rootless zone at the bottom. The value of $C$ depends on the peat thickness and geometry of the current peat column and it is calculated at each time step, so the root distribution can adjust to changing peat depth.

### 3.1.3 CH4 production

The input anaerobic respiration ($V_{anR}$) is distributed vertically along the root distribution in the anaerobic peat layers below the WTD (Eq. 6):

$$R_{anR}(z) = \frac{V_{anR}}{dz} f_{root,an}(z). \tag{6}$$

Here $R_{anR}(z)$ (mol m$^{-3}$ s$^{-1}$) is the rate of anoxic respiration at depth z, $f_{root,an}(z)$ refers to the ratio of root mass at depth $z$ to the total root mass of the anaerobic zone and $dz$ (m) is the layer thickness. In the case that peat depth exceeds the maximum rooting depth 2 m, the model calculates what would be the anaerobic respiration rate at the bottom root layer if all the input carbon was allocated in the rooting zone, then allocates 50% of that in the rootless layers, and the remainder is re-distributed to the
rooting zone.

This choice of distributing the anoxic respiration with root mass (as opposed to distributing it e.g. evenly across the peat column) was motivated by the fact that recently fixed carbon, such as root exudates, seems to be the main source of $CH_4$. For instance, according to Oikawa et al. (2017), less than 5% of $CO_2$ and $CH_4$ emissions originate from soils below 50 cm in flooded peatlands. However, in case that HIMMELI is used in a study where it is essential to simulate the different carbon sources and distribute $CH_4$ production in a different way, it is relatively easy to modify the code so that this becomes possible.

$CH_4$ production rate $R_{CH4}$ (mol m$^{-3}$ s$^{-1}$) in a peat layer at depth $z$ is calculated as a fixed fraction ($f_m$) of $R_{anR}$ but the rate may be inhibited by dissolved $O_2$, following Arah and Stephen (1998) (Eq. 7):

$$R_{CH4}(z) = f_m R_{anR}(z) \frac{1}{1+\eta C_{O2}(z)} \,, \tag{7}$$

where $\eta$ is a parameter reflecting the sensitivity of methanogenesis to $O_2$ inhibition. The $CH_4$ production rate in conditions with no $O_2$, i.e., $C_{O2}$ is zero, is called potential methane production (PMP) in this paper. The rest of the anaerobic respiration ($R_{anR}$-$R_{CH4}$) produces $CO_2$. HIMMELI does not include electron acceptors other than $O_2$ since their concentrations can be expected to depend on site characteristics, it would thus be difficult to estimate them and these estimates would not necessarily improve the accuracy of the model. However, including known factors that affect $CH_4$ production, such as the alternative electron acceptors, is important and could possibly be a way to take into account site differences in future model versions.

### 3.1.4 Aerobic respiration

All the $O_2$ in the peat is not consumed by the methanotrophs but other aerobic microbe processes like aerobic peat respiration also require $O_2$. This $O_2$ consumption rate that affects the $O_2$ availability of $CH_4$ oxidation is estimated with a Michaelis-Menten model, following Arah and Stephen (1998) (Eq. 8):

$$R_{aR}(z,T) = V_R(T) \frac{C_{O2}(z)}{K_R+C_{O2}(z)} \,, \tag{8}$$

where $R_{aR}$ (mol m$^{-3}$ s$^{-1}$) is the rate of aerobic respiration at temperature T at depth z, $V_R$ (mol m$^{-3}$ s$^{-1}$) is the potential rate of respiration at temperature T, and $K_R$ (mol m$^{-3}$) is the Michaelis constant for the reaction. This reaction produces 1 mol of $CO_2$ per each mol of $O_2$ consumed.

### 3.1.5 $CH_4$ oxidation

Rate of $CH_4$ oxidation is assumed to follow the dual-substrate Michaelis-Menten kinetics (Arah and Stephen 1998) (Eq. 9):

$$R_O(z,T) = V_O(T) \frac{C_{O2}(z)}{K_{O2}+C_{O2}(z)} \times \frac{C_{CH4}(z)}{K_{CH4}+C_{CH4}(z)}, \tag{9}$$

where $R_O$ (mol m$^{-3}$ s$^{-1}$) is the oxidation rate at temperature T at depth z, $V_O$ (mol m$^{-3}$ s$^{-1}$) is the potential oxidation rate at temperature T, $K_{O2}$ (mol m$^{-3}$) and $K_{CH4}$ (mol m$^{-3}$) are the Michaelis constants for $O_2$ and $CH_4$, respectively. Each $CH_4$ mol oxidized consumes 2 moles of $O_2$ and produces 1 mol $CO_2$.

### 3.1.6 Temperature dependency of microbial reactions

The reaction rates of oxidation and aerobic respiration depend on temperature following the form of the Arrhenius equation (Eq. 10):

$$V(T) = V_\emptyset \, exp\left(\frac{\Delta E}{R}\left(\frac{1}{T_\emptyset} - \frac{1}{T}\right)\right),\qquad(10)$$

where $V(T)$ refers here to the rate of oxidation or aerobic respiration at temperature T, $V_\emptyset$ (mol m$^{-3}$ s$^{-1}$) is the reaction rate at the reference temperature $T_\emptyset$ (K), R (J mol$^{-1}$ K$^{-1}$) is the gas constant and $\Delta E$ (J mol$^{-1}$) the activation energy of the reaction.

### 3.1.7 Ebullition

The ebullition model takes into account concentrations of $CH_4$, $CO_2$, $O_2$ and $N_2$ and uses the sum of their partial pressures to determine when ebullition occurs. This approach was previously used by Tang et al. (2010). In HIMMELI, ebullition is the only process that takes $N_2$ into account. We assume $N_2$ is always in equilibrium with the atmospheric concentration and so its partial pressure in the peat is always 78% of the atmospheric pressure. The model computes the solubilities of $CH_4$, $CO_2$ and

$O_2$ in water using the dimensionless Henry's law coefficient (see Appendix A for formulation; Sander, 2015).

If the sum of the partial pressures $pp$ (Pa) of the dissolved $CH_4$, $CO_2$, $O_2$ and $N_2$ ($pp_X$) exceeds the sum of the atmospheric and hydrostatic pressures ($P_{atm}$ and $P_{hyd}$, respectively) (Eq. 11):

$$\sum_X pp_X(z) > P_{atm} + P_{hyd}(z)\qquad(11)$$

ebullition occurs. The model first computes the fraction of ebullition, $f_e$ (Eq. 12):

$$f_e(z) = \frac{\sum_X pp_X(z) - (P_{atm} + P_{hyd}(z))}{\sum_X pp_X(z)}\qquad(12)$$

and this fraction of each gas is removed, expressed as a rate by introducing time constant $k$ (s$^{-1}$) in the equation. The ebullition rate $Q_{ebu,X}$ (mol m$^{-3}$ s$^{-1}$) of compound $X$ from a soil layer at depth $z$ thus is (Eq. 13):

$$Q_{ebu,X}(z) = -k\frac{f_e(z)pp_X\sigma}{RT},\qquad(13)$$

where σ is peat porosity. Ebullition only occurs in the water-filled peat. If the WTD is below the peat surface, the ebullited gases are transferred into the lowest air-filled soil layer and they continue from there via diffusion in the peat or in plant roots. Otherwise the ebullition is released directly into the atmosphere.

In reality, bubble movement in porous media is a highly complex problem that depends on the fine-scale structure of the media. After a bubble has been formed there are several processes that take place before the bubble reaches the surface and contributes to the CH4 flux to the atmosphere. For instance, the bubbles need to traverse through the peat column and on the way they interact with the surrounding pore water and hence alter the CH4 concentration gradients. These processes are still missing from most of the peatland CH4 models (Xu et al., 2016), including HIMMELI. This is most likely because relatively little is known about bubble movement in peat and how to describe it accurately in models, although there are some attempts to model this process (Ramirez et al., 2015). Different ebullition modelling approaches were compared by Peltola et al. (2017).

### 3.1.8 Diffusion in the peat

Simulation of diffusion in the porous water-filled or air-filled peat takes into account the reduction in the diffusivity compared with pure water or air (see e.g. Iiyama and Hasegawa, 2005). The diffusion coefficients used in this study are listed in Appendix A. The effective diffusivities in the porous peat ($D_{peat,w}$ and $D_{peat,a}$; m$^2$ s$^{-1}$) are calculated by multiplying the free-water or free–air diffusivities by (dimensionless) constant reduction factors $f_{D,w}$ and $f_{D,a}$ (Eq. 14 and 15).

$$D_{peat,w} = f_{D,w}D_w \tag{14}$$

$$D_{peat,a} = f_{D,a}D_a. \tag{15}$$

The diffusion ($F_X$; mol m$^{-2}$ s$^{-1}$) of compound $X$ between layers is calculated using a difference equation that is set up between the centre points ($i$-1 and $i$) of the layers (Eq. 16):

$$F_{i-1,i} = D_{peat,X}\frac{(c_{X,i-1}-c_{X,i})}{dx}. \tag{16}$$

Here $dx$ (m) is the distance between points $i$-1 and $i$ and $C_{X,i-1}$ and $C_{X,i}$ are the concentrations at these layers. The surface layer at the water-air interface is assumed to be in equilibrium with the gas phase concentrations according to the Henry's law. The diffusion flux across the water-air interface is then calculated from the difference in concentration between the layer centre points and water-air interface as shown by Bird et al. (1960). The final equation for the flux of compound $X$ at the interface becomes (Eq. 17):

$$F_X = \frac{2D_{peat,w,X}D_{peat,a,X}}{D_{peat,a,X}+D_{peat,w,X}k_{H,X}}\frac{c_{X,w}-k_{H,X}c_{X,a}}{dx}, \tag{17}$$

where $D_{peat,w,X}$ and $D_{peat,a,X}$ are the diffusion coefficients in the water and air-filled layers, $k_{H,X}$ is the Henry's law coefficient in dimensionless form (Appendix A) and $C_{X,w}$ and $C_{X,a}$ (mol m$^{-3}$) are the concentrations of compound $X$ in the water-filled and air-filled layer, respectively.

### 3.1.9 Plant transport

Formulation of plant transport rate $Q_{plt,X}$ of compound $X$ (mol m$^{-3}$ s$^{-1}$) is similar to many other peatland models in that it describes diffusion in air-filled tubes that represent aerenchymatous plant roots. We employ the formulation from Stephen et al. (1998) that uses the density of cross-sectional area of root endings as the variable expressing the abundance of gas-transporting vegetation (Eq. 18):

$$Q_{plt,X}(z) = \frac{\varepsilon_r(z) D_{peat,a,X}}{\tau} \frac{C_X(z,t) - C_{atm,X}}{z}. \tag{18}$$

Here $\varepsilon_r$ is the density of cross-sectional area of root endings at depth $z$ (m$^2$ m$^{-3}$) and $\tau$ is root tortuosity. To account for the porous structure of aerenchyma (Colmer, 2003), HIMMELI uses the same value as in air-filled peat, $D_{peat,a}$ (m$^2$ s$^{-1}$), as the diffusion coefficient inside roots. It is averaged over the temperatures of the different layers between each depth $z$ that the roots go through. $\varepsilon_r$ follows the root distribution and it depends on the LAI of the vegetation via (Eq. 19):

$$\varepsilon_r(z) = a_{mA} \frac{f_{root}(z)}{dz} \frac{LAI}{SLA}, \tag{19}$$

where $a_{mA}$ expresses the cross-sectional area of root endings per root dry biomass (m$^2$ kg$^{-1}$), $dz$ is the layer thickness (m) and SLA is the specific leaf area (m$^2$ kg$^{-1}$). Root mass is thus assumed to equal the aboveground biomass.

### 3.2 Model parameterization

Table 1 lists the parameter values used in this study, as well as the literature reference of cases where the value was taken directly from one study. Here we go through the parameter values that were based on several papers or some calculation. The parameterization of HIMMELI has been analyzed in more detail in a separate study by Susiluoto et al. (2017).

The CH$_4$ oxidation model has four parameters: $K_{O2}$, $K_{CH4}$, $V_O$ and $\Delta E_O$. Watson et al. (1997) used $K_{O2}$ of 0.032 mol m$^{-3}$ and we chose to use this value rounded to 0.03 mol m$^{-3}$. For $K_{CH4}$ we found several literature values: 0.001 mol m$^{-3}$ in Dunfield et al. (1993), 0.045 and 0.058 in Watson et al. (1997), and 0.001 to 0.045 in the review by Segers (1998). We chose an average of these, i.e., 0.03 mol m$^{-3}$. Dunfield et al. (1993) found that the activation energy of methanotrophy is 20 to 80 kJ mol$^{-1}$ and also here we chose the average, 50 kJ mol$^{-1}$. Using this in the Arrhenius equation (Eq. 10) fitted well with the $V_O$ values reported by Watson et al. (1997) and Dunfield et al. (1993) that were 28 µmol m$^{-3}$ s$^{-1}$ at 25°C and 12 to 15 µmol m$^{-3}$ s$^{-1}$ at 15°C, respectively and thus we set $V_O$ to 10 µmol m$^{-3}$ s$^{-1}$ at the reference temperature $T_\sigma$, 283 K.

The model of aerobic respiration has three parameters: $K_R$, $V_R$ and $\Delta E_R$. Watson et al. (1997) used $K_R$ of 0.022 mol m$^{-3}$ and Iiyama et al. (2012) found in their review a $K_R$ range of approximately 0.002 to 0.02 mol m$^{-3}$. On this basis, we set this to 0.02 mol m$^{-3}$. Stephen et al. (1998) used $\Delta E_R$ value of 50 kJ mol$^{-1}$, which was supported by Lloyd and Taylor (1994), hence, we

also used this value for the activation energy. $V_R$ was based on observed respiration rates on the Siikaneva peatland measurement site (Sect. 3.4.1) that we used in model testing. Respiration rate derived from the mean temperature, mean WTD and mean $CO_2$ emission rate observed in July 2005 at Siikaneva (Aurela et al., 2007) was 16 µmol m$^{-3}$ s$^{-1}$ at 16.5°C. Using the $\Delta E_R$ mentioned above in Eq. 10, $V_R$ at the reference temperature $T_\sigma$ 283 K was approximately 10 µmol m$^{-3}$ s$^{-1}$.

Fraction of anaerobic respiration becoming $CH_4$, $f_m$, affects $CH_4$ generation and therefore also the emission rate directly. According to Nilsson and Öquist (2009), theoretically the $CH_4$ yield from terminal mineralization of soil organic matter in optimal methanogenic conditions ranges from 0 to 70%, being around 50% when carbohydrates are mineralized. Their literature review showed, however, dominance of $CO_2$: the observed $CO_2/CH_4$ quotient in anoxic incubations had varied from

0.5 to 36,000 with median value in a filtered data set being around 6. HIMMELI does not simulate different $CH_4$ production pathways or methanogen groups but uses only this one parameter. We chose to use the conservative ratio 50/50, i.e. $f_m$ of 0.5.

Peat porosity $\sigma$ was based on the review by Rezanezhad et al. (2016) that gave a range 71 to 95%. We chose to use an average value 85%. Reduction factors for the water and air diffusion coefficients in peat, $f_{D,w}$ and $f_{D,a}$, were set by using the model by

Millington and Quirk (1961) (Eq. 20):

$$\frac{D_S}{D_0} = \sigma^{\frac{3}{4}} \tag{20}$$

where $D_s$ is the diffusion coefficient in soil and $D_0$ in free air. The resulting reduction factor was 0.80. We do not know to what extent this applies also to diffusion in water, however, we used the same value for both $f_{D,w}$ and $f_{D,a}$.

SLA values for graminoids or sedges varied widely in literature. Raivonen et al. (2015) found that the SLA of sedges in one peatland site was 7 m$^2$ kg$^{-1}$, Poorter and De Jong (1999) reported the SLA of *Carex* species on a fen to be on average 15 m$^2$ kg$^{-1}$, and Vile et al. (2005) gave 23 m$^2$ kg$^{-1}$ generally for graminoids. We decided to use an average, 15 m$^2$ kg$^{-1}$. Time constant for ebullition, $k$, was set to 1/1800 s based on model numerics, now the half-life of the excess concentrations becomes longer than the usual internal time step.

**3.3 Model testing**

We analyzed HIMMELI's sensitivity to the driving input variables, length of time step, and the description of the peat column, i.e., peat column depth and layer thickness. The model sensitivity to input variables and time step length was analyzed using steady-state tests and transition tests (see Sections 3.3.1 and 3.3.2). The effect of the peat column set-up was analysed by running HIMMELI with data from the Siikaneva peatland site with different peat column descriptions (Section 3.4.1). In

addition, we compared the modelled $CH_4$ fluxes to measured fluxes at Siikaneva and at another peatland site, Lompolojänkkä (Section 3.4.2), in order to demonstrate that when combined with realistic input, HIMMELI outputs realistic $CH_4$ fluxes.

### 3.3.1 Testing model sensitivity to input data

The steady-state tests were conducted to study how sensitive the model is to the input data and to understand how the sensitivity depends on the modelled processes. We tested the model by running it into equilibrium with several different input value combinations, starting from empty concentration profiles of all the compounds. Specifically, we tested the sensitivity of the model to peat temperature, WTD, LAI (and corresponding root mass) and rate of anoxic respiration, by varying these one by one. Temperature was always constant throughout the soil profile in these experiments, unlike in the simulations of the peatland sites. We also conducted three transition tests to study the model response to changing WTD, temperature and anoxic respiration rate. In those, the model was first equilibrated with one set of driver values and after that the WTD, peat temperature or anoxic respiration was alternated. The different input combinations, details of the tests and their names are summarized in Tables 2 and 3.

The tests are labelled so that the first letter (T for temperature, W for WTD, L for LAI and R for respiration) tells which input varied and the rest shows the values of the constant input variables, with the simplification that W03 stands for WTD of -0.3 m. The transition test names just show the changing variables; Wtr stands for WTD transition, Ttr for temperature transition and Rtr for respiration transition. The input range for LAI was based on, e.g., Slevin et al. (2015) and range of anoxic respiration on, e.g., Scanlon and Moore (2000) and Szafranek-Nakonieczna and Stepniewska (2014).

In these mechanistic sensitivity tests, the anoxic respiration rate (mol m$^{-2}$ s$^{-1}$) was independent of temperature and WTD since the purpose was to analyze the sensitivity of the processes that HIMMELI simulates, and anoxic respiration is only input for HIMMELI. We did not want to set any dependency here since it would have meant, in practice, that the test results are valid only when the dependency is as we described it. In this way we kept the tests more generic. The idea was to analyze how much and via what pathways the other driving variables (WTD, temperature, LAI) affect the output $CH_4$ emission rate when the carbon input rate is constant. The input respiration was always allocated only to the inundated peat layers. Consequently, when the WTD varied, also the number of layers into which the anoxic respiration was allocated varied, although the total respiration rate of the peat column remained constant.

### 3.3.2 Testing a time step of 30 min

In order to find out whether eliminating the diurnal temperature variation with the daily time step affects the modelled fluxes we compared a model run done on 30 min time step to a run done on the daily time step. We chose an arbitrary summer day, 1 July 2006, and took the soil and air temperature data measured at Siikaneva at 30 min intervals. All other input values were constant over the day in both runs. To avoid possible complications originating from the fact that the first and last temperatures

of the chosen day differed by 3 degrees (air) and 0.5 degrees (top soil layer) we modified slightly the temperatures measured in the evening. We interpolated new values between the high afternoon temperatures and the new last temperature that was set to be close to the first measurement of the day (Fig. 2). We ran HIMMELI over 35000 days using first these data and a 30 min time step, then using the daily average of the temperatures and a 24-h time step. Within this time, the concentrations reached reasonable saturation. WTD was set to -16 cm, the daily average WTD measured at Siikaneva on 1 July 2006, LAI was 1 $m^2$ $m^{-2}$, and anoxic respiration rate was 1 µmol $m^{-2}$ $s^{-1}$.

### 3.3.3 Testing model sensitivity to the description of the peat column

We ran the model with a seven-year input data series from the Siikaneva fen and tested how sensitive the results are to peat depth and peat layer thicknesses. We used the same input anoxic respiration, WTD and LAI for all the model runs. The only factor that changed slightly between the different set-ups was the soil temperature since the interpolated temperature profile always followed the layering. In these simulations, anoxic respiration was not constant but simulated (see App. B). The model spin-up was conducted by running the model through the entire seven-year time series of input data until the peat $CH_4$ concentrations stabilized. The spin-up time we used depended on the peat thickness, being up to 600 cycles in the case of 5 m peat.

We tested four peat depths, 1 m, 2 m, 3 m and 5 m using 0.2 m layer thickness in every case. In addition, we tested two evenly spaced layerings, 0.1 and 0.2 m, as well as one logarithmic layer structure, in a 2 m deep peat column. The logarithmic structure was based on the one used in the land surface model JSBACH (Ekici et al., 2014) and the layer thicknesses from top to bottom were 0.06, 0.13, 0.26, 0.52 and 1.03 m.

### 3.3.4 Comparison of HIMMELI and measured $CH_4$ fluxes in the Siikaneva and Lompolojänkkä sites

In order to demonstrate that HIMMELI outputs realistic fluxes when run with realistic input – which is not so evident if looking only at the mechanistic sensitivity tests – we compared the modelled and measured $CH_4$ fluxes on two sites, Siikaneva and Lompolojänkkä (Sect. 3.4) using anoxic respiration estimated for the sites as input. The purpose of this comparison also was a general evaluation of what is the significance of using HIMMELI compared to using (simulated) anoxic respiration rate directly as the basis of $CH_4$ emission estimations.

### 3.4 Peatland sites and data

### 3.4.1 Siikaneva site description

The eddy covariance flux measurement site is located in Siikaneva in Ruovesi, Southern Finland (61°49´ N, 24°11´ E, 162 m a.s.l.) (Rinne et al., 2007). The site is a boreal oligotrophic fen where the vegetation is dominated by sedges (*C. rostrata*, *C. limosa*, *E. vaginatum*), Rannoch-rush (*Scheuchzeria palustris*) and peat mosses (*Sphagnum balticum*, *S. majus*, *S. papillosum*).

Peat depth at the measurement footprint is 2 to 4 m. Annual mean temperature in 1971 to 2000 at a nearby weather station was 3.3° C and precipitation 713 mm (Drebs et al., 2002). Siikaneva is a well-established site following the common standards and requirements for eddy-covariance measurements and its characteristics and representativeness of the data has been analyzed in several papers (Aurela et al., 2007; Rinne et al., 2007).

The measurement setup for $CH_4$ fluxes consisted of an acoustic anemometer and a fast response $CH_4$ analyzer. The acoustic anemometer was Metek USA-1 during the whole measurement period, while there were changes in the methane analyzers. The $CH_4$ analyzers used were Campbell TGA-100 (2005 to 2007 and 04/2010 to 08/2010), Los Gatos RMT-200 (2008…2011) and Picarro G1301-f (04/2010 to 10/2011). For $CO_2$ and water vapor fluxes a closed path infrared absorption gas analyzer LiCor 7000 was used. The sonic anemometer and the intake for the $CH_4$ analyzer were at 2.75 m from peat surface. The sample air taken to the TGA-100 was dried using Nafion drier. For RMT-200 and G1301-f sample air was not dried. The measurement setup for 2005 to 2007 has been described in detail by Aurela et al. (2007) and Rinne et al. (2007).

The flux data were post-processed using EddyUH software (Mammarella et al., 2016). The fluxes were calculated using block-averaging and sector-wise planar fitting. High frequency losses were corrected by empirically determined transfer functions (Mammarella et al., 2009). For 2008 to 2011, the dilution effect by water vapor were corrected with Webb-Leuning-Pearman method (Webb et al., 1980), whereas for 2005 to 2007 this correction was not needed due to the usage of a drier in the sampling line.

### 3.4.2 Lompolojänkkä site description

The Lompolojänkkä measurement site is an open, nutrient-rich sedge fen located in the aapa mire region of north-western Finland (67°59.832'N, 24°12.551'E, 269 m above sea level). The vegetation layer is dominated by *Betula nana*, *Menyanthes trifoliata, Salix lapponum* and *Carex spp.* with mean vegetation height of 40 cm and one-sided leaf area index (LAI) of 1.3. The moss cover on the ground is patchy (57% coverage), consisting mainly of peat mosses *(Sphagnum angustifolium, S. riparium and S. fallax)* and some brown mosses *(Warnstorfia exannulata).* The mean annual temperature of -1.4 °C and precipitation of 484 mm have been measured at the nearest long-term weather station of Alamuonio (67°58'N, 23°41'E) during the period 1971 to 2000 (Drebs et al., 2002).

The eddy covariance system used for measuring the vertical $CO_2$ and $CH_4$ fluxes included a USA-1 (METEK) three-axis sonic anemometer/thermometer, a closed-path LI-7000 (Li-Cor, Inc.) $CO_2/H_2O$ analyser and RMT-200 (Los Gatos Research) $CH_4$ analyzer. The measurement height was 3 m and the length of the inlet tubes for the LI-7000 and RMT-200 were 8 m and 15 m, respectively. The mouths of the inlet tubes were placed 15 cm below the sonic anemometer and flow rates of 5 to 6 l min$^{-1}$ and 16 l min$^{-1}$ were used for LI-7000 and RMT-200, respectively. Synthetic air with a zero $CO_2$ concentration was used as the reference gas for LI-7000. For more details of the eddy covariance measurement system, see Aurela et al. (2009).

Half-hour flux values were calculated using standard eddy covariance methods. The original 10-Hz data were block-averaged, and a double rotation of the coordinate system was performed (McMillen, 1988). The time lag between the anemometer and gas analyzer signals, resulting from the transport through the inlet tube, was taken into account in the on-line calculations. An air density correction related to the sensible heat flux is not necessary for the present system (Rannik et al., 1997), but the corresponding correction related to the latent heat flux was made (Webb et al., 1980). Corrections for the systematic high-frequency flux loss owing to the imperfect properties and setup of the sensors (insufficient response time, sensor separation, damping of the signal in the tubing and averaging over the measurement paths) were carried out off-line using transfer functions with empirically-determined time constants (Aubinet et al., 2000). We used here a gapfilled time series, in which measurement gaps were filled with running means.

### 3.4.3 Input data preparation

We forced the model with daily averages of WTD, peat temperature profile, LAI and anoxic respiration rate, and compared the results with daily medians of $CH_4$ flux data from years 2005 to 2011 from Siikaneva and daily averages of $CH_4$ fluxes from years 2006 to 2010 from Lompolojänkkä. Simulations of LAI and anoxic respiration are described in Appendix B.

In Siikaneva, peat temperature has been monitored at five depths, -5 cm, -10 cm, -20 cm, -35 cm and -50 cm, and from Lompolojänkkä we had temperature data from -7 cm and -30 cm depths. We created the temperature profiles by interpolating linearly between the measurements. This was done also for the time step test (Sect. 3.3.2). To obtain temperatures below the deepest measurement points, we assumed that the temperature at -3 meters depth in Siikaneva is constant at +7°C that was the mean temperature of all the years at -50 cm depth (according to the measurements), and at Lompolojänkkä the temperature at -2 m depth is constant +4 °C, the mean temperature of all the years at -30 cm. Gaps in the measurement data were filled by linear interpolation. At Siikaneva, soil temperature data from levels -10 and -40 cm was missing over a longer period so this gap was filled by linear interpolation between the adjacent measurement depths. The main component of the input anoxic respiration for Siikaneva was derived from simulated NPP. The NPP model was driven with the WTD, photosynthetically active radiation (PAR) and air temperature ($T_{air}$). Long gaps in PAR and $T_{air}$ data were filled by using corresponding data from a nearby measurement station SMEAR II (Hari and Kulmala, 2005).

# 4 Results and discussion

## 4.1 Model sensitivity to input data

Via the tests, we wanted to verify that the model dynamics are robust, and to find out how sensitive the output $CH_4$ fluxes are to the input data. Table 4. summarizes the sensitivity results. In the following, we discuss the results, focusing on the most

important aspects and primarily on $CH_4$. It is worth noting that these are results from mechanistic sensitivity tests of HIMMELI, not predictions about responses of $CH_4$ emissions to environmental factors in peatland ecosystems but about how HIMMELI will behave when it is used. For example, the total input anoxic respiration rate here was independent of WTD. WTD only governed the number of peat layers into which this input was distributed and thus the total anoxic respiration rate did not decrease with dropping WTD. Moreover, although soil respiration generally is known to depend on temperature, in these tests

there was no dependency between temperature and anoxic respiration rate, which enabled observing the temperature effect within the processes in HIMMELI.

According to the model, the steady-state dissolved $CH_4$ concentrations increase when moving deeper in the peat column (Fig. 3). This results from the increasing hydrostatic pressure that controls the threshold concentration (pressure) above which gases are released as ebullition. As the solubility of $CO_2$ is higher than that of $CH_4$, the saturated $CO_2$ concentrations were higher

than $CH_4$ concentrations. In the example shown here, ebullition was driven by $CO_2$. This can be seen in the concentration plots: $CH_4$ concentrations did not reach saturation, but stabilized at a value where the sum of the partial pressures of $N_2$, $CO_2$ and $CH_4$ was in balance with the combined atmospheric and hydrostatic pressures. LAI was 0 and thus the only transport route of $O_2$ into the soil was diffusion in water-filled peat pores, therefore, $O_2$ concentrations remained very low.

Contribution of different transport routes in the total $CH_4$ flux varied according to model input. Naturally, when LAI was 0, no $CH_4$ was emitted via plants. Furthermore, because ebullition occurring when the WTD is below the peat surface is transferred to the lowest air-filled peat layer and the gases are then transported by diffusion in dry peat or plant roots (see Sect. 3.1.7), the direct ebullition to the atmosphere occurred only when WTD was at or above the peat surface. Increasing LAI

increased the relative contribution of plant transport in the total $CH_4$ emission in tests L_W0_T10_R1 and L_W03_T10_R1 (Fig 4a; Table 2). Generally, the proportion of plant transport in the total $CH_4$ emissions correlated negatively with the total emission rate, which can be seen in particular in the test R_W0_L1_T10 where LAI was constant 1 and input respiration varied (Fig. 4b). The underlying mechanism here was that high input respiration, i.e. high $CH_4$ and $CO_2$ production, enhanced ebullition (or ebullition followed by transport via diffusion in soil layers above the WTD in cases with WTD < 0) – as could

be expected.

Anoxic respiration rate and the corresponding potential methane production rate (PMP) (tests starting with R_) governed the outputted $CH_4$ emissions. The total emissions depended strongly on the PMP and were only modestly modified by LAI and

WTD. The dependency between PMP and $CH_4$ emission was linear with $R^2$ of 1.0 in the cases that LAI was zero and greater than 0.99 in the cases with LAI of 1 $m^2$ $m^{-2}$ (Fig. 5). The percentage of PMP released as $CH_4$ emission varied between 5% and (almost) 100%, the smallest percentages occurring with the lowest anoxic respiration rates. Generally, the lowest values were obtained from the test R_W0_L1_T10 because this combination allowed the highest inhibiting effect by $O_2$ (the underlying mechanism is discussed below). The highest emissions occurred when both WTD and LAI were zero in test R_W0_L0_T10. The strong dependency between anoxic respiration and $CH_4$ emission was also demonstrated in the transition test (Fig. 6). The increase/decrease in input respiration affected directly the output $CH_4$ emission rate.

In the tests in which the input respiration was constant and we analyzed the sensitivity of $CH_4$ fluxes to LAI, WTD and temperature, the final total steady-state $CH_4$ emission rates varied from 8% to almost 100% of PMP. All the test results combined (Fig. 7), the most important governing factor seemed to be LAI; the high emissions required LAI being zero because that minimized the $O_2$ transport into the soil. Secondarily, WTD controlled the fluxes. The highest emissions occurred when, in addition to zero LAI, WTD was zero or above the peat surface. Effect of temperature was the least important of the input factors, unlike probably in models that describe the total carbon cycle where the rate of anoxic respiration depends on temperature. In our tests, temperature affected only those processes that HIMMELI itself simulates (transport, oxidation, aerobic respiration). However, also with HIMMELI the largest $CH_4$ emissions occurred in the tests with high temperatures.

Although temperature did not affect significantly in steady state, temperature change in the temperature transition tests had a clear effect on the $CH_4$ emissions (Fig. 8). A two-degree abrupt temperature rise throughout the peat column caused the emissions to peak momentarily, before settling to a level only moderately higher than before. The two-degree temperature drops were, correspondingly, followed by a few days clear depression in the emissions, until they gradually recovered back to the normal level. This resulted from temperature transitions changing the gas solubilities and thus the volume of gases available for ebullition.

One interesting result was that the $CH_4$ emissions decreased with decreasing WTD in test W_L0_T10_R1 in which plant transport played no role (Fig. 9a). This was controlled by the oxidation rate that depends on the thickness of the dry oxic peat layer. However, when plant transport was included in W_L1_T10_R1, the highest emissions occurred with the deepest WTD (Fig. 9b) because then the root mass available for transporting $O_2$ into the $CH_4$-producing peat layers was at its lowest. The same trends were obvious in the transition tests with changing WTD (Wtr_L1 and Wtr_L0; Fig 10), dropping WTD caused increasing emissions when LAI was 1 but decreased them when LAI was 0.

The main conclusion that can be deduced from the results reviewed above is that $O_2$ concentration was an important player in the simulations. It affected both the inhibition of $CH_4$ production and oxidation of $CH_4$ to $CO_2$ (Equations 6 and 8). In the tests with constant input respiration (tests ending with _R1), the actualized $CH_4$ production rate varied from 38% to (very close to)

100% of the PMP, and the highest inhibition of $CH_4$ production (i.e., lowest $CH_4$ production) occurred with high LAI that allowed high $O_2$ plant transport into the soil. The same pattern was obvious in the tests on varying input respiration (R_). When LAI was zero, the $CH_4$ production was more or less equal to the PMP. When LAI was 1 and WTD was -0.3 m, the production was 95% to 98% of the PMP. When LAI was 1 and WTD was 0, i.e., all the roots were inundated, the production was at its lowest and varied between 53% and 71% of PMP. This indicates that the more $O_2$ was transported to those soil layers that produced $CH_4$, the less $CH_4$ was produced and consequently emitted. Whether the same production was distributed either in the entire 2 m peat column or only e.g. in the bottom 1.7 m, was significant since in the latter case, there was less $O_2$ transported as a whole to the $CH_4$-producing soil layers, because the greatest root mass is allocated into the topmost peat layers.

The impact of temperature on the output fluxes in the steady-state tests was also transmitted via $O_2$ availability. A one-degree increase in peat temperature increased the total methane emissions on average by 0.09 nmol m$^{-2}$ s$^{-1}$ (0.01 to 0.02%) without gas-transporting vegetation (T_W0_L0_R1) and 1.6 nmol m$^{-2}$ s$^{-1}$ (0.3%) with vegetation (T_W0_L1_R1). The dependencies were linear with $R^2$ of 0.98 and 1.0, respectively. The main reason for this was that in cold temperatures, the solubility of gases, and thus the concentrations of dissolved $O_2$ in water were higher. Therefore, the $CH_4$ oxidation and inhibition of $CH_4$ production were highest in low temperatures although the rates of these reactions were at their lowest (Eq. 9).

The tests thus revealed that $O_2$ transport and other $O_2$-related processes also deserve attention in $CH_4$ modelling, when $O_2$ concentrations are simulated. It is known that the strictly anoxic methanogens are inhibited by $O_2$ (Celis-García et al., 2004) and so it is important to have a proper description of the inhibition process in the $CH_4$ models. $O_2$ transport of aerenchymatous plants has been measured in laboratory conditions (Moog and Brüggemann, 1998) and in the field (Mainiero and Kazda, 2004) but there seem to be no studies in which the simulated plant transport of $O_2$, its dependency on model inputs like LAI or even the dissolved $O_2$ concentrations have been compared with measurements. Measuring $O_2$ fluxes with traditional chambers is challenging because detecting small changes in the high atmospheric $O_2$ concentration (21%) is difficult (Brix and Sorrell, 2013). Consequently, observational $O_2$ data for validating the $O_2$ side of $CH_4$ models is largely lacking.

As mentioned above, effects of the input factors on $CH_4$ emissions may be different when taking the whole peatland carbon cycle into consideration. For example, in test L_W0_T10_R1 high LAI meant high $CH_4$ plant transport capacity that intuitively could mean high $CH_4$ emissions. However, here the impact of increased plant transport of $O_2$ into the soil was so strong that as a result, the total $CH_4$ emissions were lower with high LAI (Fig. 11). Root exudates of gas-transporting plants have been suggested to be a significant source of $CH_4$ substrates (Whiting and Chanton, 1993), and unlike in these sensitivity tests, a greater LAI would probably also mean higher $CH_4$ substrate input in nature. We tested this by setting the input respiration to depend linearly on LAI, assuming zero respiration when LAI=0. In this case, the total $CH_4$ emissions depended on the input respiration and increased with increasing LAI, as could be expected to happen when HIMMELI is connected to a full peatland carbon model.

Direct comparison of our results and sensitivity studies done on other peatland $CH_4$ emission models is not worthwhile because the other studies have analyzed the response of the total peatland carbon model. Some observations can, however, be made. In several studies the parameters affecting the $CH_4$ production rate have been found important (Wania et al., 2010; Berrittella and van Huissteden, 2011), which corresponds to our result that the input anoxic respiration rate affects the output significantly. Wania et al. (2010) tested the effect of tiller porosity on the $CH_4$ emissions and found that at four out of five of their sites, greater porosity increased the total $CH_4$ flux because of enhanced plant transport of $CH_4$, despite the fact that also $O_2$ transport increased. However, in their model, $O_2$ did not affect the $CH_4$ production rate. In our tests, PMP was not dependent on temperature and hence the total effect of temperature was mediated via gas solubilities and rates of oxidation and inhibition. In a complete peatland model, also $CH_4$ production will depend on temperature and as the temperature sensitivity of $CH_4$ production is known to be high (Segers, 1998), probably that would outweigh the other temperature dependencies (Riley et al., 2011). For the development of process-based $CH_4$ models, it is thus useful to analyze the effects of temperature also independently of carbon input. Tang et al. (2010) studied the response of their models to changes in WTD and found that increasing the WTD retarded the $CH_4$ emissions probably because the diffusivity in water is lower than in the air. Whether the increasing WTD affected the total $CH_4$ production is not discussed in their study.

### 4.2 Effect of diurnal temperature variation and time step length

Comparing the outputs of the model run using a 30-min time step with the outputs from the run with a daily time step showed that eliminating the diurnal temperature variation does not have any significant effect on the model output. When using the shorter time step, diurnal variation in the flux was evident and, for instance, a small (around 0.05 to 0.1 degrees) temperature increase throughout the peat column below 0.5 m depth during the last hour caused a clear peak in the emissions (Fig. 12). However, within this set-up, the daily average $CH_4$ emission rate of the 30-min run and the daily output from the 1-day run were equal to two decimal places, 0.27 $\mu$mol m$^{-2}$ s$^{-1}$. The simulation did not relate the anoxic respiration rate to temperature, however, this result indicates that HIMMELI produces consistent output irrespective of the time step length.

### 4.3 Model sensitivity to the description of the peat column

The sensitivity tests with different soil layerings and peat thicknesses conducted using the input data set from Siikaneva site showed that the set-up of the peat column does not have any significant effect on the output. The mean total $CH_4$ flux was between 17.5 and 18.5 nmol m$^{-2}$ s$^{-1}$ for all the set-ups. There were no striking differences in the simulated time series (Fig. 13) and so they all followed the measured $CH_4$ fluxes similarly (Fig. 14a). The same applied to plant transport of $CH_4$; the mean plant-transported flux was approximately 14 nmol m$^{-2}$ s$^{-1}$ in all the cases. Direct ebullition to the atmosphere occurred only a few times during this seven-year simulation and so it was not a significant contribution to the total $CH_4$ emissions (thus not shown). The maximum peak direct ebullition to the atmosphere (daily average) fell between 11 to 12 nmol m$^{-2}$ s$^{-1}$ in all other cases except with the logarithmic layering it was around 17 nmol m$^{-2}$ s$^{-1}$. The remains of the total flux, the mean being between

to 4 nmol $m^{-2}$ $s^{-1}$ in each case, was transported by diffusion in the peat. This diffusion flux contained ebullited $CH_4$ that originated from the water-filled peat layers when the WTD was below the peat surface, which was mostly the case. Also the total $CO_2$ flux was similar in all the set-ups (13d). The mean total $CO_2$ flux was 1.1 to 1.2 μmol $m^{-2}$ $s^{-1}$ in all the cases.

This sensitivity test indicated that when simulating $CH_4$ fluxes with HIMMELI, it is not worthwhile to describe a deep peat column with dense layering because it does not significantly improve the accuracy of the simulation compared with a faster set-up, such as a logarithmic layer structure that is often used in land surface models. The logarithmic layering gave – within the experimental accuracy – similar result as the 10 cm layers, when the input data was the same. Principal reasons probably were that the $CH_4$ production was now allocated mainly to the topmost peat layers, following the vertical root distribution (Eq.
4) and that the $CO_2$ flux was driven by aerobic peat respiration in layers above the WTD. The emission peaks of all the different set-ups coincided in 2010, despite the fact that the peat thicknesses differed. Based on the temperature transition tests, the underlying reason here seemed to be a relatively abrupt temperature rise in peat layers, which did not occur in other years. This, probably together with sinking WTD, triggered ebullition from the water-filled peat layers similarly in all the cases, and the ebullited $CH_4$ is seen as a peak in the diffusion flux.

**4.4 Comparison of modelled and measured $CH_4$ fluxes**

The anoxic respiration inputs created for Siikaneva and Lompolojänkkä (Appendix B) had a clear annual pattern and the rates varied between 0.02 to 0.6 μmol $m^{-2}$ $s^{-1}$ for Siikaneva and between 0.01 to 1.5 μmol $m^{-2}$ $s^{-1}$ for Lompolojänkkä. This magnitude is within literature values. Szafranek-Nakonieczna and Stepniewska (2014) observed anaerobic $CO_2$ production in peat incubations ranging up to around 0.1 g($CO_2$) $kg^{-1}$ (dry weight) $d^{-1}$, which corresponds to around 4 μmol $m^{-2}$ $s^{-1}$ assuming peat
bulk density of 80 g $dm^{-3}$ (Turunen et al., 2002) and 2 m of peat. A model of peat respiration, parameterized by Riutta et al. (2007) using measurement data from a peatland site similar to Siikaneva, gave respiration rate of 0.5 μmol $m^{-2}$ $s^{-1}$ at air temperature of 20°C and WTD of zero (full inundation).

Figure 14. shows the daily observed $CH_4$ fluxes and the $CH_4$ fluxes simulated using the logarithmic layer structure in a 2 m
deep peat column at Siikaneva and Lompolojänkkä. Magnitude of the modelled emissions is comparable to the observed fluxes although there is some difference, especially at Lompolojänkkä. The measured $CH_4$ emissions were on average 80% and 140% of the modelled emissions at Siikaneva and Lompolojänkkä, respectively. It is also clear, especially at Lompolojänkkä, that the simulated annual emission pattern deviates from the observations; the modelled emissions tend to increase too late in spring and decrease too early in the autumn. This may be partly due to a biased presentation of changes in LAI but principally the
reason was a biased annual pattern of input anoxic respiration. The main component of the anoxic respiration was derived directly from simulated daily NPP and it produced $CH_4$ and $CO_2$ immediately, without any time lag, for example, via pools of decomposing organic compounds that could be important at least in the autumn. In reality, as well as in soil carbon models

with which HIMMELI could be combined, there is some lag in the process of carbon fixation turning into root exudates and further to $CH_4$. Most probably both the magnitude and the annual pattern of the emissions can be improved by more realistic simulation of anoxic respiration. However, the model explained the variation in emissions relatively well: the $R^2$ between model and measurement was 0.63 at Siikaneva and 0.70 at Lompolojänkkä.

The simulated $CO_2$ emissions were also at realistic levels both at Siikaneva and Lompolojänkkä. According to Aurela et al. (2007), the mean respiration in Siikaneva in July 2005 was 1.1 to 2.3 µmol m$^{-2}$ s$^{-1}$ and in our simulation, the mean $CO_2$ emission in July 2005 was 2.4 to 2.8 µmol m$^{-2}$ s$^{-1}$ (Fig. 15). At Lompolojänkkä, monthly respiration of July 2006 to 2008 was around 2.5 µmol m$^{-2}$ s$^{-1}$ (Aurela et al., 2009) while the model simulated a $CO_2$ flux of 3.5 µmol m$^{-2}$ s$^{-1}$ (data not shown). The model

overestimated slightly the emissions, especially given that it does not include $CO_2$ from autotrophic respiration unlike the observed fluxes, but the result is still reasonable.

Summer 2010 at Siikaneva was interesting since both model and measurements show the highest emission peaks then. The maximum emissions do not coincide exactly on the same days, but they are temporally close. In HIMMELI, the main reason

was an exceptionally abrupt temperature rise in the peat water, followed by decreasing gas solubilities and increased ebullition – as was observed in the temperature transition tests. Summer 2010 was unusually hot in Finland and so the heat can very well be the cause of the observed high emissions also in nature. We do not know whether the effect really can be transmitted via gas solubilities instead of, for instance, increased respiration. Grant and Roulet (2002) compared simulated and measured $CH_4$ emissions at a beaver pond. Their model captured some bubbling events, driven by warming soil that affected both fermentation

and methanogenesis rates and gas solubilities. In our case, the simulated input anoxic respiration did not increase noticeably during this high-emission period, but our simulation may underestimate the effect of temperature. Moreover, although the soil temperature profile used to run the model was derived from measurements, it was an approximation as it was created by linear interpolation between measurement points. The temperature change of the lower peat layers may be exaggerated compared with reality. However, the modelled $CH_4$ emission peaks nicely matched with observations.

Taking a closer look at Siikaneva only, the model was a slightly better predictor for the measured $CH_4$ emissions than the anoxic respiration as such (Fig. 15), with $R^2$ 0.63 vs. 0.60. Hence, considering the anoxic respiration simulation combined with HIMMELI as one unified $CH_4$ model, HIMMELI slightly improved the fit compared with the anoxic respiration part alone. In the data set shown in the correlation plots (Fig. 15), which was limited to those days from which the measured $CH_4$ fluxes

were available, the $R^2$ between input anoxic respiration and modelled $CH_4$ emissions was 0.65. In the complete simulated time series, this $R^2$ was 0.69 and when correlating the $CH_4$ emissions with anoxic respiration of the previous day, $R^2$ still slightly increased, up to 0.71. In the complete time series, the simulated $CH_4$ emissions were on average 15% of the input anoxic respiration or 30% of PMP. These results support the findings from the sensitivity tests (Section 4.1) that anoxic respiration rate and the corresponding PMP do govern the output $CH_4$ emissions, but indicate also that oxidation and inhibition played a

role in the site simulation of Siikaneva. The temperature responses of anoxic respiration and modelled $CH_4$ emissions were very similar (Fig. 15).

Anoxic respiration alone thus seems a good basis to estimate $CH_4$ emissions but a complete model of $CH_4$ processes is necessary, also in situations when the focus is not on studying concentration profiles or the processes in detail. Simple parameterizations have been tested against process-based $CH_4$ models. For example, Van Huissteden et al. (2009) compared the peatland model PEATLAND-VU that utilizes the Walter-Heimann $CH_4$ scheme, with an emission factor that was based on averages of measurement data on six arctic and temperate wetlands. They found that the model produced a significantly better estimate only on 50% of the sites; on the others, the simple emission factor did better or almost equally well. They concluded, however, that process models are needed for large-scale modelling. Berrittella and van Huissteden (2009) compared PEATLAND-VU to a fixed fraction of NPP as the estimate of $CH_4$ emissions when simulating northern wetlands in glacial climates. In this case, they naturally did not have real-time observational flux data to compare their results with, but they concluded that the two approaches gave *different* results, for instance, the simplistic NPP model produced smaller differences between glacial climates than PEATLAND-VU. A $CH_4$ model like HIMMELI is a significant addition to peatland carbon models, in order to be able to take into account more factors affecting $CH_4$ emissions.

## 5 Conclusions

The new model for simulating $CH_4$ build-up and emissions in peatlands, HIMMELI, is a robust tool to be used as the $CH_4$ emission model in different peatland carbon models. It runs well with different peat column set-ups and within a wide range of inputs. The simulated $CH_4$ emissions are not sensitive to the description of the peat column in case it does not affect the input variables. HIMMELI was able to simulate realistic $CH_4$ fluxes for the Finnish peatland sites Siikaneva and Lompolojänkkä when run with measured and simulated input from the sites.

Sensitivity tests conducted on HIMMELI revealed mechanisms controlling the simulated $CH_4$ emissions that may remain hidden when testing the sensitivity of a full peatland carbon cycle model. Simulated $CH_4$ fluxes largely depended on the input anoxic respiration rate and the corresponding $CH_4$ production rate. This shows that in addition to correct descriptions of $CH_4$ and $O_2$ transport and oxidation processes, it is essential that the underlying $CH_4$ substrate production rates are realistic, in order to produce realistic $CH_4$ emission estimates for different purposes. Other input variables, in particular LAI and WTD, also had an impact on the $CH_4$ emissions in the steady-state tests. With constant input anoxic respiration (which means constant potential $CH_4$ production rate), the total $CH_4$ emission varied from 5 % to almost 100 % of the potential $CH_4$ production, depending on the combination of LAI and WTD. The results indicated that the main factor governing this was the availability of $O_2$ in the peat since its concentration affected the inhibition of $CH_4$ production as well as rates of $CH_4$ oxidation to $CO_2$.

**6 Code and data availability**

The FORTRAN codes of the HIMMELI model are available as a supplement of this article. The data used in these analyses are available upon request.

**Appendix A**

The solubilities of gases are computed following Sander (2015). The temperature (T) dependence of Henry's law constants for the three simulated compounds $CH_4$, $CO_2$ and $O_2$ ($H_X$; M atm$^{-1}$) thus are (Eq. A1-A3):

$$H_{CH4}(T) = 1.3 \times 10^{-3} exp\left[1700\left(\frac{1}{T} - \frac{1}{T^\theta}\right)\right] \tag{A1}$$

$$H_{O2}(T) = 1.3 \times 10^{-3} exp\left[1500\left(\frac{1}{T} - \frac{1}{T^\theta}\right)\right] \tag{A2}$$

$$H_{CO2}(T) = 3.4 \times 10^{-2} exp\left[2400\left(\frac{1}{T} - \frac{1}{T^\theta}\right)\right], \tag{A3}$$

where $T^\theta$ is the reference temperature, 298 K. Temperature dependent diffusivities of the three compounds in water ($D_{X,w}$; m$^2$ s$^{-1}$) and in air ($D_{X,a}$; m$^2$ s$^{-1}$) are calculated following Tang et al. (2010) (Eq. A4-A9). The reference temperature $T^{\theta b}$ used in Equations A7-A9 is 273.15 K.

$$D_{CH4,w}(T) = 1.5 \times 10^{-9} \frac{T}{T^\theta} \tag{A4}$$

$$D_{O2,w}(T) = 2.4 \times 10^{-9} \frac{T}{T^\theta} \tag{A5}$$

$$D_{CO2,w}(T) = 1.81 \times 10^{-6} exp\left(\frac{-2032.6}{T}\right) \tag{A6}$$

$$D_{CH4,a}(T) = 1.9 \times 10^{-5} \left(\frac{T}{T^{\theta b}}\right)^{1.82} \tag{A7}$$

$$D_{O2,a}(T) = 1.8 \times 10^{-5} \left(\frac{T}{T^{\theta b}}\right)^{1.82} \tag{A8}$$

$$D_{CO2,a}(T) = 1.47 \times 10^{-5} \left(\frac{T}{T^{\theta b}}\right)^{1.792}. \tag{A9}$$

**Appendix B**

LAI is not continuously monitored at the peatland sites Siikaneva and Lompolojänkkä, therefore, we utilized the method introduced by Wilson et al. (2007) to obtain LAI input data for the model runs. We simulated the LAI with a lognormal function (Wilson et al., 2007) (Eq. B1):

$$LAI(j) = LAI_{max} \times e^{\left(-0.5\left(\frac{ln\left(\frac{j}{jmax}\right)}{s}\right)^2\right)} \quad (B1)$$

where $LAI_{max}$ is the peak LAI of the growing season, $j$ is the Julian date, $j_{max}$ is the Julian date when the LAI peaks, and $s$ denotes the shape of the curve. Values for the parameters $j_{max}$ and $s$ (Table B1) for Siikaneva were derived from Wilson et al.

(2007) by averaging the values reported for the species abundant at Siikaneva, but for Lompolojänkkä we used different $j_{max}$ as LAI can be expected to peak earlier at the northern latitudes (Raivonen et al., 2015). The growing season peak LAI in the eddy covariance footprint area at Siikaneva was approximately 0.4 $m^2\,m^{-2}$ (Riutta et al., 2007) and 1.3 $m^2\,m^{-2}$ at Lompolojänkkä (Aurela et al., 2009). We also chose to add a constant wintertime LAI in the model since it is known that a significant green sedge biomass, approximately 15% of the maximum, may overwinter (Bernard and Hankinson, 1979; Saarinen, 1998). This

meant overwintering LAI of up to 0.05 $m^2\,m^{-2}$ for Siikaneva and 0.195 for Lompolojänkkä. We used the same LAI for all the years.

The input anoxic respiration was created from two components: simulated net primary production (NPP) and temperature-dependent anoxic peat decomposition $V_{pR}$ (mol $m^{-2}$ $s^{-1}$). As methanogens seem to be keen on fresh, newly fixed carbon

(Couwenberg & Fritz 2012), such as the root exudates, many models relate the $CH_4$ production rate directly with the NPP of the wetland vegetation (Wania et al. 2010, Walter & Heimann 2000, Zhuang et al. 2004). We simply simulated the NPP time series for the sites, allocated the NPP vertically along the root distribution (Eq. 4), and removed the fraction that was in aerobic conditions, i.e., above the WTD (based on the measured WTD time series). The soil profile for which this was computed was 2 m of peat with 0.1 m layers. This NPP was scaled so that the output visually fitted the measured $CH_4$ fluxes at Siikaneva

using a scaling factor $f_s$ of 0.4.

The NPP of Siikaneva was calculated by running models of gross photosynthesis ($P_g$) and autotrophic respiration ($R$). We used the $P_g$ model for a sedge and dwarf shrub canopy by Riutta et al. (2007) (Eq. B2):

$$P_g = P_{max}\,\frac{I}{h+I}\left[1 - e^{-a \times LAI}\right] \times e^{-0.5\left(\frac{T_{air}-T_{opt}}{T_{tol}}\right)^2} \times e^{-0.5\left(\frac{d_W-d_{W,opt}}{d_{W,tol}}\right)^2} \quad (B2)$$

where $P_g$ is the $CO_2$ uptake rate of the canopy (mol $CO_2$ $s^{-1}$ $m^{-2}$ ground surface area), $P_{max}$ is the maximum potential $CO_2$ uptake rate (mol $CO_2$ $s^{-1}$ $m^{-2}$ ground surface area), $I$ (μmol $m^{-2}$ $s^{-1}$) is PAR, $h$ (μmol $m^{-2}$ $s^{-1}$) is PAR at which half of maximum photosynthesis is reached, $a$ is the initial slope of saturating leaf-area response function, LAI is leaf area index (Eq. B1), $T_{air}$ (°C) is air temperature, $T_{opt}$ (°C) is the optimal air temperature for photosynthesis, $T_{tol}$ (°C) is temperature tolerance, $d_W$ (cm) is WTD, $d_{W,opt}$ (cm) is the optimal WTD for photosynthesis, and $d_{W,tol}$ (cm) is WTD tolerance. The parameter values are listed

in Table B1. $R$ (mol $CO_2$ $s^{-1}$ $m^{-2}$) was simulated with a model parameterized for sedges only (Raivonen et al., 2015) (Eq. B3):

$$R = R_{ref} \times LAI \times e^{b\left(\frac{1}{T_{ref}-T_0}-\frac{1}{T_{air}-T_0}\right)} \times e^{-0.5\left(\frac{d_W-d_{W,opt}}{d_{W,tol}}\right)^2},$$
(B3)

where $R$ is the $CO_2$ release rate of the canopy, $R_{ref}$ (mol $CO_2$ $s^{-1}$ $m^{-2}$ leaf area) is the $CO_2$ release rate per unit of leaf area under reference conditions, $b$ (K) is an exponential parameter depicting the temperature sensitivity of respiration, $T_{ref}$ (K) is the reference temperature, and $T_0$ (K) is the temperature at which respiration reaches zero (Table B1).

The daily averages of net photosynthesis $P_n$ (mol $CO_2$ $s^{-1}$ $m^{-2}$) were calculated as the difference between $P_g$ and $R$. Photosynthetically active seasons were determined by searching for dates of snowmelt in spring or arrival of snow cover in autumn from the reflected PAR data or, in some cases, using air temperature (permanently > 5°C) as the criterion. No direct measurements of $P_n$ or vascular NPP exist for validation but the simulated $P_n$ of year 2005 was compared with an NPP estimate

derived from eddy covariance $CO_2$ fluxes measured that year on Siikaneva. Briefly, the estimated contributions of *Sphagnum* mosses (30%; Riutta et al., 2007) and autotrophic respiration (50%; Gifford, 1994) were subtracted from the eddy-covariance based gross primary productivity (GPP) (Aurela et al., 2007; data obtained via personal communication), and the remains were taken as an estimate of the NPP of vascular vegetation. The two NPP estimates were well correlated (with $R^2$ of 0.9) but the eddy-covariance based NPP was on average 1.56-fold compared with the simulated $P_n$. Since the latter also was lowish

compared with what has been reported for similar peatlands, the final estimate of NPP for years 2005 to 2011 was produced by scaling the simulated $P_n$ upwards by 1.56.

For Lompolojänkkä, the GPP time series over years 2006 to 2010 was available (Aurela et al., 2009), thus, we derived the NPP of vascular vegetation directly from the GPP data. Again we assumed that autotrophic respiration contributes 50% to the GPP

(Gifford, 1994) and the contribution of *Sphagnum* was estimated to be 10%, based on the biomass values reported for Siikaneva and Lompolojänkkä (Li et al., 2016).

The anoxic peat respiration for both sites was computed for the peat layers below WTD using the $Q_{10}$ model for catotelm decomposition presented in Schuldt et al. (2013) (Eq. B4):

$$V_{pR} = \sum_{z_{min}}^{WTD} Q_{10}^{\frac{T(z)-T_{ref,pR}}{10}} \frac{1}{\tau_c} \rho_C dz.$$
(B4)

Here $Q_{10}$ is the base for temperature dependence of respiration, $T_{ref,pR}$ is reference temperature for peat respiration (K), $\tau_{cato}$ is turnover time of the catotelm carbon pool (s) and $\rho_C$ (mol (C) $m^{-3}$) is the density of the carbon pool. The parameter values were taken from Schuldt et al. (2013) except for the Q10 we used a higher value 3.5 that was the average Q10 found by Szafranek-Nakonieczna and Stepniewska (2014) (Table B1).

[Supplement link]

*Author contribution:* S. Smolander and L. Backman developed the model. M. Raivonen participated in model development and designed and carried out the tests with contribution from L. Backman, J. Susiluoto, T. Aalto, T. Markkanen, J. Mäkelä and T. Vesala. M. Tomasic, X. Li, M. Heimann, S. Sevanto, T. Kleinen and V. Brovkin contributed to the model development. J. Rinne, O. Peltola, M. Aurela and A. Lohila provided observational data from the Siikaneva and Lompolojänkkä sites. T. Larmola, S. Juutinen and E.-S. Tuittila provided knowledge and advice about peatland methane processes for model development. M. Raivonen prepared the manuscript with contributions from all co-authors.

## Competing interests

The authors declare that they have no conflict of interest.

## Acknowledgements

We thank the Academy of Finland Center of Excellence (272041), Academy Professor projects (284701 and 282842), CARB-ARC (285630), ICOS Finland (281255), NCoE eSTICC (57001), EU-H2020 CRESCENDO (641816) and MONIMET (LIFE12 ENV/FI/000409) and Maj and Tor Nessling Foundation (projects 2008336, 2009067 and 2010212) for support. Academy of Finland is also acknowledged by E-S.T. (project 287039) and T.L. (121535, 286731 and 293365). T.K. acknowledges funding by the German ministry for research (BMBF) in projects CarboPerm and PalMod.

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

**Table 1: Model parameters and their values. The reference is given in cases where the value is directly from one study, otherwise the parameter value is discussed in Section 3.2.**

| Symbol | Definition | Value | Reference |
|---|---|---|---|
| $\lambda$ | decay length (in root distribution) | 0.2517 | Wania et al. (2010) |
| $f_m$ | fraction of anaerobic respiration becoming methane | 0.5 | |
| $V_R$ | potential rate of aerobic respiration at 10°C [mol m$^{-3}$ s$^{-1}$] | $1\times10^{-5}$ | |
| $K_R$ | Michaelis constant for aerobic respiration reaction [mol m$^{-3}$] | 0.02 | |
| $V_O$ | potential oxidation rate at 10°C [mol m$^{-3}$ s$^{-1}$] | $1\times10^{-5}$ | |
| $K_{O2}$ | Michaelis constant for $O_2$ in oxidation [mol m$^{-3}$] | 0.03 | |
| $K_{CH4}$ | Michaelis constant for $CH_4$ in oxidation [mol m$^{-3}$] | 0.03 | |
| $\Delta E_R$ | activation energy of aerobic respiration [J mol$^{-1}$] | 50000 | Stephen et al. (1998) |
| $\Delta E_O$ | activation energy of oxidation [J mol$^{-1}$] | 50000 | |
| $T_\phi$ | reference temperature for oxidation and aerobic respiration [K] | 283 | |
| $k$ | time constant of ebullition [s$^{-1}$] | 1/1800 | |
| $a_{mA}$ | root ending area per root dry biomass [m$^2$ kg$^{-1}$] | 0.085 | Stephen et al. (1998) |
| $\tau$ | root tortuosity | 1.5 | Stephen et al. (1998) |
| SLA | specific leaf area of gas-transporting plants [m$^2$ kg] | 15 | |
| $f_{D,w}$ | reduction factor for diffusion in water-filled peat | 0.8 | |
| $f_{D,a}$ | reduction factor for diffusion in air-filled peat | 0.8 | |
| $\eta$ | sensitivity of methanogenesis to oxygen [m$^3$ mol$^{-1}$] | 400 | Arah and Stephen (1998) |
| $\sigma$ | peat porosity | 0.85 | |

**Table 2: Summary of the steady-state sensitivity tests in which response of HIMMELI to different input combinations was analyzed.**

| Test name | T (°C) | WTD (m) | LAI ($m^2$ $m^{-2}$) | Anoxic respiration ($\mu$mol $m^{-2}$ $s^{-1}$) |
|---|---|---|---|---|
| T_W0_L0_R1 | 5, 10, 20, 25 | 0 | 0 | 1 |
| T_W0_L1_R1 | 5, 10, 20, 25 | 0 | 1 | 1 |
| L_W0_T10_R1 | 10 | 0 | 0, 0.5, 1, 2, 3 | 1 |
| L_W03_T10_R1 | 10 | -0.3 | 0, 0.5, 1, 2, 3 | 1 |
| W_L0_T10_R1 | 10 | -0.5, -0.3, -0.2, -0.1, 0, 0.05 | 0 | 1 |
| W_L1_T10_R1 | 10 | -0.5, -0.3, -0.2, -0.1, 0, 0.05 | 1 | 1 |
| R_W0_L0_T10 | 10 | 0 | 0 | 0.01, 0.1, 0.5, 1, 5, 10 |
| R_W0_L1_T10 | 10 | 0 | 1 | 0.01, 0.1, 0.5, 1, 5, 10 |
| R_W03_L0_T10 | 10 | -0.3 | 0 | 0.01, 0.1, 0.5, 1, 5, 10 |
| R_W03_L1_T10 | 10 | -0.3 | 1 | 0.01, 0.1, 0.5, 1, 5, 10 |

**Table 3: Summary of the transition tests on model sensitivity to input data and the input combinations used in the tests.**

| Test name | T (°C) | WTD (m) | LAI ($m^2$ $m^{-2}$) | Anoxic respiration ($\mu mol$ $m^{-2}$ $s^{-1}$) |
|---|---|---|---|---|
| Wtr_L1 | 10 | 0, -0.2, -0.4, -0.2, 0 | 1 | 1 |
| Wtr_L0 | 10 | 0, -0.2, -0.4, -0.2, 0 | 0 | 1 |
| Rtr_W0_L1 | 10 | 0 | 1 | 0.5, 1, 2, 1, 0.5 |
| Rtr_W0_L0 | 10 | 0 | 0 | 0.5, 1, 2, 1, 0.5 |
| Ttr_W0_L1 | 10, 12, 14, 12, 10 | 0 | 1 | 1 |
| Ttr_W0_L0 | 10, 12, 14, 12, 10 | 0 | 0 | 1 |

**Table 4: Results of the sensitivity testing. The rightmost column tells how much the CH$_4$ emissions changed when the input changed. The +/- signs in front of 'Input change' and 'Change in CH$_4$ emission' show the directions of change in input and the corresponding response in CH$_4$ emissions. This is expressed as % of PMP (see Sect. 3.1.3) for the first 6 tests and as % of change in input anoxic respiration for the tests on changing input respiration. In most cases, the response was not constant over the input range and therefore, the result is also expressed as a range.**

| Test | Changing input variable | Input change | Change in CH$_4$ emission, % of potential production/ % of change in respiration |
|------|------------------------|--------------|-----------------------------------------------------------------------------------|
| T_W0_L0_R1 | temperature | +1° | +0.01%…0.02% |
| T_W0_L1_R1 | temperature | +1° | +0.3% |
| L_W0_T10_R1 | LAI | +0.1 m$^2$ | -13%...-0.3% |
| L_W03_T10_R1 | LAI | +0.1 m$^2$ | -1.8%…-1.4% |
| W_L0_T10_R1 | WTD | -0.05 m | -1.4%...-0.2% |
| W_L1_T10_R1 | WTD | -0.05 m | -0.02%...+12% |
| R_W0_L0_T10 | respiration | + | +98%...100% |
| R_W0_L1_T10 | respiration | + | +7%...71% |
| R_W03_L0_T10 | respiration | + | +95%...97% |
| R_W03_L1_T10 | respiration | + | +20%...96% |

**Table B1: Parameter values of the models used for producing input for the Siikaneva and Lompolojänkkä runs. The value marked with * is the only one specific for the Lompolojänkkä site. The parameter value marked with ** is fitted in this study and the value *** is based on Szafranek-Nakonieczna and Stepniewska (2014), the others are from the original references of the photosynthesis and respiration models.**

| Symbol | Definition | Value |
|---|---|---|
| $P_{max}$ | maximum potential $CO_2$ uptake [mol (C) s$^{-1}$ m$^{-2}$ ground area] | $1.24 \times 10^{-5}$ |
| $k$ | PAR at which half of maximum photosynthesis is reached [μmol m$^{-2}$ s$^{-1}$] | 223.9 |
| $a$ | initial slope of saturating leaf-area response function | 0.778 |
| $T_{opt}$ | optimal air temperature [°C] | 24.88 |
| $T_{tol}$ | temperature tolerance [°C] | 14.69 |
| $d_{W,opt}$ | optimal water table depth [cm] | -29.1 |
| $d_{W,tol}$ | water table depth tolerance [cm] | 67.27 |
| $R_{ref}$ | respiration rate in reference conditions  [mol (C) s$^{-1}$ m$^{-2}$ leaf area] | $6.94 \times 10^{-7}$ |
| $b$ | activation energy/gas constant [K] | 300 |
| $T_{ref}$ | reference temperature of autotrophic respiration [K] | 283.15 |
| $T_0$ | T at which R = 0 [K] | 227.13 |
| $LAI_{max}$ | peak LAI | 0.4 |
| $LAI_{min}$ | overwintering LAI | 0.05 |
| $j_{max}$ | Julian date of the peak LAI | 209/190* |
| $c$ | parameter to adjust the LAI curve shape | 0.2 |
| $f_s$ | NPP scaling factor | 0.4** |
| $R_{ref,pR}$ | reference temperature of peat respiration [K] | 273.15 |
| $Q_{10}$ | base value for temperature dependence of peat respiration | 3.5*** |
| $\tau_C$ | turnover time of the catotelm carbon pool [y] | 30 000 |
| $\rho_C$ | density of the carbon pool [mol (C) m$^{-3}$] | 6277.73 |

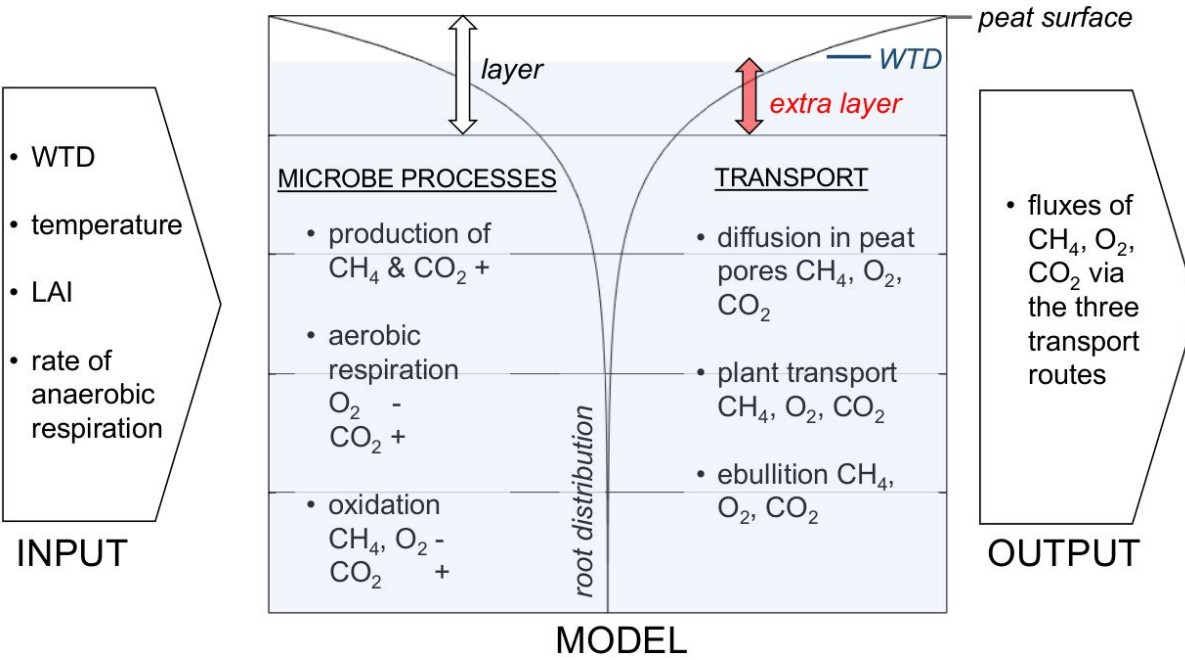

**Figure 1. HIMMELI as a simplified schematic picture. The microbial and transport processes are simulated in a vertically layered one-dimensional peat column in which roots of aerenchymatous gas-transporting plants are distributed according to the exponential root distribution function. The input anoxic respiration is distributed along the root distribution. Input water table depth (WTD) determines the thickness of the possible extra layer that is introduced in case the WTD does not match any of the fixed background layer borders. This ensures that all the simulated layers are either completely water-filled or air-filled. The + sign shows that the compound is produced in the microbial process and – sign means consumption of the compound.**

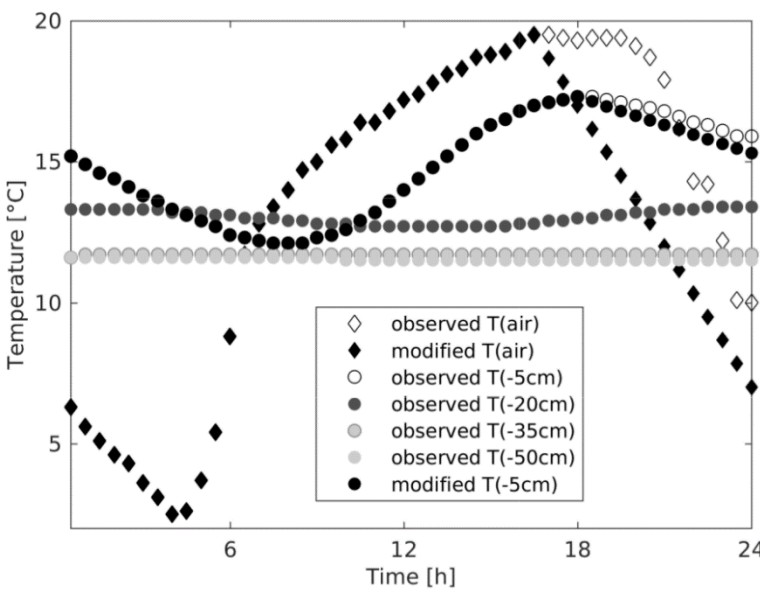

**Figure 2. Daily variation of air and soil temperatures in the time step test. Observed temperatures are directly from measurement data but in order to smooth the difference between the last and first temperatures of the day, we modified the afternoon temperatures as shown in the plot.**

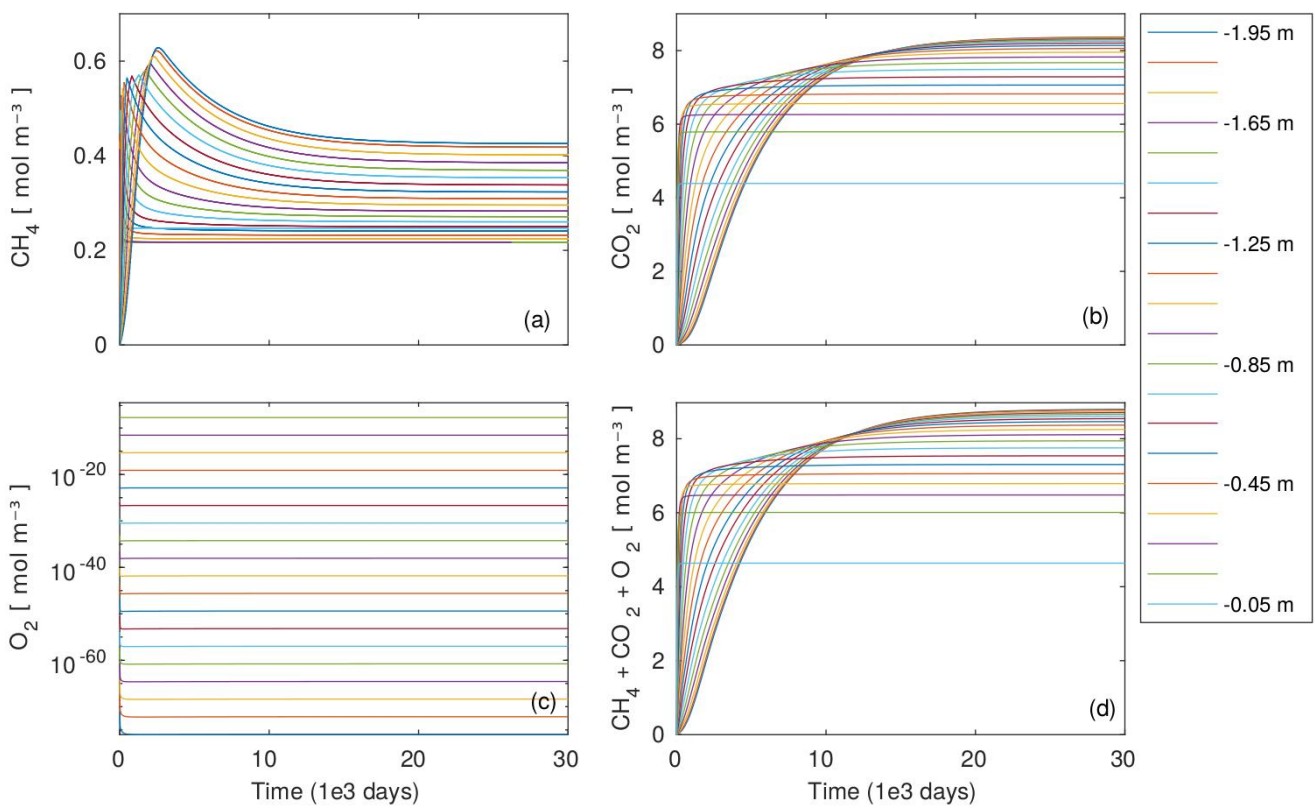

**Figure 3. Evolution of the concentration profiles of (a) CH₄, (b) CO₂ (c) O₂ and (d) their sum in a simulation where both WTD and LAI were zero, i.e., there was no plant transport of these compounds. Different colors show the concentrations at different depths in the peat. In the beginning of the simulation, all the concentrations were zero.**

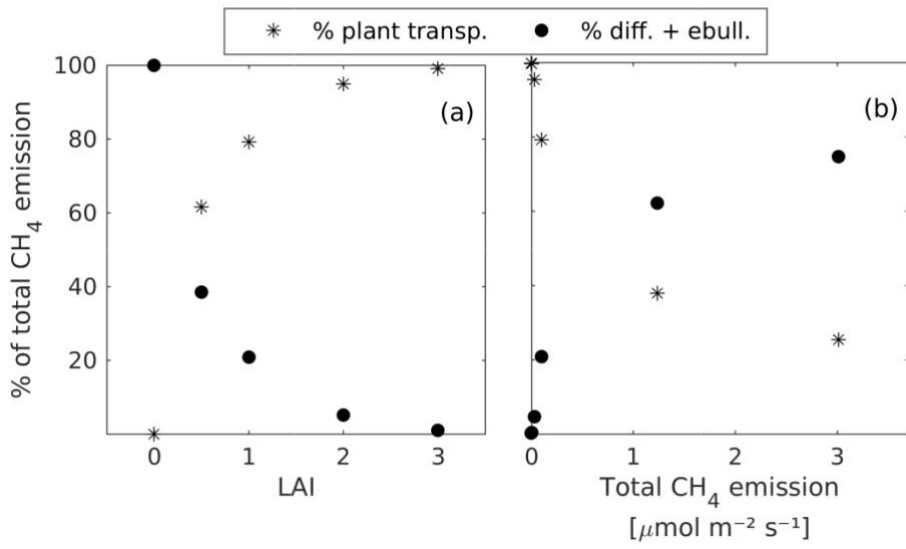

**Figure 4.** Contribution of different transport routes to the total CH₄ emission (a) as a function of LAI in test L_W0_T10_R1 and (b) as a function of total CH₄ emission in test R_W0_L1_T10.

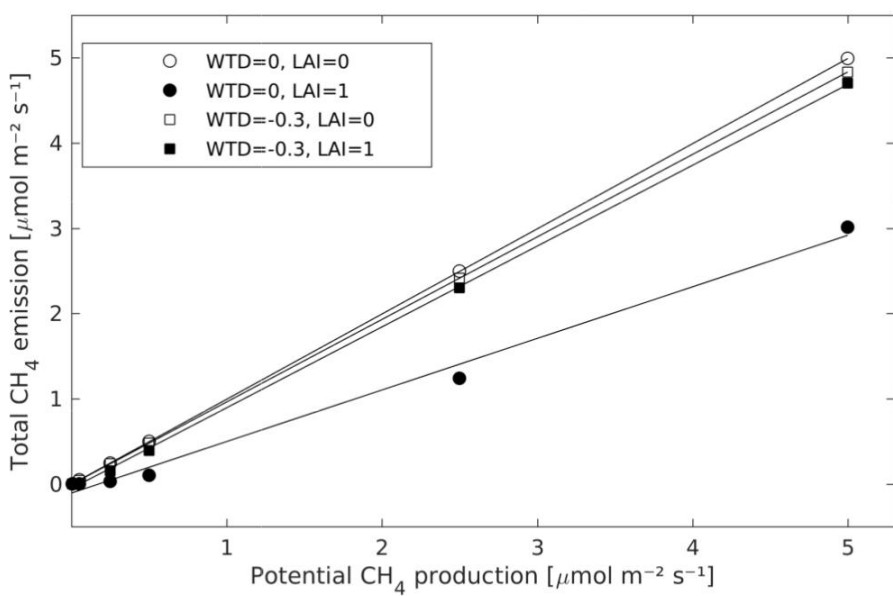

**Figure 5. Dependence of the total output CH4 emission on the potential CH4 production rate in tests on the model sensitivity to input anoxic respiration, i.e. tests that were named starting with R_.**

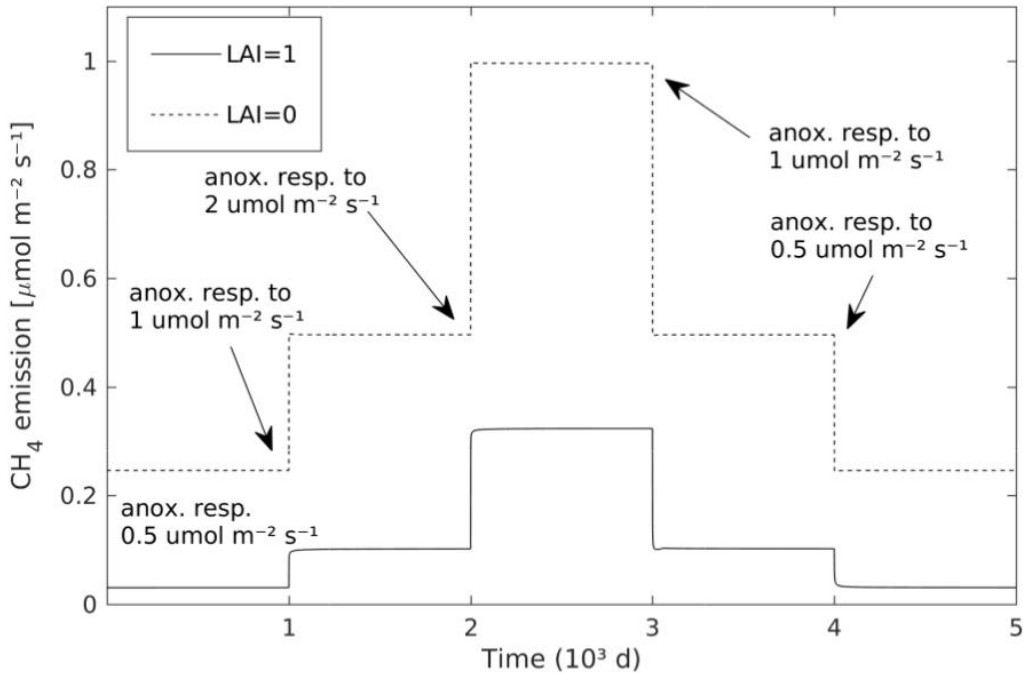

**Figure 6. Output CH$_4$ emission responded clearly to changes in the input anoxic respiration rate in the transition tests Rtr_W0_L1 (solid line) and Rtr_W0_L0 (dashed line) (see Table 3). Black arrows indicate when the input changed.**

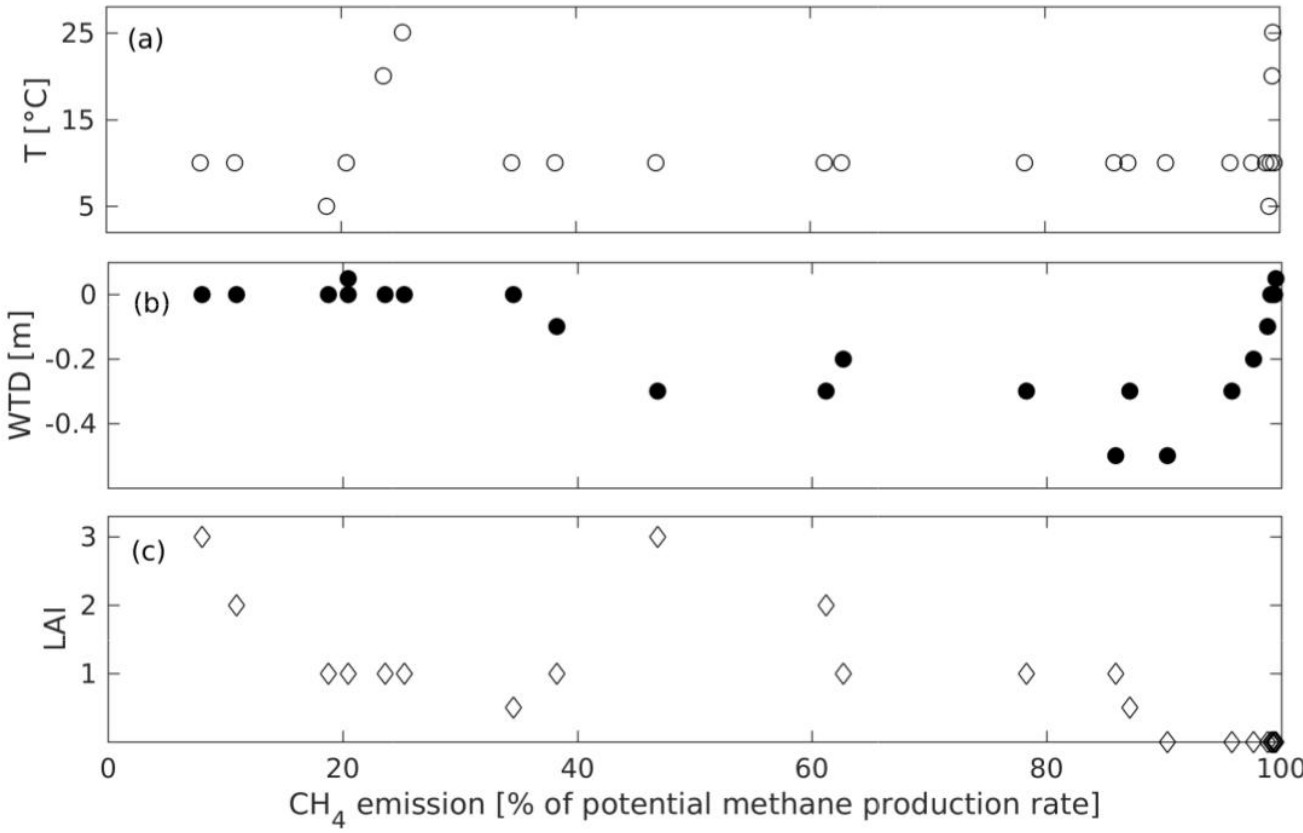

**Figure 7. Relationship between the relative CH₄ emission rate (expressed as % of PMP) and different combinations of input (a) temperature, (b) WTD and (c) LAI in the steady-state sensitivity tests with constant anoxic respiration (test names ending with _R1).**

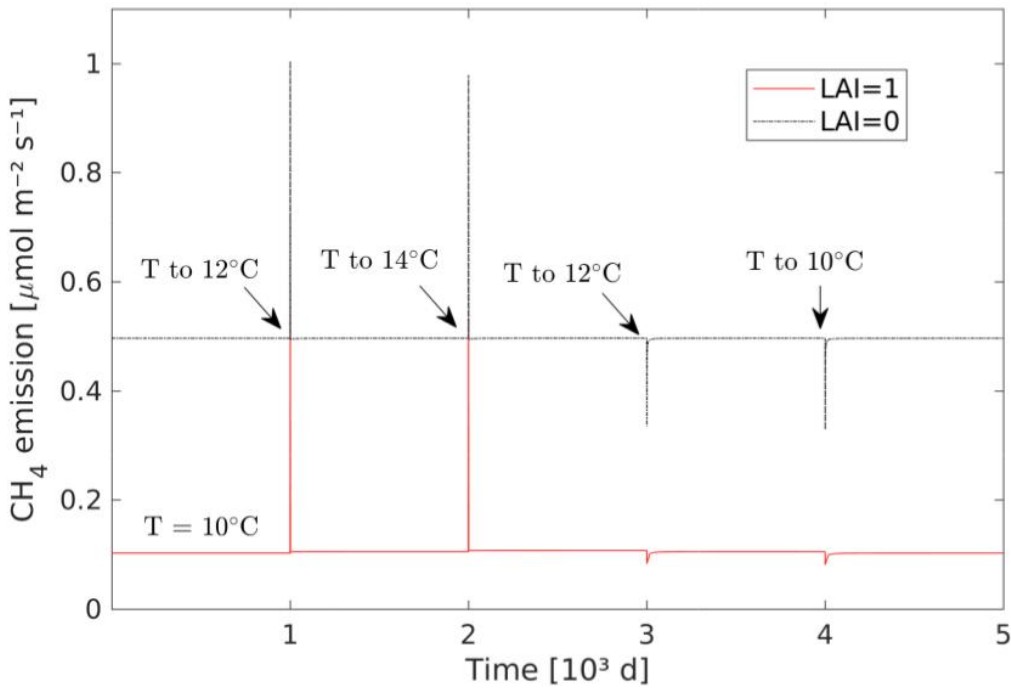

**Figure 8. Response of CH$_4$ emission to changes in peat temperature in the transition tests Ttr_W0_L1 (red line) and Ttr_W0_L0 (black dashed line) (see Table 3). Black arrows indicate when the input changed.**

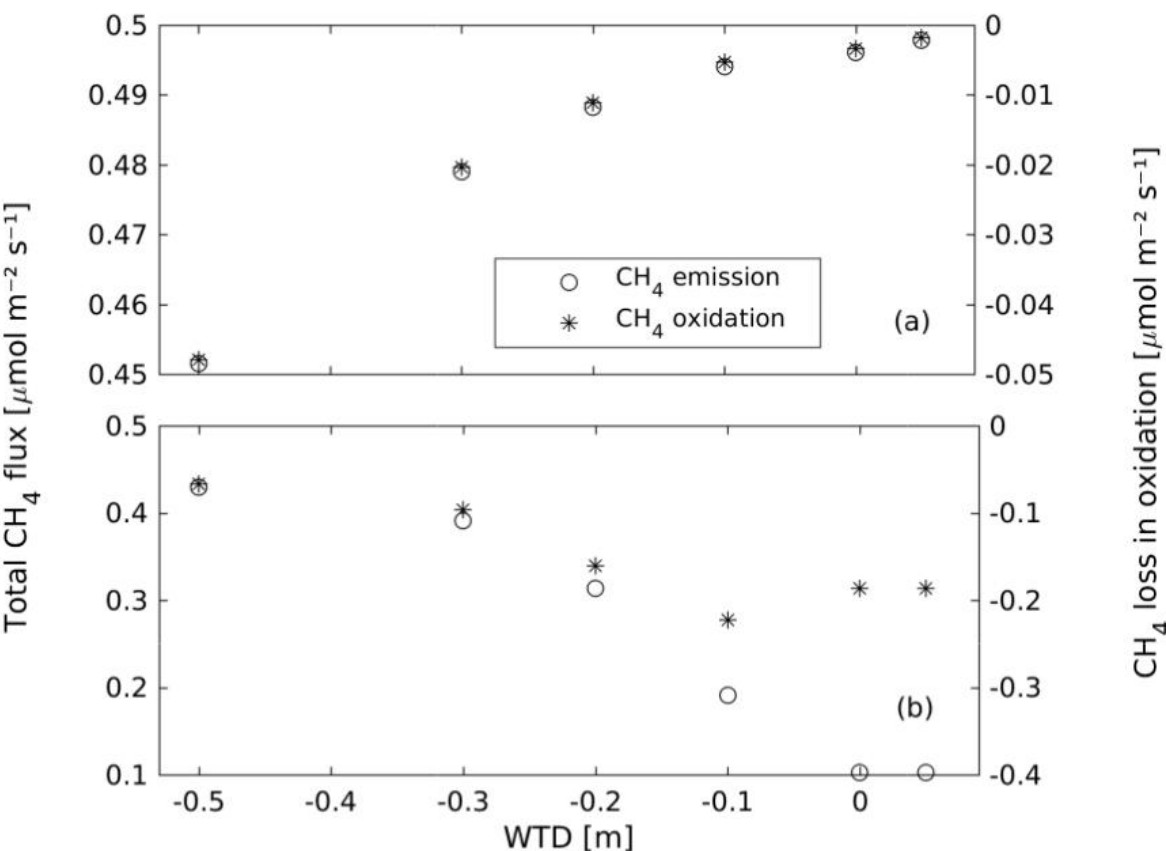

**Figure 9. Dependence of the total CH₄ flux and CH₄ oxidation rate on WTD in (a) test W_L0_T10_R1 and (b) test W_L1_T10_R1. CH₄ oxidation is a negative flux since it is loss of CH₄.**

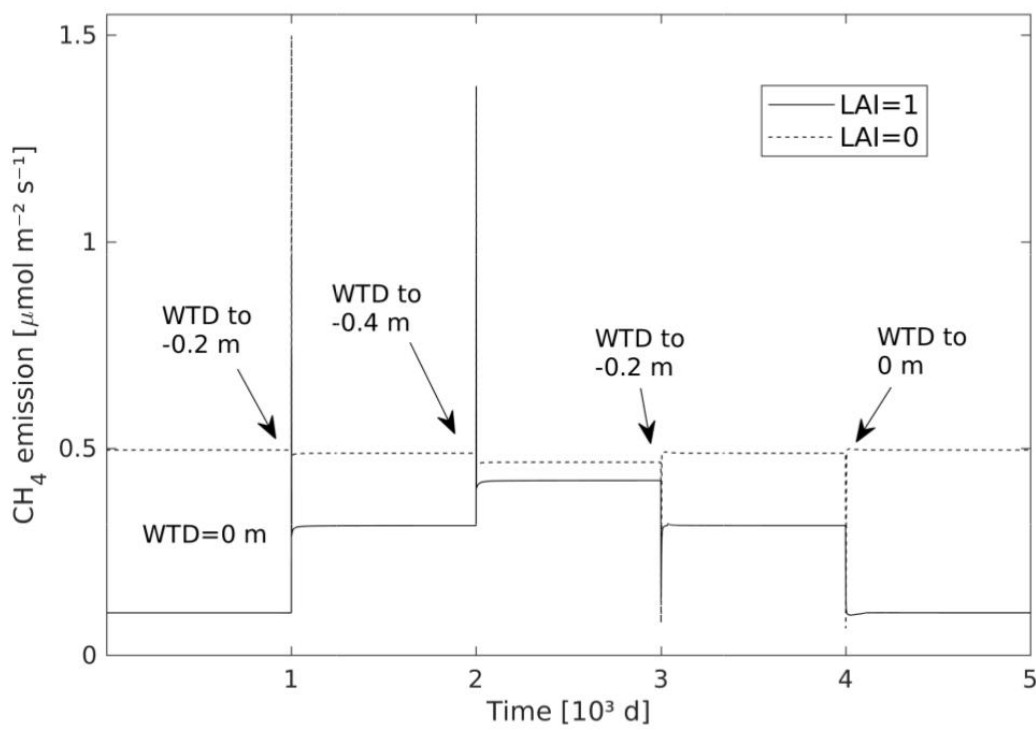

**Figure 10. Effect of abrupt changes in WTD on the total output CH₄ emissions in transition tests Wtr_L0 (dashed line) and Wtr_L1 (solid line). Black arrows indicate the change in WTD. This figure also shows how changes in the WTD cause a short peak in the flux, because of how the CH₄ (and CO₂ and O₂) in layers receiving or losing water is handled in the model (see Sect. 3.1.2).**

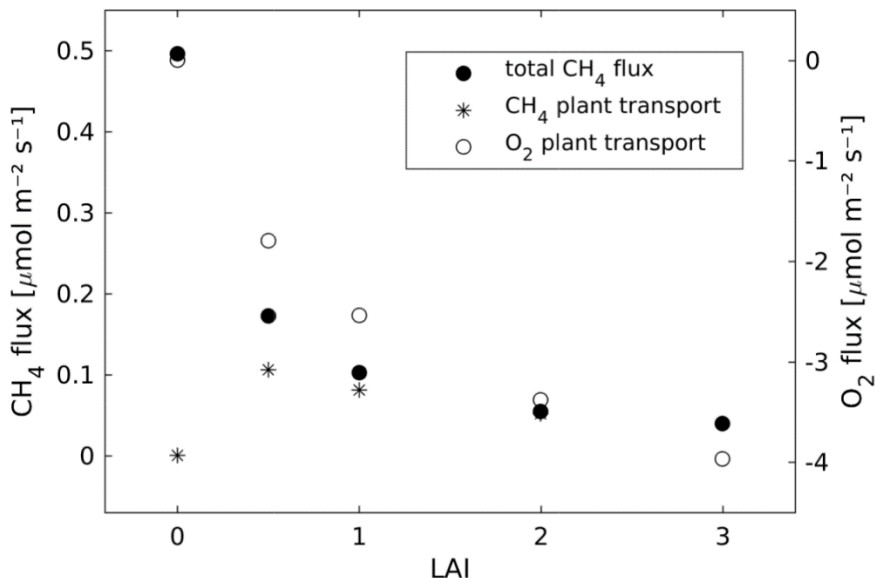

**Figure 11. Dependence of total and plant-transported fluxes of CH$_4$ and plant transport of O$_2$ on LAI in test L_W0_T10_R1.**

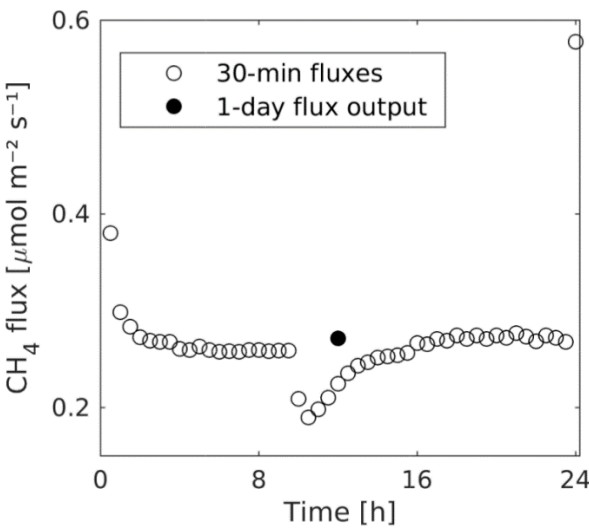

**Figure 12. Daily CH₄ flux in the test comparing 30 min and daily time steps.**

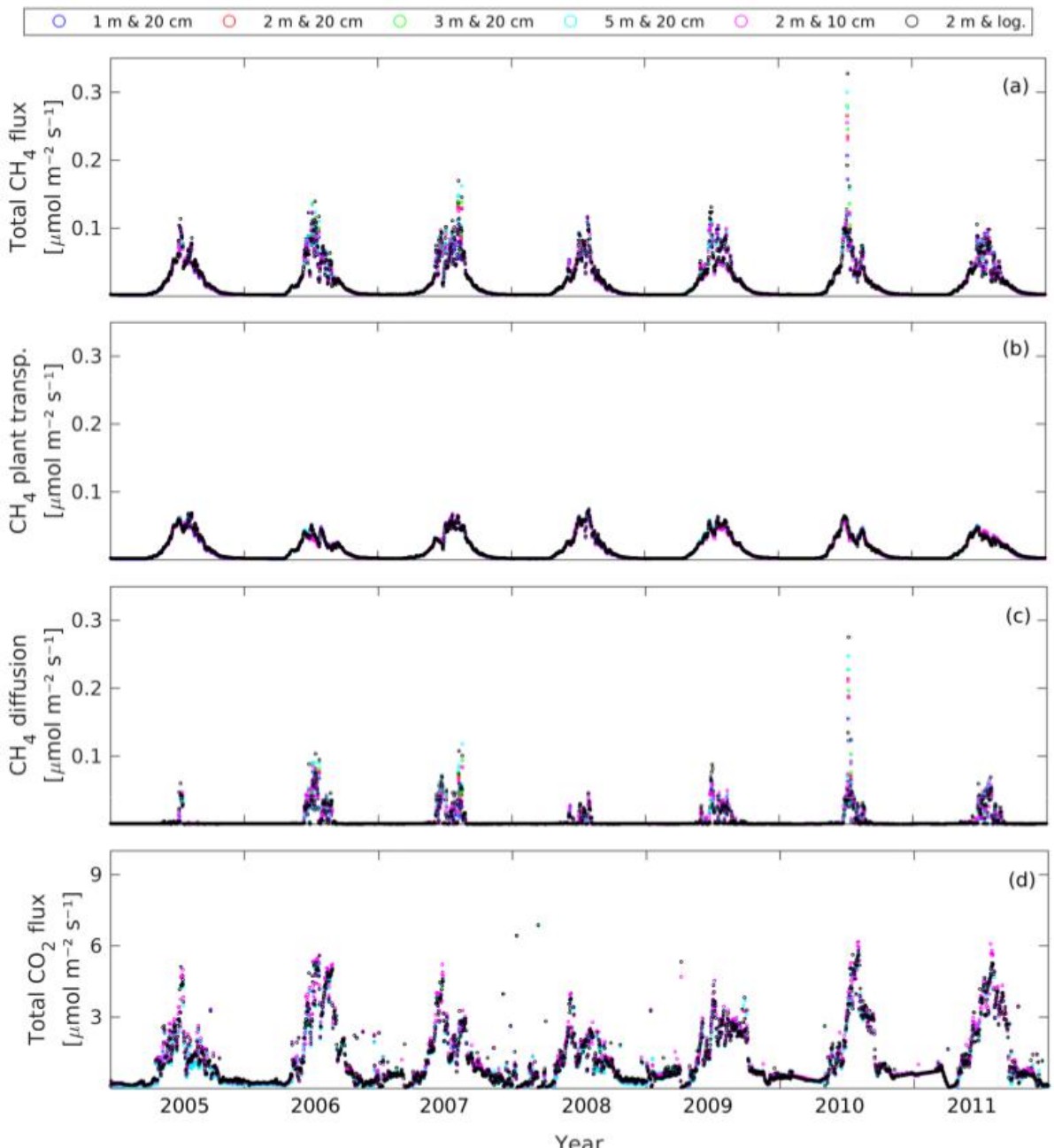

**Figure 13. Time series of CH₄ and CO₂ fluxes simulated for Siikaneva in 2005 to 2011, using different peat depths and layer thicknesses with the same input anoxic respiration rate. (a) Total CH₄ flux, (b) CH₄ plant transport, (c) CH₄ diffusion, (d) total CO₂ flux. Direct ebullition to the atmosphere was negligible and thus not shown. CH₄ ebullited when WTD was below the peat surface was transported to the atmosphere via diffusion in peat or plant roots.**

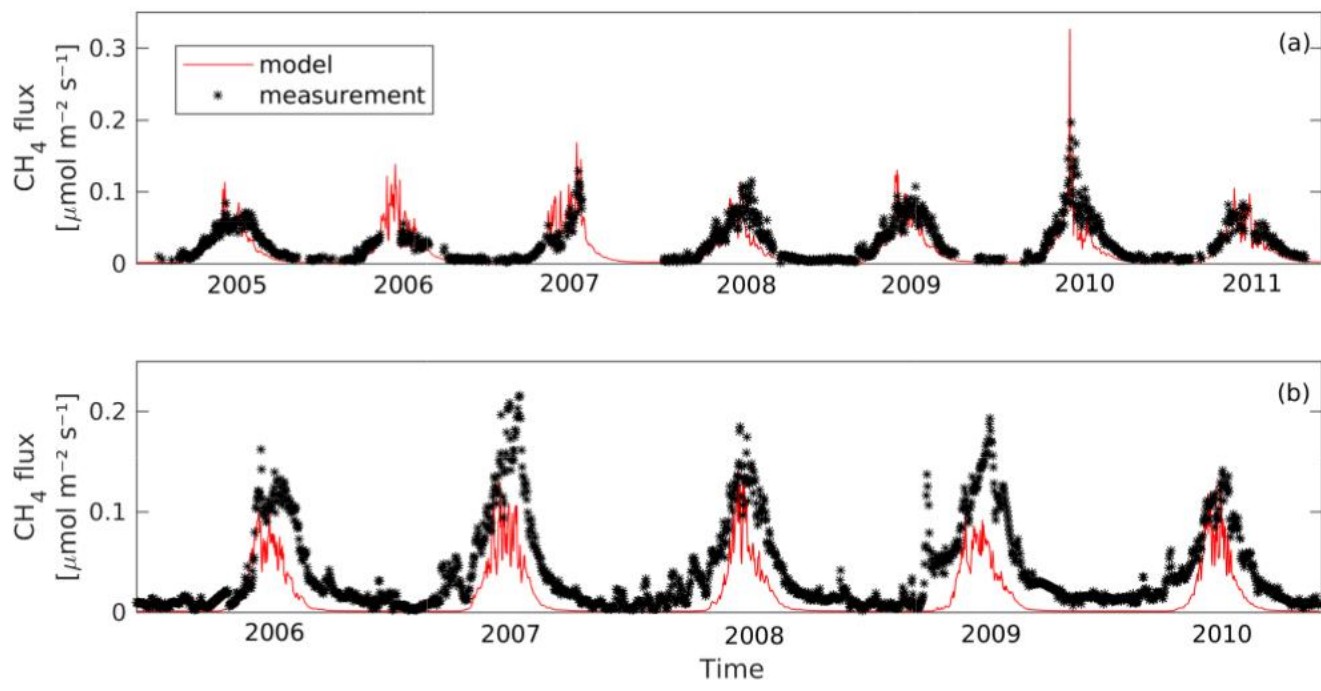

**Figure 14. Comparison of simulated and measured CH₄ emissions (a) at Siikaneva and (b) at Lompolojänkkä. The simulations used the logarithmic layer structure and 2 m of peat.**

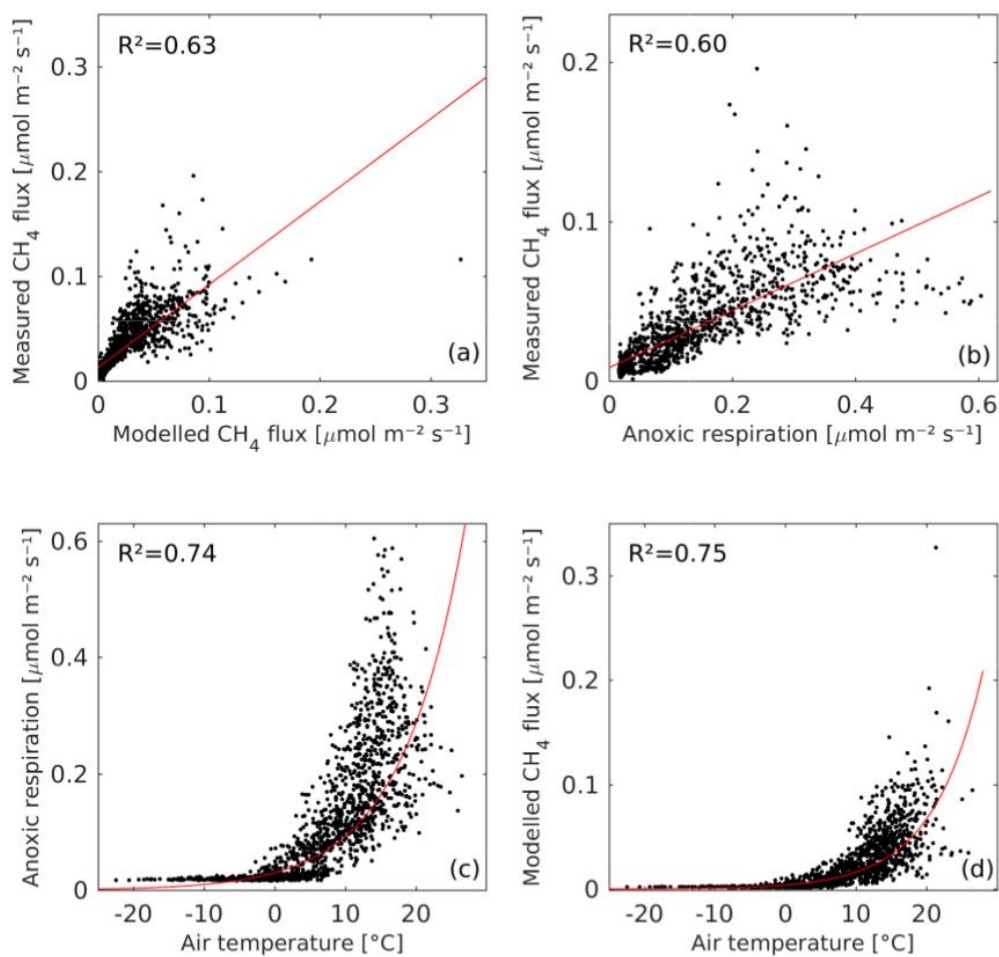

**Figure 15. Correlations between (a) modelled and measured CH₄ flux, (b) input anoxic respiration and measured CH₄ flux, (c) observed air temperature and input anoxic respiration, and (d) observed air temperature and modelled CH₄ flux. The data are from the Siikaneva test (Fig. 14 a).**