# Peer review of "HIMMELI v1.0: HelsinkI Model of MEthane buiLd-up and emIssion for peatlands"

_Geoscientific Model Development, 2017_

## Referee Comment (RC1) · Anonymous Referee #1 · 24 Apr 2017

In this paper, a methane submodel is proposed for use in a larger ecosystem C model. While this is a topic of interest to readers of the journal, this submodel has several key weaknesses that affect its acceptability for publication: (1) It is driven by inputs for anaerobic respiration calculated as a first order function of peat C and root exudation derived from assumed vertical distributions of root mass in the anoxic part of the soil profile (Eq. 6) . While I appreciate that anaerobic respiration is an input rather than an output of this model, it is nonetheless the key driver of CH4 production, as noted in p. 14 and Fig. 11. Anaerobic respiration therefore needs to be explicitly simulated as part of any CH4 model, rather than optimized for site conditions as done here, as it directly determines modelled CH4 emissions. The determination of Pmax. Rref and dWtol (a poorly constrained term) in eqs. B2 and B3 is necessarily site-specific and detracts from model robustness. This optimization overlooks the possibility that anaerobic

respiration can occur in wet soil above the water table. Model testing of anaerobic respiration could have been better constrained by including tests of modelled CO2 fluxes with modelled CH4 fluxes in Fig. 10. (2) It is unclear why total anaerobic respiration does not change with WTD on p. 12 l. 5. Simulating such changes is one of the key challenges in CH4 modelling, but is overlooked in this study. (3) The fixed fraction of respiration that generates CH4 (fm in eq. 7) should in theory be fixed at 0.5, rather than be reset to 0.25 for the field study. This fraction directly affects CH4 generation, but completely overlooks acetotrophic vs hydrogenotropic methanogenesis. (4) There was no clear distinction between gaseous and aqueous diffusive fluxes in eqs. 1 − 3, although they are very different above the water table. I presume these are aqueous fluxes below the water table, but what about gaseous transfer above the water table by which gases are exchanged with the atmosphere? Perhaps this can be easily clarified by the authors. (5) The daily time step of the model eliminates the simulation of diurnal variation in temperature, even though this can be an important driver of that in gas exchange. (6) It is very important to avoid arbitrary parameterizations, such as those associated with the assumed 2 m maximum rooting depth, as these can affect model results in unforeseen ways, and therefore limit the robustness of the model. (7) The only air-water interface that appears to be modelled is that at the surface of the water table, yet such interfaces exist throughout the soil above the water table. Gas exchange across these interfaces can cause localized anaerobic zones in which CH4 can be generated. (8) Are different root porosities considered in eq. 19? These are important in plant adaptation to wetlands, as well as in root gas transfer. (9) CH4 emissions appear to have limited sensitivity to temperature (p. 15), even though a T response of anaerobic respiration was considered in the model. However field studies indicate a large sensitivity of CH4 emission to T, as noted later on p. 16, which is likely important in climate warming studies. Has a key process been overlooked here? (10) Is it realistic that CH4 emissions should increase with WTD (p. 15), or is root-mediated O2 transport overestimated? Is root growth constrained by O2 below the WT? Or is this model result an artefact of assumptions regarding WTD and anaerobic respiration

noted in (2) above?

In the Xu et al. (2016) paper cited in the manuscript, 40 existing CH4 models were reviewed. In many of these models, the issues raised above are explicitly addressed, but some key challenges to further development of these models were raised. The question to be addressed when considering this manuscript for publication is does the model proposed here build upon this earlier work by providing further insight into the key processes by which CH4 emissions are controlled and thereby addressing these challenges? Or is this just another empirical model of CH4 emissions, the parameterization of which is site- and model-specific without reference to earlier modelling work, and therefore of limited interest to the larger modelling community. Unless the authors can provide convincing responses to the points raised above, then I fear the latter.

———————————————————————

---

## Referee Comment (RC2) · Anonymous Referee #2 · 24 Apr 2017

This manuscript presents a sensitivity analysis of a methane module that could be included in peatland models. The authors argue that the novelty of this study is that the model has been developed independent of a full peatland carbon model and can then be tested for sensitivity allowing for dependencies within the methane models itself to be assessed separately from the entire C model. The fact that it is a module without the complete C cycling that feeds input to the methane module, makes it difficult to assess the ability of the model to estimate fluxes as the test that compares it to field-measured fluxes did not optimize the anoxic respiration input and this would actually be generated from the entire peatland C model. Also, sensitivities are difficult to assess this way as important drivers (e.g., temperature driving $CH_4$ production) are not included as this would happen in the other part of the peatland C model that would drive anoxic respiration rates.

[Figure]

Aside from testing the sensitivity of a methane model outside of a full C model, the novelty of the model itself is not clear. The way in which methane production, oxidation and transport is considered in the model appears to be largely developed according to methods used in previous models and therefore it is not clear what improvement is expected here. The way in which ebullition is handled, for example, is quite simplistic and not consistent with literature that clearly illustrates trapping of free-phase gas over time as opposed to release as soon as a bubble is formed (e.g., Comas et al., 2014; Ramirez et al., 2015). I think a clear justification of why another peatland CH4 model is needed must be included to illustrate the utility of this model.

Specific additional comments are given below:

Page 2, Line 5: Maybe the 2nd largest anthropogenic radiative forcing after CO2? Water vapour causes the greatest radiative forcing in the atmosphere, followed by CO2 and then CH4

Page 2, Line 13: Saying no other alternative electron acceptors exist is a bit extreme. Many freshwater wetlands will have cycling of NO3, Fe, SO4, etc., in addition to CH4 production. I suggest rewording this sentence.

Page 4, Line 28: I guess 45-60% is meant (as opposed to . . .). This happens throughout the manuscript in my version.

Page 5, Lines 1-3: And also, peat properties and pore sizes are likely to vary within and between peatlands based on composition of the peat (i.e., sedge vs. wood vs. moss) as well as decomposition status.

Page 5, line 7: the effect of tortuosity on the diffusion coefficient indicates that it is not only the porosity that is important, but the interconnectivity and shape of that porosity and probably the pore size distribution

Page 6, Line 7: In reality WT is the not the divide between water-filled and partially water-filled pore space. Above the WT there is always some fully saturated layer as

the capillary fringe. In practice in the model it doesn't make a difference as the boundary would instead be the capillary fringe, but the way it is written here is technically incorrect.

Page 6, Line 11: When WT is above the surface it can become oxygenated by wind-mixing. Is this considered?

Equation 7: What about inhibition by other electron acceptors? I know you are not following them in the model, but they could be important in some fen systems.

Is CH4 production from the peat matrix accounted for – anaerobic respiration is driven by rooting depth, but CH4 could be produced from other substrates.

Equation 11: Is this really realistic? This would allow a bubble to form, but that doesn't mean that ebullition occurs. Also, once bubbles form, they are often trapped and this affects the concentration gradients and also the ebullition fluxes. A very large bubble release is likely to provide such a high concentration when released that even if the WT is below the surface, not all the CH4 will be oxidized (see page 14, line 15 in the manuscript).

Page 12, Lines 20-25: Was the model parameterized with the data from Siikaneva? If so, how appropriate was the test?

Page 16, lines 26-27: Does this illustrate that evaluating sensitivities in the methane only module, especially when production rates are not appropriately driven by changing conditions, is problematic? Temperature is a very important driving factor for CH4 production, but it is not included in the way the module is constructed making it very difficult to interpret the actual sensitivities of the model.

Page 17, line 10: In this model, the peat column and layering is not important, but what about if the gas is being trapped prior to ebullition or even once mobilized from one layer and then trapped in another (e.g. Comas et al., 2015). We know this happens in reality, but it is not included in this model. If it was how would the results of the study

change?

References: Comas X, Slater L, Reeve AS. 2011. Atmospheric pressure drives changes in the vertical distribution of biogenic free-phase gas in a northern peatland. Journal of Geophysical Research, 116, G04014, doi: 10.1029/2011JG001701.

Comas X, Kettridge N, Binley A, Slater L, Parsekian A, Baird AJ, Strack M, Waddington JM. 2014. Effect of peat structure on the spatial distribution of biogenic gases within bogs. Hydrological Processes, 28, 5483-5494.

Ramirez JA, Baird AJ, Coulthard TJ, Waddington JM. 2015. Ebullition of methane from peatlands: does peat act as a signal shredder? Geophysical Research Letters, 42, 3371-3379.

---

## Referee Comment (RC3) · Anonymous Referee #3 · 9 May 2017

**Overview:**

This paper presents the HelsinkI Model of MEthane buiLd-up and emIssion for peatlands (HIMMELI, v1.0). The model has been developed as a methane module that can be added to or used with peatland carbon cycle models. The model is driven with soil temperature, leaf area index (of aerenchymatous peatland vegetation), water table depth and the anaerobic carbon decomposition rate. It simulates the concentrations and fluxes of $CH_4$, $CO_2$ and oxygen ($O_2$) in a one-dimensional peat column. Sensitivity runs are undertaken and comparison is made with $CH_4$ flux measurements from one site in Finland.

The Global Methane cycle is currently a topic of much interest following the near-zero growth in atmospheric methane concentrations in the early 2000's and its renewed growth since 2007. Various papers have suggested specific sources, including emis-

sions from wetlands, and more recently changes in the atmospheric $CH_4$ sinks, as possible explanations. Wetlands globally are the largest single source of methane, anthropogenic or natural. Boreal wetlands are significant and could become more important still with the faster warming of the Arctic and high northern latitudes.

The Wetland Model Intercomparison (cited paper by Melton et al., 2013) highlighted the current performance of wetland models and the large range of wetland areas and methane fluxes simulated by the participating models. The recent paper by Saunois et al. (2016) on the Global Methane Budget removed some of this model uncertainty by prescribing the wetland extent/area.

I was expecting the present paper to make this connection to the bigger picture. From the material presented, it is very unclear what the intended application of this new model is (local, regional, global??), how it would be used in practice (standalone or coupled) and indeed what advantage it offers over those mentioned in the paper (i.e., Peatland-VU, CLM) and those already in the literature (see cited paper by Xu et al.).

**Specific Comments:**
Model Formulation
The model considers the major $CH_4$ release pathways to the atmosphere (diffusion, plant vascular transport and ebullition) and includes oxidation by $O_2$. $O_2$ is the only electron acceptor considered. What about others?

I am little concerned at the realism of completely oxic layers sitting above the water-saturated anoxic layers. In reality, one might expect a continuous transition, as acknowledged by the authors.

The model runs on a daily timestep. This may be appropriate for large-scale decadal or centennial runs but no justification is given. How was this timestep selected and what are the implications for the modelled methane fluxes?

Many of the model parameters are optimised using the measurements made at a site

in southern Finland (see Table 1, p. 30). These results are included in a second paper (Susiluoto et al., 2017), which is in preparation. This makes it hard to assess their significance, especially in the light of the statement *The uncertainty of some of these parameters is rather high, and a more complete analysis can be found in Susiluoto et al. (2017, in prep.).*

The cited paper by Rinne et al. (2007) shows an exponential dependence of the measured flux on the peat temperature to day 200 (Figure 6 in paper). The lack of a temperature dependence presumably indicates that the temperature dependence is effectively determined by that of the input 'anaerobic carbon decomposition rate'. The temperature-dependence revealed in Fig. 6 is presumably associated with the modelled transport and loss processes.

Many of the key driving variables (soil temperature, leaf area index, water table depth) could be taken either from observations or modelled. It is not clear that this is the case for the anaerobic carbon decomposition rate. If it could be measured, this would improve the utility of HIMMELI.

Comparison versus observations
The model is setup and the modelled $CH_4$ fluxes are compared to eddy covariance flux measurements of $CH_4$ made at Siikaneva, a peatland site in Southern Finland. The intake for the $CH_4$ flux measurements is given as 2.75 m above the peat surface (p. 13). Presumably the surface is fairly homogeneous as no information is given about the footprint nor the prevailing wind direction.

A good fit of the observed and measured fluxes is seen over several annual cycles. This site is effectively used for both model calibration/optimisation and evaluation. This begs the question of how general the derived parameter values are or whether are they specific to this site. There is an obvious need for comparison against measurements from other sites.

Upscaling

It would have been interesting to see upscaled fluxes to the regional/boreal scale and hence an estimate of methane emissions from boreal peatlands.

Code availability

The supplement contains the FORTRAN source code of the model.

**Technical comments:**

The ellipsis (...) is used throughout the paper for 'to', e.g., page 12, line 14: '10...50 cycles' instead of '10 to 50 cycles'

Intercomparison is used in several places when 'comparison' is sufficient (a) Page 1, line 30; (b) Page 12, lines 20 and 22; (c) Page 17, line 18.

**References:**

Saunois et al., 2016: The global methane budget 2000-2012, Earth Syst. Sci. Data, 8, 697-751, doi:10.5194/essd-8-697-2016.

---

## Author Comment (AC1) · 20 Jun 2017

Dear Referee,

Thank you for taking the time to read our manuscript and for giving good and useful comments. They helped us to see shortcomings in this work and taking your comments into account would improve it. Especially the aim of the model and the manuscript in general should indeed be better defined. Below we reply point-by-point to your comments. We hope you find the responses satisfactory and we can prepare and submit a revised version of this manuscript along the lines we suggest below.

A change that we suggest, not directly as a response to any Referee comment, is related to the model parameters. A set of parameter values that we use in the manuscript was taken from an optimization done using Markov chain Monte Carlo methods with observational CH4 flux data from the Siikaneva peatland site, and we refer to Susiluoto et al. (2017, in prep.). The final results and a more exact description of the calibration work are now reported in Susiluoto et al. (2017, GMDD-2017-66). However, there were some major differences between the approaches used here and in the final version of Susiluoto et al., which led to some difference in the values. As the parameter values are not the main point of the present work and as the values here produce a good fit with observations, we suggest that we keep the current values for the revised manuscript. However, in the revised version we will not anymore refer to Susiluoto et al. but add in the Materials and Methods section the description of the optimization.

Referee comments are typed in italics. They are followed by our responses and suggestions of how we would revise the manuscript, as plain text and numbered referring to the comment number.

*In this paper, a methane submodel is proposed for use in a larger ecosystem C model. While this is a topic of interest to readers of the journal, this submodel has several key weaknesses that affect its acceptability for publication:*

*(1) It is driven by inputs for anaerobic respiration calculated as a first order function of peat C and root exudation derived from assumed vertical distributions of root mass in the anoxic part of the soil profile (Eq. 6). While I appreciate that anaerobic respiration is an input rather than an output of this model, it is nonetheless the key driver of CH4 production, as noted in p. 14 and Fig. 11. Anaerobic respiration therefore needs to be explicitly simulated as part of any CH4 model, rather than optimized for site conditions as done here, as it directly determines modelled CH4 emissions. The determination of Pmax. Rref and dWtol (a poorly constrained term) in eqs. B2 and B3 is necessarily site-specific and detracts from model robustness. This optimization overlooks the possibility that anaerobic respiration can occur in wet soil above the water table. Model testing of anaerobic respiration could have been better constrained by including tests of modelled CO2 fluxes with modelled CH4 fluxes in Fig. 10.*

(1) Response:

We agree that simulation of anaerobic respiration is a significant, perhaps the most significant, component of a complete model of peatland CH4 emissions. However, the target of our work was to produce a module that simulates only the transport and oxidation of CH4. The reason for this is that there are soil carbon models that simulate anoxic respiration and the interface to a CH4 module would be through the respiration. Two examples of this kind of model environments are land surface models JSBACH and JULES. Therefore, we think that a CH4 module that is driven with anoxic respiration is a justifiable modelling unit and in order to ensure its functionality for further use it is reasonable to analyze its sensitivity and performance independently.

The purpose of presenting the (already published) NPP model in the Appendix B is not to claim it is a general photosynthesis model for all the peatlands but, by contrast, to show that we

created as realistic NPP as possible for the model testing. The Siikaneva test was done to demonstrate that combined with realistic input, HIMMELI does output realistic $CH_4$ fluxes, which is not so evident if looking only at the mechanistic sensitivity tests. It is true that the parameter values for the NPP model are mainly from a study done at an oligotrophic fen, like Siikaneva. However, we think that in this model-data comparison it is not a downside to use a carbon input that corresponds to reality as closely as possible.

We also agree with the Referee that the choice of using water table depth (WTD) as a strict divider of the peat to oxic and anoxic parts is a simplification and as mentioned in Section 2. 'Key factors for $CH_4$ transport and oxidation', water-filled, anoxic sites can occur above it. In our opinion, however, it is uncertain to what extent the model estimate would be improved e.g., by assuming a certain volume of anoxic microsites in the peat above the WTD, which in practice would mean adding new unknown parameters. In any case, most of the peatlands have microtopography, hollows and hummocks, and even the observation-based site-level WTD is only an approximate value for the peatland, not to speak of a modelled WTD. On these grounds, we think this strict division to anoxic and oxic parts is, although being simple, a robust enough approach to be used in land-surface models.

Including modelled CO2 fluxes into Fig. 10 is indeed a good suggestion.

(1) Suggested changes to the manuscript:

In Introduction, we will clarify the aim of HIMMELI and explain more clearly why simulation of anoxic respiration is not included in the model.

We will clarify the role of the Siikaneva test in the paper and, as suggested by another Referee, we will add a comparison of the model with data from another peatland site. This will be a test of how well the current parameterization fits to other peatland sites.

We will add discussion about how realistic is the strict division to oxic/anoxic parts of peat, on page 7, Section 3.1.2.

We will add CO2 fluxes in Figure 10.

*(2) It is unclear why total anaerobic respiration does not change with WTD on p. 12 l. 5. Simulating such changes is one of the key challenges in CH4 modelling, but is overlooked in this study.*

(2) Response:

In this part of the work, we tested the sensitivity of the simulated CH4 emissions to input. Anoxic respiration is taken by HIMMELI as input, in mol $s^{-1}$ per $m^2$ of ground surface area, i.e., per the simulated peat column. HIMMELI itself cannot change the total anoxic respiration rate as it is the input, but what it does is that it distributes the given input respiration to the inundated layers of the peat column along the root distribution. Number of those layers depends on WTD and so the anoxic respiration per cubic meter changes with WTD.

As we say above, we agree that simulating anoxic respiration is highly important in CH4 modelling, however, the case here is that another model (e.g. a soil carbon model of a land surface scheme) has already taken care of it. Most probably the total anoxic respiration rate provided by this other model depends on WTD, but we did not want to set any dependency here, in the mechanistic sensitivity tests, since it would have meant in practice that the test results are valid only when the dependency is as we described it. In this way we keep the tests as more generic and avoid inherent mixing with a soil carbon model.

The idea in our mechanistic tests was to analyze how much and via what pathways the other driving variables (WTD, temperature, LAI) affect the output CH4 emission rate when the carbon input rate is constant. Given that the anoxic respiration rate largely governs the CH4 emissions, it is important to standardize it and find out what kind of dependencies there are

inside the CH4 model alone. As far as we know, this has not been thoroughly analyzed earlier without the mask of changing non-CH4 carbon processes.

(2) Suggested changes to the manuscript:

We will add text that clarifies this issue on p. 12.

*(3) The fixed fraction of respiration that generates CH4 (fm in eq. 7) should in theory be fixed at 0.5, rather than be reset to 0.25 for the field study. This fraction directly affects CH4 generation, but completely overlooks acetotrophic vs hydrogenotropic methanogenesis.*

(3) Response:

This is one of the parameters that has high uncertainty and indeed affects CH4 generation directly. It would be great to simulate the different methane production pathways and microbial groups and this way perhaps enable tuning the model to e.g. different peatland types, but we have not done it so far. CH4 production is only modelled via this one bulk parameter.

Nilsson and Öquist (2009) state in their article that theoretically, the CH4/CO2 quotient from terminal mineralization of soil organic matter in optimal methanogenic conditions ranges from 0 to 0.7, being ~0.5 when carbohydrates are mineralized. Their literature review showed, however, dominance of CO2: the observed CO2/CH4 quotient in anoxic incubations had varied from 0.5 to 36,000 with median value in a filtered data set being 6-7. Also models have used ratios other than 50/50, e.g. the CH4 model by Wania et al. (2010) used CH4/CO2 ratio of 0.1. On this basis, the value 0.25 used in the model calibration is within a realistic range.

We can run the Siikaneva simulations again, with fm of 0.5. However, because the model calibration used 0.25, changing to 0.5 will most probably rise the CH4 emission level higher than the observations, if we keep the other parameter values, in particular the fraction of NPP allocated to root exudates, the same as now. A compromise that we suggest is to present results from both runs in a supplement, which would also illustrate the effect of changing this parameter.

(3) Suggested changes to the manuscript:

We will discuss this parameter in light of the article by Nilsson & Öquist (2009), in the end of Section 3.2.1. We will rerun the Siikaneva simulation done with the logarithmic layer structure using fm of 0.5, and the result (compared with the original run) will be added as a supplement.

*(4) There was no clear distinction between gaseous and aqueous diffusive fluxes in eqs. 1 – 3, although they are very different above the water table. I presume these are aqueous fluxes below the water table, but what about gaseous transfer above the water table by which gases are exchanged with the atmosphere? Perhaps this can be easily clarified by the authors.*

(4) Response:
Yes, diffusion happens in water below the water table and in the air above it and the model calculates it accordingly. This, including the description of how the flux is calculated at the water-air interface is explained into more detail in Section 3.1.8. "Diffusion in the peat". We are sorry that this is left unclear in the Section 3.1.1 and around the equations 1-3.

(4) Suggested changes to the manuscript:
We will clarify the text in Section 3.1.1 after the equations 1-3 by explicitly mentioning that the diffusive fluxes in the peat are in water below WTD and in air above it.

*(5) The daily time step of the model eliminates the simulation of diurnal variation in temperature, even though this can be an important driver of that in gas exchange.*

(5) Response:
We agree that in this work, which specifically aims at testing the transport model, it would be reasonable to test how the model works if it is run on a shorter time step. The reason for running it on daily time step was that the main plan for HIMMELI is to use it with models that provide daily input and so the present results were needed for that. We can test running the model on a shorter time step.

(5) Suggested changes to the manuscript:
We will test running HIMMELI with realistic input data at frequency shorter than one day, with diurnal variation of soil temperature. Results of this model run will be compared with simulation done on daily timestep, in which input data are daily averages of the previous test. The outcome will be added to Results.

*(6) It is very important to avoid arbitrary parameterizations, such as those associated with the assumed 2 m maximum rooting depth, as these can affect model results in unforeseen ways, and therefore limit the robustness of the model.*

(6) Response:
Maximum rooting depth is, in fact, not fixed to 2 m but it depends on peat depth. If peat depth is less than 2 m, the model uses the peat depth as the maximum rooting depth. In case peat depth is more than 2 m, the model is also prepared to handle the situation, but rooting depth is then set to 2 m according to literature (described in Section 3.1.3). Maximum rooting depth could be changed to a parameter whose value could be determined by the model user but for the current model version, we will not change it.

(6) Suggested changes to the manuscript:
-

*(7) The only air-water interface that appears to be modelled is that at the surface of the water table, yet such interfaces exist throughout the soil above the water table. Gas exchange across these interfaces can cause localized anaerobic zones in which CH4 can be generated.*

(7) Response:
We agree, anaerobic zones can exist above the WTD. However, simulating them would mean increasing the number of uncertain parameters in the model, and simulation of this process would be demanding since, for instance, we do not have corresponding experimental data. As mentioned above, we consider the current approach is robust enough to be used in land surface models.

(7) Suggested changes to the manuscript:
As for the comment (1) above, we will add discussion about how realistic the strict division to oxic/anoxic parts of peat is, on page 7, Section 3.1.2.

*(8) Are different root porosities considered in eq. 19? These are important in plant adaptation to wetlands, as well as in root gas transfer.*

(8) Response:
In the current model version they are not considered. A single porosity is assumed for all the gas-transporting vegetation. They could be implemented in further model versions but similarly

to the previous comment response (7) adding them would mean increasing the number of uncertain parameters in the model.

(8) Suggested changes to the manuscript:
-

*(9) CH4 emissions appear to have limited sensitivity to temperature (p. 15), even though a T response of anaerobic respiration was considered in the model. However field studies indicate a large sensitivity of CH4 emission to T, as noted later on p. 16, which is likely important in climate warming studies. Has a key process been overlooked here?*

(9) Response:

We regret for having described the test set-up unclearly. In the sensitivity tests described in Sects. 3.2.1 and 4.1, the anoxic respiration was a constant input value that was independent of temperature. This constant respiration was given to HIMMELI that was then driven with different temperatures. Thus, temperature affected the output only by affecting the processes that are included in HIMMELI, that is, inhibition of CH4 production, aerobic respiration, CH4 oxidation, and the transport processes. Therefore, the result of the test tells about the sensitivity of these other processes to temperature.

When running HIMMELI for the Siikaneva site, we used non-constant simulated anoxic respiration, part of which was produced using the temperature dependent Q10 model. In this case both anoxic respiration and output CH4 emission correlated with temperature, however, this was at least principally due to the input correlating with temperature, which is always the case when HIMMELI is used together with an independent carbon cycle model.

According to literature (e.g. Nilsson & Öquist, 2009), CH4 production is more sensitive to temperature than CO2 production, or conversely, the CO2/CH4 quotient decreases when temperature increases. In a way this has now been taken into account to some extent since increasing temperature reduces O2 solubility and thus the dissolved O2 concentration available for inhibiting CH4 production and enhancing CH4 oxidation is reduced.

(9) Suggested changes to the manuscript:
We will clarify the role of the input respiration in different tests, in the section about the tests (3.2) and in the Results and discussion section. We will add a figure that shows the correlation of soil temperature with anoxic respiration used as input in the Siikaneva simulations, as well as correlation between temperature and modelled CH4 emissions.

*(10) Is it realistic that CH4 emissions should increase with WTD (p. 15), or is root-mediated O2 transport overestimated? Is root growth constrained by O2 below the WT? Or is this model result an artefact of assumptions regarding WTD and anaerobic respiration noted in (2) above?*

(10) Response:
The simple answer to this question is that we do not know how realistic the root oxygen transport is. Here we indeed again face the fact that in our tests, input respiration was not dependent on WTD. In reality, and with a model that simulates anoxic respiration dependent on WTD, probably the anoxic respiration rate per peatland surface area would decrease with decreasing WTD and therefore also the CH4 emissions would decrease, despite of the decreasing root transport of O2 into the inundated soil. But now when WTD had no impact on the anoxic respiration rate, the result was reverse, which is an interesting result as such and not discussed earlier since non-CH4 carbon processes are masking the dynamics of CH4 processes

There are a few previous CH4 models that also simulate the O2 transport to the peat and the consequent O2 concentrations affect different processes in the soil (e.g. Wania et al., 2010, Riley et al., 2011). However, since the observational data on O2 transport is scarce or nonexistent, it is by definition not possible to validate these results. Our analysis shows that it should be done.

Root mass is vertically distributed according to the exponential function (Eq. 4). This means that the root transport capacity is small in the bottom peat layers.

(10) Suggested changes to the manuscript:
We will write a more thorough explanation of how anoxic respiration depended on WTD in the first paragraph of Section 4.1, 'Model sensitivity to input data'.

*(11) In the Xu et al. (2016) paper cited in the manuscript, 40 existing CH4 models were reviewed. In many of these models, the issues raised above are explicitly addressed, but some key challenges to further development of these models were raised. The question to be addressed when considering this manuscript for publication is does the model proposed here build upon this earlier work by providing further insight into the key processes by which CH4 emissions are controlled and thereby addressing these challenges? Or is this just another empirical model of CH4 emissions, the parameterization of which is site- and model-specific without reference to earlier modelling work, and therefore of limited interest to the larger modelling community. Unless the authors can provide convincing responses to the points raised above, then I fear the latter.*

(11) Response:

We acknowledge the fact that HIMMELI does not bring any new processes as such into the CH4 model world and the process descriptions largely are from earlier models, which we explicitly mention on p3, lines 11-12. HIMMELI was developed in order to have a CH4 module that could be plugged into different peatland carbon models and that simulates transport of all CH4, O2 and CO2. The parameterization in the current manuscript is based on only one peatland site, however, the aim has been to have physically sound parameter values. When moving to other peatlands and especially if using HIMMELI in large-scale methane modeling, the model needs to be re-calibrated.

We agree that we explained very vaguely how HIMMELI relates to the existing methane models. Xu et al. (2016) listed 40 terrestrial ecosystem models for $CH_4$ cycling. However, when considering only their CH4 emission parts, this number seems to be slightly reduced. For instance, Ringeval et al. (2011) say that they included the Walter et al. CH4 model in ORCHIDEE and Spahni et al. (2011) that they applied LPJ-WhyMe in LPI-Bern for biogeochemical modelling of CH4 emissions.

Although HIMMELI does not include all processes that already exist in some models (e.g. alternative $e^-$ acceptors, anaerobic CH4 oxidation), it is among the most complete models considering the transport of compounds. According to Xu et al., there are only 5 models that simulate all vertically resolved biogeochemistry, O2 availability to CH4 oxidation, and three pathways of CH4 transport. Of these, the Xu model (Xu et al. 2007), CLM-Microbe (Xu et al. 2014) and VISIT (Ito & Inatomi, 2012) do not explicitly simulate O2 transport between the atmosphere and peat. On the other hand, LPJ-WhyMe (Wania et al. 2010), a revised multi-substance version of TEM (Tang et al. 2010) and a recent model by Kaiser et al. (2017) - that were not included in the list by Xu et al. -- do simulate all these. HIMMELI also simulates $CO_2$ transport via all three transport pathways. This is not a common feature in CH4 models: to our knowledge, only the multi-substance version of TEM (Tang et al. 2010) and the Segers model (Segers & Leffelaar, 2001) included it.

Xu et al. (2016) raised some needs and key challenges to further development of the CH4 models. Some of them are not relevant for HIMMELI as they concern complete peatland ecosystem models -- however, we admit that HIMMELI does not address all of those that were relevant. We can, however, point out two issues. Firstly, Xu et al. emphasized that the models should consider the vertical distribution of processes, which is something that HIMMELI does. Secondly, Xu et al. stated that well-validated CH4 modules should be included in Earth system modeling frameworks. Although not mentioned in the manuscript, the main goal of HIMMELI is to use it as a module in large-scale land surface models (JSBACH, JULES) that are part of ESMs.

When it comes to the modeling community's interest in the manuscript, we think that especially because this model does include components similar to earlier CH4 models, the results of the sensitivity analysis should be interesting. Xu et al. (2016) wrote: "Furthermore, evidence demonstrating that incorporating all of these processes would lead to more accurate prediction is needed". We think our paper is a statement in this type of discussion since it indicates that A) although vertically resolved transport and oxidation processes have significance, the CH4 emissions simulated by this type of models are largely determined by the CH4 production rate and B) adding complexity like e.g. transport of oxygen and effect of O2 concentration on the process rates can have a high impact on the output. However, because of general lack of data, it remains unclear if anyone has validated the realism of the oxygen processes.

(11) Suggested changes to the manuscript:

In the Introduction, we will clarify the aim of HIMMELI and describe how this model relates to earlier methane models by adding the above text that refers to the review by Xu et al. (2016).

REFERENCES

Ito and Inatomi: Use of process-based model for assessing the methane budget of global terrestrial ecosystems and evaluation of uncertainty, Biogeosciences 9, 759-773, doi:10.5194/bg-9-759-2012, 2012.

Nilsson and Öquist: Partitioning litter mass loss into carbon dioxide and methane in peatland ecosystems, Geoph. Monog. Series, 184, Carbon Cycling in Northern Peatlands, 131-144, 2009.

Riley, Subin, Lawrence, Swenson, Torn, Meng, Mahowald, and Hess: Barriers to predicting changes in global terrestrial methane fluxes: analyses using CLM4Me, a methane biogeochemistry model integrated in CESM, Biogeosciences, 8, 1925-1953, 2011.

Ringeval, Friedlingstein, Koven, Ciais, de Noblet-Ducoudré, Decharme, and Cadule: Climate-CH$_4$ feedback from wetlands and its interaction with the climate-CO$_2$ feedback, Biogeosciences, 8, 2137-2157, 2011.

Segers and Leffelaar: Modeling methane fluxes in wetlands with gas-transporting plants 1-3, J. Geophys. Res. 106, 2001.

Spahni, Wania, Neef, van Weele, Pison, Bousquet, Frankenberg, Foster, Joos, Prentice, and van Velthoven: Constraining global methane emissions and uptake by ecosystems. Biogeosciences 8, 1643-1665, doi: 10.5194/bg-81643-2011, 2011.

Tang, Zhuang, Shannon, and White: Quantifying wetland methane emissions with process-based models of different complexities, Biogeosciences, 7, 3817-3837, 2010.

Wania, Ross, and Prentice.: Implementation and evaluation of a new methane model within a dynamic global vegetation model: LPJ-WHyMe v.1.3.1, Geosci. Model Dev., 3, 565-584, 2010.

Xu, Jaffe, and Mauzerall: A process-based model for methane emission from flooded rice paddy systems. Ecol Model 205, 475-491. 2007.

Xu, Schimel, Thornton, Song, Yan, and Goswami: Substrate and environmental controls on microbial assimilation of soil organic carbon: a framework for Earth system models, Ecol. Lett., 17, 547-555. 2014.

Xu, Yuan, Hanson, Wullschleger, Thornton, Riley, Song, Graham, Song, and Tian: Reviews and syntheses: Four decades of modeling methane cycling in terrestrial ecosystems, Biogeosciences, 13, 3735–3755, 2016.

---

## Author Comment (AC2) · 20 Jun 2017

Dear Referee,

Thank you for the good and useful comments regarding the manuscript. They helped us to see shortcomings in the work and taking your comments into account would improve the paper. Especially the aim of the model and this work in general should indeed be better described. Below we reply point-by-point to your comments. We hope you find our responses satisfactory and we can prepare and submit a revised version of this manuscript along the lines we suggest below.

A change that we suggest, not directly as a response to any Referee comment, is related to the model parameters. A set of parameter values that we use in the manuscript was taken from an optimization done using Markov chain Monte Carlo methods with observational $CH_4$ flux data from the Siikaneva peatland site, and we refer to Susiluoto et al. (2017, in prep.). The final results and a more exact description of the calibration work are now reported in Susiluoto et al. (2017, GMDD-2017-66). However, there were some major differences between the approaches used here and in the final version of Susiluoto et al., which led to some difference in the values. As the parameter values are not the main point of the present work and as they produce a good fit with observations, we suggest that we keep the current values for the revised manuscript. However, in the revised version we will not anymore refer to Susiluoto et al. but add in the Materials and Methods section a description of the optimization.

Referee comments are typed in italics. They are followed by our responses and suggestions of how we would revise the manuscript, as plain text and numbered referring to the comment number.

*(1) This manuscript presents a sensitivity analysis of a methane module that could be included in peatland models. The authors argue that the novelty of this study is that the model has been developed independent of a full peatland carbon model and can then be tested for sensitivity allowing for dependencies within the methane models itself to be assessed separately from the entire C model. The fact that it is a module without the complete C cycling that feeds input to the methane module, makes it difficult to assess the ability of the model to estimate fluxes as the test that compares it to field-measured fluxes did not optimize the anoxic respiration input and this would actually be generated from the entire peatland C model. Also, sensitivities are difficult to assess this way as important drivers (e.g., temperature driving CH4 production) are not included as this would happen in the other part of the peatland C model that would drive anoxic respiration rates.*

(1) Response:

We agree with the Referee that it is difficult to evaluate the model's ability to predict $CH_4$ fluxes and sensitivities to input when it does not include the whole carbon cycle. The reason for developing this kind of methane model was to produce a module that can be used in different purposes, as a platform for specific studies on methane processes, but principally as a component of large-scale biosphere models that provide the anoxic respiration input. We believe it can be useful for the community. For instance, the $CH_4$ model of Walter & Heimann (1996) has been utilized in several peatland ecosystem modelling frameworks to simulate methane (e.g. in Ringeval et al. 2011, van Huissteden et al. 2009).

We considered the test with Siikaneva data (Sections 3.2.3, 4.2 and 4.3) as a test of whether HIMMELI produces realistic output, as we aimed at using as realistic input respiration as possible in this test. On the other hand, as written in the manuscript, we especially think that as the anoxic respiration rate largely governs the $CH_4$ emissions, it is important to standardize it and find out what kind of dependencies there are inside the $CH_4$ model alone, given that it usually takes a relatively large portion of a complete peatland carbon model.

(1) Suggested changes to the manuscript:

In the Introduction, we will clarify the aim of HIMMELI and explain more clearly why simulation of anoxic respiration is not included in the model.

We will clarify the role of the Siikaneva test in the paper and, as suggested by another Referee, we will add a comparison of the model with data from another peatland site. This will be a test of how well the current parameterisation fits to other peatland sites.

*(2) Aside from testing the sensitivity of a methane model outside of a full C model, the novelty of the model itself is not clear. The way in which methane production, oxidation and transport is considered in the model appears to be largely developed according to methods used in previous models and therefore it is not clear what improvement is expected here. The way in which ebullition is handled, for example, is quite simplistic and not consistent with literature that clearly illustrates trapping of free-phase gas over time as opposed to release as soon as a bubble is formed (e.g., Comas et al., 2014; Ramirez et al., 2015). I think a clear justification of why another peatland CH4 model is needed must be included to illustrate the utility of this model.*

(2) Response:

About ebullition: please see our responses to your comments number (11) and (14).

We acknowledge the fact that HIMMELI does not bring any new processes into $CH_4$ modelling and the process descriptions are based on earlier models. This is mentioned on P3, lines 11-12. We wanted to produce a model that simulates the transport of all $CH_4$, $O_2$ and $CO_2$ that is not a common feature among $CH_4$ models. We decided that rather than taking directly one of the existing model codes that are developed with and thus closely connected to some biosphere model, we would systematically start from fundamental elements and combine the process descriptions in a format that can be flexibly applied for different uses as, for instance, the peat column structure is not fixed.

Although HIMMELI does not include all processes that already exist in some models (e.g. alternative e⁻ acceptors, anaerobic CH4 oxidation), it is among the most complete models considering the transport of compounds. According to Xu et al. (2016), who reviewed 40 existing terrestrial ecosystem models for $CH_4$ cycling, there are only 5 models that simulate all these: vertically resolved biogeochemistry, $O_2$ availability to $CH_4$ oxidation, and three pathways of $CH_4$ transport. Of these 5, the Xu model (Xu et al. 2007), CLM-Microbe (Xu et al. 2014) and VISIT (Ito & Inatomi, 2012) do not explicitly simulate $O_2$ transport between the atmosphere and peat. On the other hand, LPJ-WhyMe (Wania et al. 2010), a revised multi-substance version of TEM (Tang et al. 2010) and a recent model by Kaiser et al. (2017) - that were not included in the list by Xu et al. -- do simulate all these. HIMMELI also simulates $CO_2$ transport via all three transport pathways. To our knowledge, only the multi-substance version of TEM (Tang et al. 2010) and the Segers model (Segers and Leffelaar, 2001) included it.

We think that as the anoxic respiration rate largely governs the $CH_4$ emissions, it is important to standardize it and find out what kind of dependencies there are inside the $CH_4$ model alone, given that it takes a relatively large portion of a complete peatland carbon model. Here the fact that HIMMELI contains similar components as other methane transport models means that the results reveal and clarify inherent assumptions and process dynamics in the other models.

(2) Suggested changes to the manuscript:

We will justify the necessity of the new model by adding the contents of the above text (reference to Xu et al. 2016) in the Introduction.

*(3) Page 2, Line 5: Maybe the 2nd largest anthropogenic radiative forcing after CO2? Water vapour causes the greatest radiative forcing in the atmosphere, followed by CO2 and then CH4*

(3) Response:
Agreed.

(3) Suggested changes to the manuscript:
We will correct this sentence: "…inducing the second largest radiative forcing among well-mixed greenhouse gases."

*(4) Page 2, Line 13: Saying no other alternative electron acceptors exist is a bit extreme. Many freshwater wetlands will have cycling of NO3, Fe, SO4, etc., in addition to CH4 production. I suggest rewording this sentence.*

(4) Response:
Yes, we agree.

(4) Suggested changes to the manuscript:
We will remove this sentence.

*(5) Page 4, Line 28: I guess 45-60% is meant (as opposed to : : :). This happens throughout the manuscript in my version.*

(5) Response:
The manuscript preparation guidelines of this journal say: "A range of numbers should be specified as "a to b" or "a…b". The expression "a-b" is only acceptable in cases when no confusion with "a minus b" is possible." We thought it would be clearest to use the same convention consistently throughout the manuscript and therefore the expression "a…b" everywhere but we can change this.

(5) Suggested changes to the manuscript:
We leave the "a…b" expression only in tables but within the text change it to "a to b".

*(6) Page 5, Lines 1-3: And also, peat properties and pore sizes are likely to vary within and between peatlands based on composition of the peat (i.e., sedge vs. wood vs. moss) as well as decomposition status.*

(6) Response:
Yes, true, that is relevant information here. Thank you for pointing out these.

(6) Suggested changes to the manuscript:
We will add this information on Page 5.

*(7) Page 5, line 7: the effect of tortuosity on the diffusion coefficient indicates that it is not only the porosity that is important, but the interconnectivity and shape of that porosity and probably the pore size distribution*

(7) Response:
Yes, this is a good point. We are sorry for the inadequate piece of text.

(7) Suggested changes to the manuscript:
We will modify this text so that it also describes the significance of tortuosity.

*(8) Page 6, Line 7: In reality WT is the not the divide between water-filled and partially water-filled pore space. Above the WT there is always some fully saturated layer as the capillary fringe. In practice in the model it doesn't make a difference as the boundary would instead be the capillary fringe, but the way it is written here is technically incorrect.*

(8) Response:
We agree, this is an incorrect statement as it is, this should refer only to the model.

(8) Suggested changes to the manuscript:
We will correct this sentence to: "In the model, WTD is taken as a strict divider of the peat into water-filled and air-filled parts."

*(9) Page 6, Line 11: When WT is above the surface it can become oxygenated by windmixing. Is this considered?*

(9) Response:

Yes, windmixing can affect the $O_2$ concentrations but this is not considered in the model yet. The model naturally is a rough simplification of reality: so far it assumes a pure water layer on top of the peat surface, although there often is vegetation growing in the peat. Vegetation would hinder the windmixing via affecting wind speed and generally modifying the physical conditions affecting thin boundary layers regulating gas transfer across the water-air interface. These processes are not fully understood even for open water surfaces of inland water bodies (we are also working with these issues) and in our opinion the inclusion of partly unknown processes is out of the scope of the present manuscript.

(9) Suggested changes to the manuscript:

We will discuss this point in the Section 3.1.2 in which the possible water layer on top of the peat surface is mentioned.

*(10) Equation 7: What about inhibition by other electron acceptors? I know you are not following them in the model, but they could be important in some fen systems. Is CH4 production from the peat matrix accounted for – anaerobic respiration is driven by rooting depth, but CH4 could be produced from other substrates.*

(10) Response:

We think that other electron acceptors are an important issue. We did not include them in the model because we thought their concentrations depend on site characteristics, such as the water source, and it would be difficult to estimate them. Therefore, these estimates would necessarily not improve the accuracy of the model. However, given that our results (and also earlier works) indicate that methane production rate largely drives the simulated emissions and the oxygen inhibition thus plays a significant role, including other e- acceptors could possibly be a way to take into account site differences, for instance, bog vs. fen. This could be done in future model versions.

In the current model version, anoxic respiration is one bulk input stream and HIMMELI does not take a stand on what organic compounds are decomposed, whether they are root exudates or other substrates. For simplicity, everything is distributed with the root mass except in the case that peat depth exceeds 2 m when some respiration also is allocated in the rootless peat layers. This choice (as opposed to distributing the input e.g. evenly across the peat column) was motivated by the fact that recently fixed carbon seems to be the main source of methane. For instance, according to Oikawa et al. (2017), less than 5% of $CO_2$ and $CH_4$ emissions originate from soils below 50 cm in flooded peatlands. However, in case that

HIMMELI is used in a context where it is essential to simulate the different carbon sources, it is not a big task to modify the code so that this becomes possible.

(10) Suggested changes to the manuscript:

We will add text/discussion about the possible other electron acceptors and distribution of input carbon in the Section 3.1.3 on $CH_4$ production.

*(11) Equation 11: Is this really realistic? This would allow a bubble to form, but that doesn't mean that ebullition occurs. Also, once bubbles form, they are often trapped and this affects the concentration gradients and also the ebullition fluxes. A very large bubble release is likely to provide such a high concentration when released that even if the WT is below the surface, not all the CH4 will be oxidized (see page 14, line 15 in the manuscript).*

(11) Response:

We agree with the Referee that after a bubble has been formed there are still several processes that take place before the bubble reaches the surface and contributes to the $CH_4$ flux to the atmosphere. For instance, the bubbles still need to traverse through the peat column up to the atmosphere. Also, like the Referee mentions, during the time that the bubbles travel upwards towards the atmosphere they constantly interact with the surrounding pore water and hence alter e.g. the $CH_4$ concentration gradients.

Such processes are still missing from most of the peatland $CH_4$ models (Xu et al., 2016), including HIMMELI. This is most likely because relatively little is known about bubble movement in peat and how to describe it accurately in models, although there are some attempts to model this process (Ramirez et al., 2015). In general, bubble movement in porous media is a highly complex problem, which depends on the fine-scale structure of the media in which the bubbles are moving in. How to incorporate such complex phenomenon in a simple, yet accurate way in peatland $CH_4$ cycling models is still unsolved. However, as it happens, we are at the moment preparing a manuscript in which we are comparing different ebullition modelling approaches and one of them incorporated a simple scheme to take into account the bubble movement. Nevertheless, as mentioned this is a topic for the other study.

Considering the reviewer's comment about the page 14, line 15: in this manuscript direct ebullition to the surface takes place only when WTD is above the surface. If WTD is below surface, then the $CH_4$ in the bubbles is released to the lowest air layer from where it is transported via diffusion in the air-peat column to the atmosphere. Hence even large bubbles are first released to the bottom air layer below the peat surface, before reaching the atmosphere. We argue that this is how it happens also in reality and hence is the correct way to describe this process in a model.

(11) Suggested changes to the manuscript

We will discuss the points mentioned above (how the bubble movement would happen in reality, compared with the model) in Section 3.1.7. In addition, we will clarify the sentence on page 14. It now is: "…the direct ebullition fluxes to the atmosphere were zero when WTD was below the peat surface" but we will rephrase it: "ebullition to the atmosphere occurred only when WTD was at or above the peat surface". We hope this slightly modified version is clearer.

*(12) Page 12, Lines 20-25: Was the model parameterized with the data from Siikaneva? If so, how appropriate was the test?*

(12) Response:

Yes, the parameter set used in this study was a combination of literature values and values set by calibrating HIMMELI with Siikaneva data. In this sense the test was not appropriate for evaluating the model fit. However, the purpose of this test was to demonstrate that combined with realistic input, HIMMELI does output realistic $CH_4$ fluxes, which is not so evident if looking only at the mechanistic sensitivity tests. We admit this is not said very clearly in the manuscript. In addition, we think that comparing the explanatory power of input respiration only with the input + HIMMELI combination is continuation to the sensitivity tests, as it addresses the question of how necessary the transport+oxidation model is.

(12) Suggested changes to the manuscript:

As already mentioned (point 1), we will clarify the role of the Siikaneva test in the paper and, as suggested by another Referee, we will add a comparison of the model with data from another peatland site. This will be a test of how well the current parameterisation fits to other peatland sites.

*(13) Page 16, lines 26-27: Does this illustrate that evaluating sensitivities in the methane only module, especially when production rates are not appropriately driven by changing conditions, is problematic? Temperature is a very important driving factor for CH4 production, but it is not included in the way the module is constructed making it very difficult to interpret the actual sensitivities of the model.*

(13) Response:

We do not think it is problematic. The purpose of the mechanical sensitivity tests was precisely to find out what kind of physical mechanisms govern the behavior of this $CH_4$-only-module, which facilitates its evaluation. For instance, we find it relevant knowledge that the impact of temperature on the processes included in HIMMELI is principally mediated via gas solubilities by affecting the  concentrations of dissolved $O_2$.

Perhaps we misunderstand the Referee's comment but temperature is included in the way that the module is constructed. We agree that simulating anoxic respiration is highly important in $CH_4$ modelling, however, the idea here is that another model (e.g. the soil carbon model of a land surface scheme) has already taken care of it. Most probably the total anoxic respiration rate provided by this other model depends on temperature, but we did not want to set any dependency here since it would have meant, in practice, that the test results are valid only when the dependency is as we described it.

(13) Suggested changes to the manuscript:

We will emphasize on p. 16 that finding different temperature sensitivities when the carbon input is independent of temperature is not a downside but new relevant information for $CH_4$ model development.

*(14) Page 17, line 10: In this model, the peat column and layering is not important, but what about if the gas is being trapped prior to ebullition or even once mobilized from one layer and then trapped in another (e.g. Comas et al., 2015). We know this happens in reality, but it is not included in this model. If it was how would the results of the study change?*

(14) Response:

Like the Reviewer mentions, bubble movement in peat is not included in HIMMELI. However, as mentioned before, we are preparing a manuscript about comparing different ebullition modelling approaches and one of the approaches that are compared in that study included a simple bubble movement scheme. We will shortly describe our findings in that study now here. As expected, if bubble movement (attach, detach) are included, then smaller amount of bubbles reach the surface, i.e. ebullition flux to the atmosphere is smaller. On the other hand, the bubbles that are attached during their ascent release $CH_4$ to the pore water if the pore water $CH_4$ concentration is low enough (Henry's law). Hence, the vertical $CH_4$ concentration gradient is smoother when compared to the case without bubble movement. However, we did not test different layerings with the other ebullition modelling approaches and thus, unfortunately, cannot say whether the model would be more sensitive to the layering if the ebullition model took trapping of gas into account.

(14) Suggested changes to the manuscript:

None, except what was suggested for the comment 11.

REFERENCES

Ito and Inatomi: Use of process-based model for assessing the methane budget of global terrestrial ecosystems and evaluation of uncertainty, Biogeosciences 9, 759-773, doi:10.5194/bg-9-759-2012, 2012.

Kaiser, Göckede, Castro-Morales, Knoblauch, Ekici, Kleinen, Zubrzycki, Sachs, Wille, and Beer: Process-based modelling of the methane balance in periglacial landscapes (JSBACH-methane), Geosci. Model Dev., 10, 333-358, 2017.

Oikawa, Jenerette, Knox, Sturtevant, Verfaillie, Dronova, Poindexter, Eichelmann, and Baldocchi: Evaluation of a hierarchy of models reveals importance of substrate limitation for predicting carbon dioxide and methane exchange in restored wetlands. J. Geophys. Res. Biogeosci., 122, 145-167, doi:10.1002/2016JG003438, 2017.

Ramirez, Baird, Coulthard, and Waddington: Ebullition of methane from peatlands: does peat act as a signal shredder? Geophysical Research Letters, 42, 3371-3379. 2015.

Ringeval, Friedlingstein, Koven, Ciais, de Noblet-Ducoudré, Decharme, and Cadule: Climate-$CH_4$ feedback from wetlands and its interaction with the climate-$CO_2$ feedback, Biogeosciences, 8, 2137-2157, 2011.

Segers and Leffelaar: Modeling methane fluxes in wetlands with gas-transporting plants 1-3, J. Geophys. Res. 106, 2001.

Tang, Zhuang, Shannon, and White: Quantifying wetland methane emissions with process-based models of different complexities, Biogeosciences, 7, 3817-3837, 2010.

van Huissteden, Petrescu, Hendriks, and Rebel: Sensitivity analysis of a wetland methane emission model based on temperate and arctic wetland sites, Biogeosciences, 6, 3035-3051, 2009.

Walter, Heimann, Shannon, and White: A process-based model to derive methane emissions from natural wetlands, Geophys. Res. Lett., 23(25), 3731-3734, 1996.

Wania, Ross, and Prentice.: Implementation and evaluation of a new methane model within a dynamic global vegetation model: LPJ-WHyMe v.1.3.1, Geosci. Model Dev., 3, 565-584, 2010.

Xu, Jaffe, and Mauzerall: A process-based model for methane emission from flooded rice paddy systems. Ecol Model 205, 475-491. 2007.

Xu, Schimel, Thornton, Song, Yan, and Goswami: Substrate and environmental controls on microbial assimilation of soil organic carbon: a framework for Earth system models, Ecol. Lett., 17, 547-555. 2014.

Xu, Yuan, Hanson, Wullschleger, Thornton, Riley, Song, Graham, Song, and Tian: Reviews and syntheses: Four decades of modeling methane cycling in terrestrial ecosystems, Biogeosciences, 13, 3735–3755, 2016.

---

## Author Comment (AC3) · 20 Jun 2017

Dear Referee,

Thank you for taking the time to read our manuscript and for giving good and useful comments. Taking them into account would improve the paper, especially the aim of the model and this work in general should indeed be better described. Below we reply point-by-point to your comments. We hope you find our responses satisfactory and we can prepare and submit a revised version of this manuscript, along the lines we suggest below.

A change that we suggest, not as a direct response to any Referee comment, is related to the model parameters. A set of parameter values that we use in the manuscript was taken from an optimization done using Markov chain Monte Carlo methods with observational $CH_4$ flux data from the Siikaneva peatland site, and we refer to Susiluoto et al. (2017, *in prep.*). The final results and a more exact description of the calibration work are now reported in Susiluoto et al. (2017, GMDD-2017-66). However, there were some major differences between the approaches used here and in the final version of Susiluoto et al., which led to some difference in the values. As the parameter values are not the main point of the present work and as they produce a good fit with observations, we suggest that we keep the current values for the revised manuscript. However, in the revised version we will not anymore refer to Susiluoto et al. but add in the Materials and Methods section a description of the optimization.

Referee comments are typed in italics. They are followed by our responses and suggestions of how we would revise the manuscript, typed as plain text and numbered referring to the comment number.

**Overview:**
*The Global Methane cycle is currently a topic of much interest following the near-zero growth in atmospheric methane concentrations in the early 2000's and its renewed growth since 2007. Various papers have suggested specific sources, including emissions from wetlands, and more recently changes in the atmospheric CH4 sinks, as possible explanations. Wetlands globally are the largest single source of methane, anthropogenic or natural. Boreal wetlands are significant and could become more important still with the faster warming of the Arctic and high northern latitudes. The Wetland Model Intercomparison (cited paper by Melton et al., 2013) highlighted the current performance of wetland models and the large range of wetland areas and methane fluxes simulated by the participating models. The recent paper by Saunois et al. (2016) on the Global Methane Budget removed some of this model uncertainty by prescribing the wetland extent/area.*

*(1) I was expecting the present paper to make this connection to the bigger picture. From the material presented, it is very unclear what the intended application of this new model is (local, regional, global??), how it would be used in practice (standalone or coupled) and indeed what advantage it offers over those mentioned in the paper (i.e., Peatland-VU, CLM) and those already in the literature (see cited paper by Xu et al.).*

(1) Response:
We are sorry that we failed to describe the aim of this model clearly. The motivation for our work was to produce a methane model that can be used in different purposes, ranging from a component of a large-scale biosphere model to a platform for specific studies on methane processes. Therefore it is not clearly stated how the model should be used; the idea is that it can be used within different environments and scales. What is true is that most probably the parameter values we used are not always optimal but the model needs to be re-calibrated, especially if using it in large-scale modelling. This we do not mention in the manuscript, but it should be done.

We acknowledge the fact that HIMMELI does not bring any new processes into $CH_4$ modelling and the process descriptions are based on earlier models. HIMMELI was developed in order to

have a $CH_4$ module that could be plugged into different peatland carbon models and that simulates transport of $CH_4$, $O_2$ and $CO_2$. Rather than taking, modifying and testing directly one of the existing model codes that are developed e.g. with some biosphere model, we decided to systematically start from fundamental elements and combine the process descriptions in a format that can be flexibly applied for different uses as, for instance, the peat column structure is not fixed. We believe this is the advantage: HIMMELI is intended to be a $CH_4$ module that can be used with different input sources. On the other hand, we think that as the model has components similar to other methane models, results of the sensitivity analysis can be generally relevant. In many models, oxygen is simulated but is it known whether the fluxes and effects are realistic.

We also agree that we explained very vaguely how HIMMELI relates to the existing methane models. Xu et al. (2016) listed 40 terrestrial ecosystem models for $CH_4$ cycling. However, when considering only their $CH_4$ emission parts, this number seems to be slightly reduced. For instance, Ringeval et al. (2011) wrote that they included the Walter et al. $CH_4$ model in ORCHIDEE and Spahni et al. (2011) that they applied LPJ-WhyMe in LPI-Bern for biogeochemical modelling of $CH_4$ emissions.

Although HIMMELI does not include all processes that already exist in some models (e.g. alternative $e^-$ acceptors, anaerobic $CH_4$ oxidation), it is among the most complete models considering the transport of compounds. According to Xu et al., there are only 5 models that simulate all vertically resolved biogeochemistry, $O_2$ availability to $CH_4$ oxidation, and three pathways of $CH_4$ transport. Of these, the Xu model (Xu et al. 2007), CLM-Microbe (Xu et al. 2014) and VISIT (Ito & Inatomi, 2012) do not explicitly simulate $O_2$ transport between the atmosphere and peat. On the other hand, LPJ-WhyMe (Wania et al. 2010), a revised multi-substance version of TEM (Tang et al. 2010) and a recent model by Kaiser et al. (2017) - that were not included in the list by Xu et al. -- do simulate all these. HIMMELI also simulates $CO_2$ transport via all three transport pathways. This is not a common feature in $CH_4$ models: to our knowledge, only the multi-substance version of TEM (Tang et al. 2010) and the Segers model (Segers and Leffelaar, 2001) included it.

(1) Suggested changes to the manuscript:
In the Introduction, we will clarify the aim of HIMMELI and describe how this model relates to earlier methane models by adding approximately the above text that refers to the review by Xu et al. (2016).

*(2) The model considers the major CH4 release pathways to the atmosphere (diffusion, plant vascular transport and ebullition) and includes oxidation by O2. O2 is the only electron acceptor considered. What about others?*

(2) Response:

We agree that other electron acceptors are an important issue. We did not include them in the model because we thought their concentrations depend on site characteristics, such as the water source, and it would be difficult to estimate them. Therefore, these estimates would not necessarily improve the accuracy of the model. However, given that our results (and also earlier works) indicate that methane production rate largely drives the simulated emissions and the oxygen inhibition thus plays a significant role, including other e- acceptors could possibly be a way to take into account site differences, for instance, bog vs. fen. This could be done in model version 2.

(2) Suggested changes to the manuscript:

We will add text/discussion about the possible other electron acceptors and distribution of input carbon in the Section 3.1.3 on $CH_4$ production.

*(3) I am little concerned at the realism of completely oxic layers sitting above the watersaturated anoxic layers. In reality, one might expect a continuous transition, as acknowledged by the authors.*

(3) Response:
We agree with the Referee that the choice of using water table depth (WTD) as a strict divider of the peat to oxic and anoxic parts is a simplification and as mentioned in Section 2. 'Key factors for $CH_4$ transport and oxidation', water-filled, anoxic sites can occur above it. In our opinion, however, it is uncertain to what extent the model-based estimate of $CH_4$ emissions of a peatland site or larger area would be improved e.g., by assuming a certain volume of anoxic microsites in the peat above the WTD. Peatlands have microtopography, hollows and hummocks, and even the observation-based site-level WTD is only an approximate value for the peatland, not to speak of a modelled WTD. In addition, simulating partially anoxic peat layers would bring new uncertain parameters in the model. On these grounds, we think this strict division to anoxic and oxic parts is a robust and simple approach.

(3) Suggested changes to the manuscript:
We will add discussion about how realistic is the strict division to oxic/anoxic parts of peat, on page 7, Section 3.1.2.

*(4) The model runs on a daily timestep. This may be appropriate for large-scale decadal or centennial runs but no justification is given. How was this timestep selected and what are the implications for the modelled methane fluxes?*

(4) Response:
The reason for running the model on a daily time step was that the main plan for HIMMELI is to use it with models that provide daily input and so these test results are useful for that purpose. However, we agree that in this work that specifically aims at testing the transport model it would be reasonable to test the effect of time step length on e.g. daily $CH_4$ fluxes. So far we have not done it and thus do not know the effect on output $CH_4$ fluxes, but we can test this.

(4) Suggested changes to the manuscript:
We will test running HIMMELI with realistic input data at frequency shorter than one day, with diurnal variation of soil temperature (as Referee 1 asked about diurnal temperature variation). Results of this model run will be compared with simulation done on daily time step, in which input data are daily averages of the previous test. The outcome will be added to results.

*(5) Many of the model parameters are optimised using the measurements made at a site in southern Finland (see Table 1, p. 30). These results are included in a second paper (Susiluoto et al., 2017), which is in preparation. This makes it hard to assess their significance, especially in the light of the statement The uncertainty of some of these parameters is rather high, and a more complete analysis can be found in Susiluoto et al. (2017, in prep.).*

(5) Response:
This is true. Originally we planned to include a detailed description of the MCMC parameter optimization in this manuscript but since it was already being done for the other paper, Susiluoto et al., we decided to just refer to it. However, we agree this is now left too vague and

as mentioned above, it is necessary to describe the optimisation in this manuscript also because there were some major differences between the approaches used here and in the final version of Susiluoto et al. We can add a new section in Materials and Methods that describes the parameter optimisation.

(5) Suggested changes to the manuscript:

We will add a new Section 3.2 (changing current 3.2 'Model testing' to 3.3) that describes the parameter optimization process done for this manuscript and remove references to Susiluoto et al. 2017 in, e.g. Table 1.

*(6) The cited paper by Rinne et al. (2007) shows an exponential dependence of the measured flux on the peat temperature to day 200 (Figure 6 in paper). The lack of a temperature dependence presumably indicates that the temperature dependence is effectively determined by that of the input 'anaerobic carbon decomposition rate'. The temperature-dependence revealed in Fig. 6 is presumably associated with the modelled transport and loss processes.*

(6) Response:
Yes, this is correct. The temperature dependence in Fig 6 in our manuscript results from the impact of temperature to the processes simulated by HIMMELI, when its input respiration did not depend on temperature. Presumably the exponential dependence of $CH_4$ emissions on temperature would be observed if the anoxic respiration rate depended exponentially on temperature, which often is the case with soil respiration.

(6) Suggested changes to the manuscript:
We will emphasize this when discussing the Fig 6. (current p. 15).

*(7) Many of the key driving variables (soil temperature, leaf area index, water table depth) could be taken either from observations or modelled. It is not clear that this is the case for the anaerobic carbon decomposition rate. If it could be measured, this would improve the utility of HIMMELI.*

(7) Response:
This is true; direct measurement of the anoxic respiration rate is complex or impossible, it can be only estimated/simulated. Apparently the closest possible direct measurement would be on the $CO_2$ flux, which would require that the model includes simulation of photosynthesis, probably driven with solar irradiation. This would of course be possible, as we already now simulated photosynthesis (Appendix B) but our modelling work aimed at creating a module that takes anoxic respiration rate as input and thus is dependent on another model.

(7) Suggested changes:
-

*(8) The model is setup and the modelled CH4 fluxes are compared to eddy covariance flux measurements of CH4 made at Siikaneva, a peatland site in Southern Finland. The intake for the CH4 flux measurements is given as 2.75 m above the peat surface (p. 13). Presumably the surface is fairly homogeneous as no information is given about the footprint nor the prevailing wind direction.*

(8) Response:
Siikaneva is a well-established site following the common standards and requirements for eddy-covariance measurements and its characteristics and representativeness of the data has been analyzed in several papers (Aurela et al. 2007, Rinne et al. 2007). The site is under

ICOS (Integrated Carbon Observation System) labelling process to get accepted as an ICOS Class 2 site.

(8) Suggested changes to the manuscript:
We can add the information given above into the manuscript.

*(9) A good fit of the observed and measured fluxes is seen over several annual cycles. This site is effectively used for both model calibration/optimisation and evaluation. This begs the question of how general the derived parameter values are or whether are they specific to this site. There is an obvious need for comparison against measurements from other sites.*

(9) Response:
The purpose of running the test with data from the Siikaneva site was principally to demonstrate that combined with realistic input, HIMMELI does output realistic $CH_4$ fluxes, which is not so evident if looking at the mechanistic sensitivity tests only. The parameter values are chosen to be physically sound and so they should, in principle, fit also other peatland sites but they are not given as general values for large-scale modelling. They were used here since they were optimized for the Siikaneva site. When moving to other peatlands and especially for large-scale modelling, the model needs to be recalibrated.

We agree that all this was left quite vague in the manuscript and that it would be interesting to see how well the current parameterisation fits to other peatland sites.

(9) Suggested changes to the manuscript:
We will define the scope of this part of the work and the validity of these parameter values better. In addition, we can add a comparison against 5 years of $CH_4$ flux measurements from another peatland site, Lompolojänkkä, a subarctic fen site in Northern Finland (Aurela et al. 2009). This would be a test on how well the current parameterisation fits to another peatland site.

*(10) It would have been interesting to see upscaled fluxes to the regional/boreal scale and hence an estimate of methane emissions from boreal peatlands.*

(10) Response:
This is certainly true, however, this is not within the scope of this paper. This will be done in future works when HIMMELI is combined with a large-scale land surface model.

(10) Suggested changes to the manuscript:
-

**Technical comments:**
*(11) The ellipsis (...) is used throughout the paper for 'to', e.g., page 12, line 14: '10...50 cycles' instead of '10 to 50 cycles'*

(11) Response:
The manuscript preparation guidelines of this journal say: "A range of numbers should be specified as "a to b" or "a…b". We chose to use "a…b" everywhere, however, we can change this.

(11) Suggested changes to the manuscript:
We leave the "a…b" expression only in tables but within the text change it to "a to b".

*(12) Intercomparison is used in several places when 'comparison' is sufficient (a) Page 1, line 30; (b) Page 12, lines 20 and 22; (c) Page 17, line 18.*

(12) Response:
Agreed.

(12) Suggested changes to the manuscript:
We will change 'intercomparison' to 'comparison'.

REFERENCES

Aurela, Riutta, Laurila, Tuovinen, Vesala, Tuittila, Rinne, Haapanala, and Laine: $CO_2$ exchange of a sedge fen in southern Finland – the impact of a drought period, Tellus, 59B, 826-837, 2007.

Ito and Inatomi: Use of process-based model for assessing the methane budget of global terrestrial ecosystems and evaluation of uncertainty, Biogeosciences 9, 759-773, doi:10.5194/bg-9-759-2012, 2012.

Kaiser, Göckede, Castro-Morales, Knoblauch, Ekici, Kleinen, Zubrzycki, Sachs, Wille, and Beer: Process-based modelling of the methane balance in periglacial landscapes (JSBACH-methane), Geosci. Model Dev., 10, 333-358, 2017.

Ringeval, Friedlingstein, Koven, Ciais, de Noblet-Ducoudré, Decharme, and Cadule: Climate-$CH_4$ feedback from wetlands and its interaction with the climate-$CO_2$ feedback, Biogeosciences, 8, 2137-2157, 2011.

Rinne, Riutta, Pihlatie, Aurela, Haapanala, Tuovinen, Tuittila, and Vesala: Annual cycle of methane emission from a boreal fen measured by the eddy covariance technique, Tellus B, 59, 449–457, 2007.

Segers and Leffelaar: Modeling methane fluxes in wetlands with gas-transporting plants 1-3, J. Geophys. Res. 106, 2001.

Spahni, Wania, Neef, van Weele, Pison, Bousquet, Frankenberg, Foster, Joos, Prentice, and van Velthoven: Constraining global methane emissions and uptake by ecosystems. Biogeosciences 8, 1643-1665, doi: 10.5194/bg-81643-2011, 2011.

Tang, Zhuang, Shannon, and White: Quantifying wetland methane emissions with process-based models of different complexities, Biogeosciences, 7, 3817-3837, 2010.

Wania, Ross, and Prentice.: Implementation and evaluation of a new methane model within a dynamic global vegetation model: LPJ-WHyMe v.1.3.1, Geosci. Model Dev., 3, 565-584, 2010.

Xu, Jaffe, and Mauzerall: A process-based model for methane emission from flooded rice paddy systems. Ecol Model 205, 475-491. 2007.

Xu, Schimel, Thornton, Song, Yan, and Goswami: Substrate and environmental controls on microbial assimilation of soil organic carbon: a framework for Earth system models, Ecol. Lett., 17, 547-555. 2014.

Xu, Yuan, Hanson, Wullschleger, Thornton, Riley, Song, Graham, Song, and Tian: Reviews and syntheses: Four decades of modeling methane cycling in terrestrial ecosystems, Biogeosciences, 13, 3735–3755, 2016.

---

## Author Response (AR1)

Dear Editor and Referees,

We have now revised the manuscript. We followed the suggestions and comments of the Referees and tried to clarify the message in places that were unclear. In addition, thanks to the good comments, we noticed that a couple of additional major changes were necessary. We hope that our response is satisfactory and the manuscript is now suitable for publication.

Below please find: A) a list of the relevant changes we have made to the manuscript, B) the original response letters to the Referees, in which we have added in blue text the changes we really made in the manuscript, and C) the marked-up version of the manuscipt. However, the page and line numbers refer to the manuscript version (the separate pdf) in which **no markup** is shown.

A) The relevant changes we made to the manuscript

1. As Referee #1 suggested to add $CO_2$ fluxes in Fig. 10 (now Fig. 13), we also compared the simulated $CO_2$ with respiration rates observed at Siikaneva. We noticed that the $CO_2$ emissions simulated with the previous parameterization were too high, especially due to aerobic respiration rates. Therefore, we decided to change the parameter values. The parameter controlling aerobic respiration is now based on the observed respiration at Siikaneva, and other parameter values are also taken from general literature. The new parameterization is described in the new Section 3.2 and it also affected the description of simulating anoxic respiration in App. B, p. 26 l. 13-20. This change improved the $CO_2$ emission levels. Otherwise the test results are very similar to the earlier version, there were no essential changes. The exact values (in tables and the Results and discussion section) just have changed slightly, and naturally all results in the figures were re-plotted (figures other than Fig. 1). The $CO_2$ fluxes are discussed on p. 23 l. 6-11.

2. The Reviewers asked about the effect of eliminating diurnal temperature variation with daily time step. We tested this and thanks to the test, noticed that although steady-state temperature does not have any significant effect on the output fluxes, *change* in peat temperature did affect, by affecting gas solubilities. Therefore, we added a temperature transition test. The main changes related to the temperature transition are in p. 14 l. 9 & 14-16; p. 19 l. 18-23, and Fig. 8. This also helped us to understand why the model predicted $CH_4$ emission peaks at Siikaneva in 2010, see new text on p. 23, l. 13-24.

3. We realized that it was not reasonable to have 0°C temperature in the steady-state temperature test as this model version does not yet take into account freezing of water. So we removed those results from the analysis (Table 2 and Fig. 7, previously Fig. 6) and mention this limitation now in model description, p. 6 l. 20-21.

4. We added a model vs. observations comparison on another peatland site Lompolojänkkä. The main changes are in Sects. 3.3.4 'Comparison of HIMMELI…', 3.4 'Peatland sites and data' (we re-organized the text by adding a separate section on input data preparation), and 4.4 'Comparison of modelled and measured $CH_4$ fluxes'. Figure 14 b. shows the Lompolojänkkä results. Also a new author, responsible for the new site data, was added.

5. We added text in Introduction p. 3, l. 7-27. It clarifies the aim of HIMMELI and describes how this model relates to earlier methane models. Also the aims of the paper were partly re-formulated, p. 4, l. 5-9.

6. P. 7 l. 11-15: new text about the time step. The new test on shorter time step is in Sects. 3.3.2 and 4.2. and new figures 2. and 12., related to this test, were inserted.

7. P. 7, l. 26-29: we explain why the strict division of air- and water-filled peat is justified.

8. P. 9, l. 13-16: we added text about alternative electron acceptors.

9. P. 11 l. 1-8: we added text about bubble movement.

10. P. 12, Eq. 19: we noticed an error in the equation and removed area *A* from the denominator.

11. We re-formulated Section 3.3 'Model testing' because the new tests were added and because we wanted to clarify the aim of testing the model on the peatland sites.

12. P. 14, l. 20-27: we added clarifying text about why and how the anoxic respiration was independent of WTD and temperature.

**13.** P. 18, first chapter: we added text that still clarifies the role of anoxic respiration in these tests.

**14.** P. 22, l. 16-22: we added comparison of our input anoxic respiration to literature values, to show that the magnitude of the input respiration was realistic.

**15**. The last figure was divided into two, Figs. 14 and 15. Fig. 14 now compares modelled and measured $CH_4$ fluxes at two peatland sites and Fig. 15 shows old and new correlation plots.
* * *
B) Responses to the Referees

Referee comments are typed in *italics*. They are followed by our earlier responses as black plain text and the changes we really made as blue text. The blue page and line numbers refer to the manuscript version in which no markup is shown.

**Referee #1**

*In this paper, a methane submodel is proposed for use in a larger ecosystem C model. While this is a topic of interest to readers of the journal, this submodel has several key weaknesses that affect its acceptability for publication:*
*(1) It is driven by inputs for anaerobic respiration calculated as a first order function of peat C and root exudation derived from assumed vertical distributions of root mass in the anoxic part of the soil profile (Eq. 6). While I appreciate that anaerobic respiration is an input rather than an output of this model, it is nonetheless the key driver of CH4 production, as noted in p. 14 and Fig. 11. Anaerobic respiration therefore needs to be explicitly simulated as part of any CH4 model, rather than optimized for site conditions as done here, as it directly determines modelled CH4 emissions. The determination of Pmax. Rref and dWtol (a poorly constrained term) in eqs. B2 and B3 is necessarily site-specific and detracts from model robustness. This optimization overlooks the possibility that anaerobic respiration can occur in wet soil above the water table. Model testing of anaerobic respiration could have been better constrained by including tests of modelled CO2 fluxes with modelled CH4 fluxes in Fig. 10.*

(1) Response:

We agree that simulation of anaerobic respiration is a significant, perhaps the most significant, component of a complete model of peatland CH4 emissions. However, the target of our work was to produce a module that simulates only the transport and oxidation of CH4. The reason for this is that there are soil carbon models that simulate anoxic respiration and the interface to a CH4 module would be through the respiration. Two examples of this kind of model environments are land surface models JSBACH and JULES. Therefore, we think that a CH4 module that is driven with anoxic respiration is a justifiable modelling unit and in order to ensure its functionality for further use it is reasonable to analyze its sensitivity and performance independently.

The purpose of presenting the (already published) NPP model in the Appendix B is not to claim it is a general photosynthesis model for all the peatlands but, by contrast, to show that we created as realistic NPP as possible for the model testing. The Siikaneva test was done to demonstrate that combined with realistic input, HIMMELI does output realistic $CH_4$ fluxes, which is not so evident if looking only at the mechanistic sensitivity tests. It is true that the parameter values for the NPP model are mainly from a study done at an oligotrophic fen, like Siikaneva. However, we think that in this model-data comparison it is not a downside to use a carbon input that corresponds to reality as closely as possible.

We also agree with the Referee that the choice of using water table depth (WTD) as a strict divider of the peat to oxic and anoxic parts is a simplification and as mentioned in Section 2. 'Key factors for $CH_4$ transport and oxidation', water-filled, anoxic sites can occur above it. In our opinion, however, it is uncertain to what extent the model estimate would be improved e.g., by assuming a certain volume of anoxic microsites in the peat above the WTD, which in practice would mean adding new unknown parameters. In any case, most of the peatlands have microtopography, hollows and hummocks, and even the observation-based site-level WTD is only an approximate value for the peatland, not to speak of a modelled WTD. On these grounds, we think this strict division to anoxic and oxic parts is, although being simple, a robust enough approach to be used in land-surface models.

Including modelled CO2 fluxes into Fig. 10 is indeed a good suggestion.

(1) Suggested changes to the manuscript and the changes we made:

In Introduction, we will clarify the aim of HIMMELI and explain more clearly why simulation of anoxic respiration is not included in the model.

We added text on p. 3 l. 7-10.

We will clarify the role of the Siikaneva test in the paper and, as suggested by another Referee, we will add a comparison of the model with data from another peatland site. This will be a test of how well the current parameterization fits to other peatland sites.

Role of Siikaneva has been clarified especially in Sect. 3.3.4.

Comparison with Lompolojänkkä is reported mainly in Sects. 3.4., 4.4 and Appendix B. Fluxes are plotted in Fig. 14.

We will add discussion about how realistic is the strict division to oxic/anoxic parts of peat, on page 7, Section 3.1.2.

This was added, p. 7, l. 26-29.

We will add CO2 fluxes in Figure 10.

These fluxes were added, now Fig. number 13.

*(2) It is unclear why total anaerobic respiration does not change with WTD on p. 12 l. 5. Simulating such changes is one of the key challenges in CH4 modelling, but is overlooked in this study.*

(2) Response:

In this part of the work, we tested the sensitivity of the simulated CH4 emissions to input. Anoxic respiration is taken by HIMMELI as input, in mol s$^{-1}$ per m$^2$ of ground surface area, i.e., per the simulated peat column. HIMMELI itself cannot change the total anoxic respiration rate as it is the input, but what it does is that it distributes the given input respiration to the inundated layers of the peat column along the root distribution. Number of those layers depends on WTD and so the anoxic respiration per cubic meter changes with WTD.

As we say above, we agree that simulating anoxic respiration is highly important in CH4 modelling, however, the case here is that another model (e.g. a soil carbon model of a land surface scheme) has already taken care of it. Most probably the total anoxic respiration rate provided by this other model depends on WTD, but we did not want to set any dependency here, in the mechanistic sensitivity tests, since it would have meant in practice that the test results are valid only when the dependency is as we described it. In this way we keep the tests as more generic and avoid inherent mixing with a soil carbon model.

The idea in our mechanistic tests was to analyze how much and via what pathways the other driving variables (WTD, temperature, LAI) affect the output CH4 emission rate when the carbon input rate is constant. Given that the anoxic respiration rate largely governs the CH4 emissions, it is important to standardize it and find out what kind of dependencies there are inside the CH4 model alone. As far as we know, this has not been thoroughly analyzed earlier without the mask of changing non-CH4 carbon processes.

(2) Suggested changes to the manuscript:

We will add text that clarifies this issue on p. 12.

This was added, now page 14, l. 21-27.

*(3) The fixed fraction of respiration that generates CH4 (fm in eq. 7) should in theory be fixed at 0.5, rather than be reset to 0.25 for the field study. This fraction directly affects CH4 generation,*
*but completely overlooks acetotrophic vs hydrogenotropic methanogenesis.*

(3) Response:

This is one of the parameters that has high uncertainty and indeed affects CH4 generation directly. It would be great to simulate the different methane production pathways and microbial groups and this way perhaps enable tuning the model

to e.g. different peatland types, but we have not done it so far. CH4 production is only modelled via this one bulk parameter.

Nilsson and Öquist (2009) state in their article that theoretically, the CH4/CO2 quotient from terminal mineralization of soil organic matter in optimal methanogenic conditions ranges from 0 to 0.7, being ~0.5 when carbohydrates are mineralized. Their literature review showed, however, dominance of CO2: the observed CO2/CH4 quotient in anoxic incubations had varied from 0.5 to 36,000 with median value in a filtered data set being 6-7. Also models have used ratios other than 50/50, e.g. the CH4 model by Wania et al. (2010) used CH4/CO2 ratio of 0.1. On this basis, the value 0.25 used in the model calibration is within a realistic range.

We can run the Siikaneva simulations again, with fm of 0.5. However, because the model calibration used 0.25, changing to 0.5 will most probably rise the CH4 emission level higher than the observations, if we keep the other parameter values, in particular the fraction of NPP allocated to root exudates, the same as now. A compromise that we suggest is to present results from both runs in a supplement, which would also illustrate the effect of changing this parameter.

(3) Suggested changes to the manuscript:

We will discuss this parameter in light of the article by Nilsson & Öquist (2009), in the end of Section 3.2.1. We will rerun the Siikaneva simulation done with the logarithmic layer structure using fm of 0.5, and the result (compared with the original run) will be added as a supplement.

What we did, in the end, was to run all the tests using fm of 0.5. We also discuss this parameter on p. 13, l. 9-14.

*(4) There was no clear distinction between gaseous and aqueous diffusive fluxes in eqs. 1 – 3, although they are very different above the water table. I presume these are aqueous fluxes below the water table, but what about gaseous transfer above the water table by which gases are exchanged with the atmosphere? Perhaps this can be easily clarified by the authors.*

(4) Response:
Yes, diffusion happens in water below the water table and in the air above it and the model calculates it accordingly. This, including the description of how the flux is calculated at the water-air interface is explained into more detail in Section 3.1.8. "Diffusion in the peat". We are sorry that this is left unclear in the Section 3.1.1 and around the equations 1-3.

(4) Suggested changes to the manuscript:
We will clarify the text in Section 3.1.1 after the equations 1-3 by explicitly mentioning that the diffusive fluxes in the peat are in water below WTD and in air above it.
This was done, p. 7 l. 6.

*(5) The daily time step of the model eliminates the simulation of diurnal variation in temperature, even though this can be an important driver of that in gas exchange.*

(5) Response:
We agree that in this work, which specifically aims at testing the transport model, it would be reasonable to test how the model works if it is run on a shorter time step. The reason for running it on daily time step was that the main plan for HIMMELI is to use it with models that provide daily input and so the present results were needed for that. We can test running the model on a shorter time step.

(5) Suggested changes to the manuscript:
We will test running HIMMELI with realistic input data at frequency shorter than one day, with diurnal variation of soil temperature. Results of this model run will be compared with simulation done on daily timestep, in which input data are daily averages of the previous test. The outcome will be added to Results.
We did this test. It is described in Sections 3.3.2 and 4.2.

*(6) It is very important to avoid arbitrary parameterizations, such as those associated with the assumed 2 m maximum rooting depth, as these can affect model results in unforeseen ways, and therefore limit the robustness of the model.*

(6) Response:
Maximum rooting depth is, in fact, not fixed to 2 m but it depends on peat depth. If peat depth is less than 2 m, the model uses the peat depth as the maximum rooting depth. In case peat depth is more than 2 m, the model is also prepared to handle the situation, but rooting depth is then set to 2 m according to literature (described in Section 3.1.3). Maximum rooting depth could be changed to a parameter whose value could be determined by the model user but for the current model version, we will not change it.

(6) Suggested changes to the manuscript:
-

*(7) The only air-water interface that appears to be modelled is that at the surface of the water table, yet such interfaces exist throughout the soil above the water table. Gas exchange across these interfaces can cause localized anaerobic zones in which CH4 can be generated.*

(7) Response:
We agree, anaerobic zones can exist above the WTD. However, simulating them would mean increasing the number of uncertain parameters in the model, and simulation of this process would be demanding since, for instance, we do not have corresponding experimental data. As mentioned above, we consider the current approach is robust enough to be used in land surface models.

(7) Suggested changes to the manuscript:
As for the comment (1) above, we will add discussion about how realistic the strict division to oxic/anoxic parts of peat is, on page 7, Section 3.1.2.
This was added, p. 7, l. 26-29.

*(8) Are different root porosities considered in eq. 19? These are important in plant adaptation to wetlands, as well as in root gas transfer.*

(8) Response:
In the current model version they are not considered. A single porosity is assumed for all the gas-transporting vegetation. They could be implemented in further model versions but similarly to the previous comment response (7) adding them would mean increasing the number of uncertain parameters in the model.

(8) Suggested changes to the manuscript:
-

*(9) CH4 emissions appear to have limited sensitivity to temperature (p. 15), even though a T response of anaerobic respiration was considered in the model. However field studies indicate a large sensitivity of CH4 emission to T, as noted later on p. 16, which is likely important in climate warming studies. Has a key process been overlooked here?*

(9) Response:

We regret for having described the test set-up unclearly. In the sensitivity tests described in Sects. 3.2.1 and 4.1, the anoxic respiration was a constant input value that was independent of temperature. This constant respiration was given to HIMMELI that was then driven with different temperatures. Thus, temperature affected the output only by affecting the processes that are included in HIMMELI, that is, inhibition of CH4 production, aerobic respiration, CH4 oxidation, and the transport processes. Therefore, the result of the test tells about the sensitivity of these other processes to temperature.

When running HIMMELI for the Siikaneva site, we used non-constant simulated anoxic respiration, part of which was produced using the temperature dependent Q10 model. In this case both anoxic respiration and output CH4 emission correlated with temperature, however, this was at least principally due to the input correlating with temperature, which is always the case when HIMMELI is used together with an independent carbon cycle model.

According to literature (e.g. Nilsson & Öquist, 2009), CH4 production is more sensitive to temperature than CO2 production, or conversely, the CO2/CH4 quotient decreases when temperature increases. In a way this has now been taken into account to some extent since increasing temperature reduces O2 solubility and thus the dissolved O2 concentration available for inhibiting CH4 production and enhancing CH4 oxidation is reduced.

(9) Suggested changes to the manuscript:
We will clarify the role of the input respiration in different tests, in the section about the tests (3.2) and in the Results and discussion section. We will add a figure that shows the correlation of soil temperature with anoxic respiration used as input in the Siikaneva simulations, as well as correlation between temperature and modelled CH4 emissions.
New text has been added on p. 14, l. 22-27 and p. 18 l. 9-11. Fig. 15 shows the temperature correlations in the Siikaneva test.

*(10) Is it realistic that CH4 emissions should increase with WTD (p. 15), or is root-mediated O2 transport overestimated? Is root growth constrained by O2 below the WT? Or is this model result an artefact of assumptions regarding WTD and anaerobic respiration noted in (2) above?*

(10) Response:
The simple answer to this question is that we do not know how realistic the root oxygen transport is. Here we indeed again face the fact that in our tests, input respiration was not dependent on WTD. In reality, and with a model that simulates anoxic respiration dependent on WTD, probably the anoxic respiration rate per peatland surface area would decrease with decreasing WTD and therefore also the CH4 emissions would decrease, despite of the decreasing root transport of O2 into the inundated soil. But now when WTD had no impact on the anoxic respiration rate, the result was reverse, which is an interesting result as such and not discussed earlier since non-CH4 carbon processes are masking the dynamics of CH4 processes

There are a few previous CH4 models that also simulate the O2 transport to the peat and the consequent O2 concentrations affect different processes in the soil (e.g. Wania et al., 2010, Riley et al., 2011). However, since the observational data on O2 transport is scarce or nonexistent, it is by definition not possible to validate these results. Our analysis shows that it should be done.

Root mass is vertically distributed according to the exponential function (Eq. 4). This means that the root transport capacity is small in the bottom peat layers.

(10) Suggested changes to the manuscript:
We will write a more thorough explanation of how anoxic respiration depended on WTD in the first paragraph of Section 4.1, 'Model sensitivity to input data'.
The new text on p. 14, l. 22-29 explains this, and we also added a short clarifying sentence on p. 18. l. 7-9.

*(11) In the Xu et al. (2016) paper cited in the manuscript, 40 existing CH4 models were reviewed. In many of these models, the issues raised above are explicitly addressed, but some key challenges to further development of these models were raised. The question to be addressed when considering this manuscript for publication is does the model proposed here build upon this earlier work by providing further insight into the key processes by which CH4 emissions are controlled and thereby addressing these challenges? Or is this just another empirical model of CH4 emissions, the parameterization of which is site- and model-specific without reference to earlier modelling work, and therefore of limited interest to the larger modelling community? Unless the authors can provide convincing responses to the points raised above, then I fear the latter.*

(11) Response:

We acknowledge the fact that HIMMELI does not bring any new processes as such into the CH4 model world and the process descriptions largely are from earlier models, which we explicitly mention on p3, lines 11-12. HIMMELI was developed in order to have a CH4 module that could be plugged into different peatland carbon models and that simulates transport of all CH4, O2 and CO2. The parameterization in the current manuscript is based on only one peatland site, however, the aim has been to have physically sound parameter values. When moving to other peatlands and especially if using HIMMELI in large-scale methane modeling, the model needs to be re-calibrated.

We agree that we explained very vaguely how HIMMELI relates to the existing methane models. Xu et al. (2016) listed 40 terrestrial ecosystem models for CH$_4$ cycling. However, when considering only their CH4 emission parts, this number seems to be slightly reduced. For instance, Ringeval et al. (2011) say that they included the Walter et al. CH4 model in ORCHIDEE and Spahni et al. (2011) that they applied LPJ-WhyMe in LPI-Bern for biogeochemical modelling of CH4 emissions.

Although HIMMELI does not include all processes that already exist in some models (e.g. alternative e$^-$ acceptors, anaerobic CH4 oxidation), it is among the most complete models considering the transport of compounds. According to

Xu et al., there are only 5 models that simulate all vertically resolved biogeochemistry, $O_2$ availability to $CH_4$ oxidation, and three pathways of $CH_4$ transport. Of these, the Xu model (Xu et al. 2007), CLM-Microbe (Xu et al. 2014) and VISIT (Ito & Inatomi, 2012) do not explicitly simulate $O_2$ transport between the atmosphere and peat. On the other hand, LPJ-WhyMe (Wania et al. 2010), a revised multi-substance version of TEM (Tang et al. 2010) and a recent model by Kaiser et al. (2017) - that were not included in the list by Xu et al. -- do simulate all these. HIMMELI also simulates $CO_2$ transport via all three transport pathways. This is not a common feature in $CH_4$ models: to our knowledge, only the multi-substance version of TEM (Tang et al. 2010) and the Segers model (Segers & Leffelaar, 2001) included it.

Xu et al. (2016) raised some needs and key challenges to further development of the $CH_4$ models. Some of them are not relevant for HIMMELI as they concern complete peatland ecosystem models -- however, we admit that HIMMELI does not address all of those that were relevant. We can, however, point out two issues. Firstly, Xu et al. emphasized that the models should consider the vertical distribution of processes, which is something that HIMMELI does. Secondly, Xu et al. stated that well-validated $CH_4$ modules should be included in Earth system modeling frameworks. Although not mentioned in the manuscript, the main goal of HIMMELI is to use it as a module in large-scale land surface models (JSBACH, JULES) that are part of ESMs.

When it comes to the modeling community's interest in the manuscript, we think that especially because this model does include components similar to earlier $CH_4$ models, the results of the sensitivity analysis should be interesting. Xu et al. (2016) wrote: "Furthermore, evidence demonstrating that incorporating all of these processes would lead to more accurate prediction is needed". We think our paper is a statement in this type of discussion since it indicates that A) although vertically resolved transport and oxidation processes have significance, the $CH_4$ emissions simulated by this type of models are largely determined by the $CH_4$ production rate and B) adding complexity like e.g. transport of oxygen and effect of $O_2$ concentration on the process rates can have a high impact on the output. However, because of general lack of data, it remains unclear if anyone has validated the realism of the oxygen processes.

(11) Suggested changes to the manuscript:

In the Introduction, we will clarify the aim of HIMMELI and describe how this model relates to earlier methane models by adding the above text that refers to the review by Xu et al. (2016).

We added a shortened version of the text and some clarification in Introduction, p. 3, l. 15-26 and p. 4 l.1-2.

About ebullition: please see our responses to your comments number (11) and (14).

We acknowledge the fact that HIMMELI does not bring any new processes into $CH_4$ modelling and the process descriptions are based on earlier models. This is mentioned on P3, lines 11-12. We wanted to produce a model that simulates the transport of all $CH_4$, $O_2$ and $CO_2$ that is not a common feature among $CH_4$ models. We decided that rather than taking directly one of the existing model codes that are developed with and thus closely connected to some

biosphere model, we would systematically start from fundamental elements and combine the process descriptions in a format that can be flexibly applied for different uses as, for instance, the peat column structure is not fixed.

Although HIMMELI does not include all processes that already exist in some models (e.g. alternative e⁻ acceptors, anaerobic $CH_4$ oxidation), it is among the most complete models considering the transport of compounds. According to Xu et al. (2016), who reviewed 40 existing terrestrial ecosystem models for $CH_4$ cycling, there are only 5 models that simulate all these: vertically resolved biogeochemistry, $O_2$ availability to $CH_4$ oxidation, and three pathways of $CH_4$ transport. Of these 5, the Xu model (Xu et al. 2007), CLM-Microbe (Xu et al. 2014) and VISIT (Ito & Inatomi, 2012) do not explicitly simulate $O_2$ transport between the atmosphere and peat. On the other hand, LPJ-WhyMe (Wania et al. 2010), a revised multi-substance version of TEM (Tang et al. 2010) and a recent model by Kaiser et al. (2017) - that were not included in the list by Xu et al. -- do simulate all these. HIMMELI also simulates $CO_2$ transport via all three transport pathways. To our knowledge, only the multi-substance version of TEM (Tang et al. 2010) and the Segers model (Segers and Leffelaar, 2001) included it.

We think that as the anoxic respiration rate largely governs the $CH_4$ emissions, it is important to standardize it and find out what kind of dependencies there are inside the $CH_4$ model alone, given that it takes a relatively large portion of a complete peatland carbon model. Here the fact that HIMMELI contains similar components as other methane transport models means that the results reveal and clarify inherent assumptions and process dynamics in the other models.

(2) Suggested changes to the manuscript:

We will justify the necessity of the new model by adding the contents of the above text (reference to Xu et al. 2016) in the Introduction.
We added text in Introduction, p. 3 on lines 15-27.

*(3) Page 2, Line 5: Maybe the 2nd largest anthropogenic radiative forcing after CO2? Water vapour causes the greatest radiative forcing in the atmosphere, followed by CO2 and then CH4*

(3) Response:
Agreed.
(3) Suggested changes to the manuscript:
We will correct this sentence: "…inducing the second largest radiative forcing among well-mixed greenhouse gases."
This was done, p. 2 l. 6.

*(4) Page 2, Line 13: Saying no other alternative electron acceptors exist is a bit extreme. Many freshwater wetlands will have cycling of NO3, Fe, SO4, etc., in addition to CH4 production. I suggest rewording this sentence.*

(4) Response:
Yes, we agree.
(4) Suggested changes to the manuscript:
We will remove this sentence.
We removed it (p. 2 l. 14).

*(5) Page 4, Line 28: I guess 45-60% is meant (as opposed to : : :). This happens throughout the manuscript in my version.*
(5) Response:
The manuscript preparation guidelines of this journal say: "A range of numbers should be specified as "a to b" or "a…b". The expression "a-b" is only acceptable in cases when no confusion with "a minus b" is possible." We thought it would be clearest to use the same convention consistently throughout the manuscript and therefore the expression "a…b" everywhere but we can change this.
(5) Suggested changes to the manuscript:
We leave the "a…b" expression only in tables but within the text change it to "a to b".
This was done.

*(6) Page 5, Lines 1-3: And also, peat properties and pore sizes are likely to vary within and between peatlands based on composition of the peat (i.e., sedge vs. wood vs. moss) as well as decomposition status.*

(6) Response:
Yes, true, that is relevant information here. Thank you for pointing out these.
(6) Suggested changes to the manuscript:
We will add this information on Page 5.
We added this on p. 5 l. 21-22.

*(7) Page 5, line 7: the effect of tortuosity on the diffusion coefficient indicates that it is not only the porosity that is important, but the interconnectivity and shape of that porosity and probably the pore size distribution*

(7) Response:
Yes, this is a good point. We are sorry for the inadequate piece of text.

(7) Suggested changes to the manuscript:
We will modify this text so that it also describes the significance of tortuosity.
We changed this, p. 5 l. 25-27.

*(8) Page 6, Line 7: In reality WT is the not the divide between water-filled and partially water-filled pore space. Above the WT there is always some fully saturated layer as the capillary fringe. In practice in the model it doesn't make a difference as the boundary would instead be the capillary fringe, but the way it is written here is technically incorrect.*

(8) Response:
We agree, this is an incorrect statement as it is, this should refer only to the model.

(8) Suggested changes to the manuscript:
We will correct this sentence to: "In the model, WTD is taken as a strict divider of the peat into water-filled and air-filled parts."
This was done, p. 7 l. 21.

*(9) Page 6, Line 11: When WT is above the surface it can become oxygenated by windmixing. Is this considered?*

(9) Response:

Yes, windmixing can affect the $O_2$ concentrations but this is not considered in the model yet. The model naturally is a rough simplification of reality: so far it assumes a pure water layer on top of the peat surface, although there often is vegetation growing in the peat. Vegetation would hinder the windmixing via affecting wind speed and generally modifying the physical conditions affecting thin boundary layers regulating gas transfer across the water-air interface. These processes are not fully understood even for open water surfaces of inland water bodies (we are also working with these issues) and in our opinion the inclusion of partly unknown processes is out of the scope of the present manuscript.

(9) Suggested changes to the manuscript:

We will discuss this point in the Section 3.1.2 in which the possible water layer on top of the peat surface is mentioned.

Discussion was added, p. 8 l. 2-3.

*(10) Equation 7: What about inhibition by other electron acceptors? I know you are not following them in the model, but they could be important in some fen systems. Is CH4 production from the peat matrix accounted for – anaerobic respiration is driven by rooting depth, but CH4 could be produced from other substrates.*

(10) Response:

We think that other electron acceptors are an important issue. We did not include them in the model because we thought their concentrations depend on site characteristics, such as the water source, and it would be difficult to estimate them. Therefore, these estimates would necessarily not improve the accuracy of the model. However, given that our results (and also earlier works) indicate that methane production rate largely drives the simulated emissions and the oxygen

inhibition thus plays a significant role, including other e- acceptors could possibly be a way to take into account site differences, for instance, bog vs. fen. This could be done in future model versions.

In the current model version, anoxic respiration is one bulk input stream and HIMMELI does not take a stand on what organic compounds are decomposed, whether they are root exudates or other substrates. For simplicity, everything is distributed with the root mass except in the case that peat depth exceeds 2 m when some respiration also is allocated in the rootless peat layers. This choice (as opposed to distributing the input e.g. evenly across the peat column) was motivated by the fact that recently fixed carbon seems to be the main source of methane. For instance, according to Oikawa et al. (2017), less than 5% of $CO_2$ and $CH_4$ emissions originate from soils below 50 cm in flooded peatlands. However, in case that HIMMELI is used in a context where it is essential to simulate the different carbon sources, it is not a big task to modify the code so that this becomes possible.

(10) Suggested changes to the manuscript:

We will add text/discussion about the possible other electron acceptors and distribution of input carbon in the Section 3.1.3 on $CH_4$ production.

Discussion was added, p. 9 l. 2-6 and 13-16.
* * *
*(11) Equation 11: Is this really realistic? This would allow a bubble to form, but that doesn't mean that ebullition occurs. Also, once bubbles form, they are often trapped and this affects the concentration gradients and also the ebullition fluxes. A very large bubble release is likely to provide such a high concentration when released that even if the WT is below the surface, not all the CH4 will be oxidized (see page 14, line 15 in the manuscript).*
(11) Response:

We agree with the Referee that after a bubble has been formed there are still several processes that take place before the bubble reaches the surface and contributes to the $CH_4$ flux to the atmosphere. For instance, the bubbles still need to traverse through the peat column up to the atmosphere. Also, like the Referee mentions, during the time that the bubbles travel upwards towards the atmosphere they constantly interact with the surrounding pore water and hence alter e.g. the $CH_4$ concentration gradients.

Such processes are still missing from most of the peatland $CH_4$ models (Xu et al., 2016), including HIMMELI. This is most likely because relatively little is known about bubble movement in peat and how to describe it accurately in models, although there are some attempts to model this process (Ramirez et al., 2015). In general, bubble movement in porous media is a highly complex problem, which depends on the fine-scale structure of the media in which the bubbles are moving in. How to incorporate such complex phenomenon in a simple, yet accurate way in peatland $CH_4$ cycling models is still unsolved. However, as it happens, we are at the moment preparing a manuscript in which we are comparing different ebullition modelling approaches and one of them incorporated a simple scheme to take into account the bubble movement. Nevertheless, as mentioned this is a topic for the other study.

Considering the reviewer's comment about the page 14, line 15: in this manuscript direct ebullition to the surface takes place only when WTD is above the surface. If WTD is below surface, then the $CH_4$ in the bubbles is released to the lowest air layer from where it is transported via diffusion in the air-peat column to the atmosphere. Hence even large bubbles are first released to the bottom air layer below the peat surface, before reaching the atmosphere. We argue that this is how it happens also in reality and hence is the correct way to describe this process in a model.

(11) Suggested changes to the manuscript

We will discuss the points mentioned above (how the bubble movement would happen in reality, compared with the model) in Section 3.1.7. In addition, we will clarify the sentence on page 14. It now is: "…the direct ebullition fluxes to the atmosphere were zero when WTD was below the peat surface" but we will rephrase it: "ebullition to the atmosphere occurred only when WTD was at or above the peat surface". We hope this slightly modified version is clearer.
Discussion was added, p. 11 l. 1-8 and the sentence was clarified.

*(12) Page 12, Lines 20-25: Was the model parameterized with the data from Siikaneva? If so, how appropriate was the test?*

(12) Response:

Yes, the parameter set used in this study was a combination of literature values and values set by calibrating HIMMELI with Siikaneva data. In this sense the test was not appropriate for evaluating the model fit. However, the purpose of this test was to demonstrate that combined with realistic input, HIMMELI does output realistic $CH_4$ fluxes, which is not so evident if looking only at the mechanistic sensitivity tests. We admit this is not said very clearly in the manuscript. In addition, we think that comparing the explanatory power of input respiration only with the input + HIMMELI combination is continuation to the sensitivity tests, as it addresses the question of how necessary the transport+oxidation model is.

(12) Suggested changes to the manuscript:

As already mentioned (point 1), we will clarify the role of the Siikaneva test in the paper and, as suggested by another Referee, we will add a comparison of the model with data from another peatland site. This will be a test of how well the current parameterisation fits to other peatland sites.
We changed the parameterization of the model so that it is not anymore optimized for Siikaneva (Sect. 3.2) and the model is tested also with data from another site Lompolojänkkä (main additions in Sects. 3.4., 4.4 and Appendix B).

*(13) Page 16, lines 26-27: Does this illustrate that evaluating sensitivities in the methane only module, especially when production rates are not appropriately driven by changing conditions, is problematic? Temperature is a very important driving factor for CH4 production, but it is not included in the way the module is constructed making it very difficult to interpret the actual sensitivities of the model.*

(13) Response:

We do not think it is problematic. The purpose of the mechanical sensitivity tests was precisely to find out what kind of physical mechanisms govern the behavior of this $CH_4$-only-module, which facilitates its evaluation. For instance, we find it relevant knowledge that the impact of temperature on the processes included in HIMMELI is principally mediated via gas solubilities by affecting the concentrations of dissolved $O_2$.

Perhaps we misunderstand the Referee's comment but temperature is included in the way that the module is constructed. We agree that simulating anoxic respiration is highly important in $CH_4$ modelling, however, the idea here is that another model (e.g. the soil carbon model of a land surface scheme) has already taken care of it. Most probably the total anoxic respiration rate provided by this other model depends on temperature, but we did not want to set any dependency here since it would have meant, in practice, that the test results are valid only when the dependency is as we described it.

(13) Suggested changes to the manuscript:

We will emphasize on p. 16 that finding different temperature sensitivities when the carbon input is independent of temperature is not a downside but new relevant information for $CH_4$ model development.
A sentence was added, p. 21 l. 12-13.

*(14) Page 17, line 10: In this model, the peat column and layering is not important, but what about if the gas is being trapped prior to ebullition or even once mobilized from one layer and then trapped in another (e.g. Comas et al., 2015). We know this happens in reality, but it is not included in this model. If it was how would the results of the study change?*

(14) Response:
Like the Reviewer mentions, bubble movement in peat is not included in HIMMELI. However, as mentioned before, we are preparing a manuscript about comparing different ebullition modelling approaches and one of the approaches that are compared in that study included a simple bubble movement scheme. We will shortly describe our findings in that study now here. As expected, if bubble movement (attach, detach) are included, then smaller amount of bubbles reach the surface, i.e. ebullition flux to the atmosphere is smaller. On the other hand, the bubbles that are attached during their ascent release $CH_4$ to the pore water if the pore water $CH_4$ concentration is low enough (Henry's law). Hence, the vertical $CH_4$ concentration gradient is smoother when compared to the case without bubble movement. However, we did not test

different layerings with the other ebullition modelling approaches and thus, unfortunately, cannot say whether the model would be more sensitive to the layering if the ebullition model took trapping of gas into account.

(14) Suggested changes to the manuscript:
None, except what was suggested for the comment 11.

We are sorry that we failed to describe the aim of this model clearly. The motivation for our work was to produce a methane model that can be used in different purposes, ranging from a component of a large-scale biosphere model to a platform for specific studies on methane processes. Therefore it is not clearly stated how the model should be used; the idea is that it can be used within different environments and scales. What is true is that most probably the parameter

values we used are not always optimal but the model needs to be re-calibrated, especially if using it in large-scale modelling. This we do not mention in the manuscript, but it should be done.

We acknowledge the fact that HIMMELI does not bring any new processes into $CH_4$ modelling and the process descriptions are based on earlier models. HIMMELI was developed in order to have a $CH_4$ module that could be plugged into different peatland carbon models and that simulates transport of $CH_4$, $O_2$ and $CO_2$. Rather than taking, modifying and testing directly one of the existing model codes that are developed e.g. with some biosphere model, we decided to systematically start from fundamental elements and combine the process descriptions in a format that can be flexibly applied for different uses as, for instance, the peat column structure is not fixed. We believe this is the advantage: HIMMELI is intended to be a $CH_4$ module that can be used with different input sources. On the other hand, we think that as the model has components similar to other methane models, results of the sensitivity analysis can be generally relevant. In many models, oxygen is simulated but is it known whether the fluxes and effects are realistic.

We also agree that we explained very vaguely how HIMMELI relates to the existing methane models. Xu et al. (2016) listed 40 terrestrial ecosystem models for $CH_4$ cycling. However, when considering only their $CH_4$ emission parts, this number seems to be slightly reduced. For instance, Ringeval et al. (2011) wrote that they included the Walter et al. $CH_4$ model in ORCHIDEE and Spahni et al. (2011) that they applied LPJ-WhyMe in LPI-Bern for biogeochemical modelling of $CH_4$ emissions.

Although HIMMELI does not include all processes that already exist in some models (e.g. alternative $e^-$ acceptors, anaerobic $CH_4$ oxidation), it is among the most complete models considering the transport of compounds. According to Xu et al., there are only 5 models that simulate all vertically resolved biogeochemistry, $O_2$ availability to $CH_4$ oxidation, and three pathways of $CH_4$ transport. Of these, the Xu model (Xu et al. 2007), CLM-Microbe (Xu et al. 2014) and VISIT (Ito & Inatomi, 2012) do not explicitly simulate $O_2$ transport between the atmosphere and peat. On the other hand, LPJ-WhyMe (Wania et al. 2010), a revised multi-substance version of TEM (Tang et al. 2010) and a recent model by Kaiser et al. (2017) - that were not included in the list by Xu et al. -- do simulate all these. HIMMELI also simulates $CO_2$ transport via all three transport pathways. This is not a common feature in $CH_4$ models: to our knowledge, only the multi-substance version of TEM (Tang et al. 2010) and the Segers model (Segers and Leffelaar, 2001) included it.

(1) Suggested changes to the manuscript and the changes we made:
In the Introduction, we will clarify the aim of HIMMELI and describe how this model relates to earlier methane models by adding approximately the above text that refers to the review by Xu et al. (2016).
This was added in Introduction, p. 3, l. 7-27.

*(2) The model considers the major CH4 release pathways to the atmosphere (diffusion, plant vascular transport and ebullition) and includes oxidation by O2. O2 is the only electron acceptor considered. What about others?*

(2) Response:

We agree that other electron acceptors are an important issue. We did not include them in the model because we thought their concentrations depend on site characteristics, such as the water source, and it would be difficult to estimate them. Therefore, these estimates would not necessarily improve the accuracy of the model. However, given that our results (and also earlier works) indicate that methane production rate largely drives the simulated emissions and the oxygen inhibition thus plays a significant role, including other e- acceptors could possibly be a way to take into account site differences, for instance, bog vs. fen. This could be done in model version 2.

(2) Suggested changes to the manuscript:

We will add text/discussion about the possible other electron acceptors and distribution of input carbon in the Section 3.1.3 on $CH_4$ production.
Discussion was added, p. 9 l. 13-16 (and 2-6 about carbon distribution).

*(3) I am little concerned at the realism of completely oxic layers sitting above the watersaturated anoxic layers. In reality, one might expect a continuous transition, as acknowledged by the authors.*

(3) Response:
We agree with the Referee that the choice of using water table depth (WTD) as a strict divider of the peat to oxic and anoxic parts is a simplification and as mentioned in Section 2. 'Key factors for $CH_4$ transport and oxidation', water-filled, anoxic sites can occur above it. In our opinion, however, it is uncertain to what extent the model-based estimate of $CH_4$ emissions of a peatland site or larger area would be improved e.g., by assuming a certain volume of anoxic microsites in the peat above the WTD. Peatlands have microtopography, hollows and hummocks, and even the observation-based site-level WTD is only an approximate value for the peatland, not to speak of a modelled WTD. In addition, simulating partially anoxic peat layers would bring new uncertain parameters in the model. On these grounds, we think this strict division to anoxic and oxic parts is a robust and simple approach.

(3) Suggested changes to the manuscript:
We will add discussion about how realistic is the strict division to oxic/anoxic parts of peat, on page 7, Section 3.1.2. This was added, p. 7, l. 26-29.

*(4) The model runs on a daily timestep. This may be appropriate for large-scale decadal or centennial runs but no justification is given. How was this timestep selected and what are the implications for the modelled methane fluxes?*

(4) Response:
The reason for running the model on a daily time step was that the main plan for HIMMELI is to use it with models that provide daily input and so these test results are useful for that purpose. However, we agree that in this work that specifically aims at testing the transport model it would be reasonable to test the effect of time step length on e.g. daily $CH_4$ fluxes. So far we have not done it and thus do not know the effect on output $CH_4$ fluxes, but we can test this.

(4) Suggested changes to the manuscript:
We will test running HIMMELI with realistic input data at frequency shorter than one day, with diurnal variation of soil temperature (as Referee 1 asked about diurnal temperature variation). Results of this model run will be compared with simulation done on daily time step, in which input data are daily averages of the previous test. The outcome will be added to results.
We did this test. It is described in Sections 3.3.2 and 4.2.

*(5) Many of the model parameters are optimised using the measurements made at a site in southern Finland (see Table 1, p. 30). These results are included in a second paper (Susiluoto et al., 2017), which is in preparation. This makes it hard to assess their significance, especially in the light of the statement The uncertainty of some of these parameters is rather high, and a more complete analysis can be found in Susiluoto et al. (2017, in prep.).*

(5) Response:
This is true. Originally we planned to include a detailed description of the MCMC parameter optimization in this manuscript but since it was already being done for the other paper, Susiluoto et al., we decided to just refer to it. However, we agree this is now left too vague and as mentioned above, it is necessary to describe the optimisation in this manuscript also because there were some major differences between the approaches used here and in the final version of Susiluoto et al. We can add a new section in Materials and Methods that describes the parameter optimisation.

(5) Suggested changes to the manuscript:

We will add a new Section 3.2 (changing current 3.2 'Model testing' to 3.3) that describes the parameter optimization process done for this manuscript and remove references to Susiluoto et al. 2017 in, e.g. Table 1.

As mentioned above, we changed the parameterization of the model completely. It is not anymore based on the study by Susiluoto et al. (2017) but on general literature values that are described in Section 3.2. Susiluoto et al. (2017) is a separate and more complete analysis on the significance of the parameters.

*(6) The cited paper by Rinne et al. (2007) shows an exponential dependence of the measured flux on the peat temperature to day 200 (Figure 6 in paper). The lack of a temperature dependence presumably indicates that the temperature dependence is effectively determined by that of the input 'anaerobic carbon decomposition rate'. The temperature-dependence revealed in Fig. 6 is presumably associated with the modelled transport and loss processes.*

(6) Response:
Yes, this is correct. The temperature dependence in Fig 6 in our manuscript results from the impact of temperature to the processes simulated by HIMMELI, when its input respiration did not depend on temperature. Presumably the exponential dependence of $CH_4$ emissions on temperature would be observed if the anoxic respiration rate depended exponentially on temperature, which often is the case with soil respiration.

(6) Suggested changes to the manuscript:
We will emphasize this when discussing the Fig 6. (current p. 15).
We added a sentence emphasizing this issue, p. 19 l. 15-16.

*(7) Many of the key driving variables (soil temperature, leaf area index, water table depth) could be taken either from observations or modelled. It is not clear that this is the case for the anaerobic carbon decomposition rate. If it could be measured, this would improve the utility of HIMMELI.*

(7) Response:
This is true; direct measurement of the anoxic respiration rate is complex or impossible, it can be only estimated/simulated. Apparently the closest possible direct measurement would be on the $CO_2$ flux, which would require that the model includes simulation of photosynthesis, probably driven with solar irradiation. This would of course be possible, as we already now simulated photosynthesis (Appendix B) but our modelling work aimed at creating a module that takes anoxic respiration rate as input and thus is dependent on another model.

(7) Suggested changes:
-

*(8) The model is setup and the modelled CH4 fluxes are compared to eddy covariance flux measurements of CH4 made at Siikaneva, a peatland site in Southern Finland. The intake for the CH4 flux measurements is given as 2.75 m above the peat surface (p. 13). Presumably the surface is fairly homogeneous as no information is given about the footprint nor the prevailing wind direction.*

(8) Response:
Siikaneva is a well-established site following the common standards and requirements for eddy-covariance measurements and its characteristics and representativeness of the data has been analyzed in several papers (Aurela et al. 2007, Rinne et al. 2007). The site is under ICOS (Integrated Carbon Observation System) labelling process to get accepted as an ICOS Class 2 site.

(8) Suggested changes to the manuscript:
We can add the information given above into the manuscript.
We added this, p. 16 l. 2-4.

*(9) A good fit of the observed and measured fluxes is seen over several annual cycles. This site is effectively used for both model calibration/optimisation and evaluation. This begs the question of how general the derived parameter values are or whether are they specific to this site. There is an obvious need for comparison against measurements from other sites.*

(9) Response:
The purpose of running the test with data from the Siikaneva site was principally to demonstrate that combined with realistic input, HIMMELI does output realistic CH4 fluxes, which is not so evident if looking at the mechanistic sensitivity tests only. The parameter values are chosen to be physically sound and so they should, in principle, fit also other peatland sites but they are not given as general values for large-scale modelling. They were used here since they were optimized for the Siikaneva site. When moving to other peatlands and especially for large-scale modelling, the model needs to be recalibrated.

We agree that all this was left quite vague in the manuscript and that it would be interesting to see how well the current parameterisation fits to other peatland sites.

(9) Suggested changes to the manuscript:
We will define the scope of this part of the work and the validity of these parameter values better. In addition, we can add a comparison against 5 years of $CH_4$ flux measurements from another peatland site, Lompolojänkkä, a subarctic fen site in Northern Finland (Aurela et al. 2009). This would be a test on how well the current parameterisation fits to another peatland site.
We changed the parameterisation (Sect. 3.2), re-formulated Section 3.3.4 and added the comparison with fluxes from Lompolojänkkä (main additions are in Sects. 3.4., 4.4 and Appendix B).

*(10) It would have been interesting to see upscaled fluxes to the regional/boreal scale and hence an estimate of methane emissions from boreal peatlands.*
(10) Response:
This is certainly true, however, this is not within the scope of this paper. This will be done in future works when HIMMELI is combined with a large-scale land surface model.
(10) Suggested changes to the manuscript:
-

**Technical comments:**
*(11) The ellipsis (...) is used throughout the paper for 'to', e.g., page 12, line 14: '10...50 cycles' instead of '10 to 50 cycles'*
(11) Response:
The manuscript preparation guidelines of this journal say: "A range of numbers should be specified as "a to b" or "a…b". We chose to use "a…b" everywhere, however, we can change this.
(11) Suggested changes to the manuscript:
We leave the "a…b" expression only in tables but within the text change it to "a to b".
This was done.

*(12) Intercomparison is used in several places when 'comparison' is sufficient (a) Page 1, line 30; (b) Page 12, lines 20 and 22; (c) Page 17, line 18.*
(12) Response:
Agreed.
(12) Suggested changes to the manuscript:
We will change 'intercomparison' to 'comparison'.
Done.

[revised manuscript text omitted]

Parameter values used in the present study are listed in Table 1. Some of them were taken directly from earlier literature (Arah and Stephen, 1998; Vile et al., 2005), but a set of parameter values was obtained by optimizing with Bayesian methods with respect to fluxes measured at the Siikaneva peatland site (see Sect. 3.3) using a least squares objective function with the adaptive Metropolis Markov chain Monte Carlo method (Haario et al., 2001). The values used are the best estimate of these values, i.e., the parameter set of the lowest negative log-likelihood. The uncertainty of some of these parameters is rather high, and a more complete analysis can be found in Susiluoto et al. (2017, *in prep.*).

[revised manuscript text omitted]
 tests we used a value of 0.5 for the parameter $f_m$ (Table 1), the fraction of anaerobic respiration converted to $CH_4$, which was different from the value used for Siikaneva (0.25; see Sect. 3.2.2). However, this parameter practically only controls the output $CH_4$ emission levels, and its effects on the model dynamics are minor.

In these mechanistic sensitivity tests, the anoxic respiration rate (mol m$^{-2}$ s$^{-1}$) was independent of temperature and WTD since the purpose was to analyze the sensitivity of the processes that HIMMELI simulates, and anoxic respiration is only input for HIMMELI. We did not want to set any dependency here since it would have meant, in practice, that the test results are valid only when the dependency is as we described it. In this way we kept the tests more generic. The idea was to analyze how much and via what pathways the other driving variables (WTD, temperature, LAI) affect the output $CH_4$ emission rate when the carbon input rate is constant. The input respiration was always allocated only to the inundated peat layers. Consequently, when the WTD varied, also the number of layers into which the anoxic respiration was allocated varied, although the total respiration rate of the peat column remained constant. Temperature was always constant throughout the soil profile in these experiments, unlike in the simulations of the Siikaneva site.

**3.3.2 Testing a time step of 30 min**

In order to find out whether eliminating the diurnal temperature variation with the daily time step affects the modelled fluxes we compared a model run done on 30 min time step to a run done on the daily time step. We chose an arbitrary summer day, 1 July 2006, and took the soil and air temperature data measured at Siikaneva at 30 min intervals. All other input values were constant over the day in both runs. To avoid possible complications originating from the fact that the first and last temperatures of the chosen day differed by 3 degrees (air) and 0.5 degrees (top soil layer) we modified slightly the temperatures measured in the evening. We interpolated new values between the high afternoon temperatures and the new last temperature that was set

to be close to the first measurement of the day (Fig. 2). We ran HIMMELI over 35000 days using first these data and a 30 min time step, then using the daily average of the temperatures and a 24-h time step. Within this time, the concentrations reached reasonable saturation. WTD was set to -16 cm, the daily average WTD measured at Siikaneva on 1 July 2006, LAI was 1 m$^2$ m$^{-2}$, and anoxic respiration rate was 1 µmol m$^{-2}$ s$^{-1}$.

**3..23 Testing model sensitivity to the description of the peat column**

We ran the model with a seven-year input data series from the Siikaneva fen  and tested how sensitive the results are to peat depth and peat layer thicknesses. We used the same input anoxic respiration, WTD and LAI for all the model runs. The only factor that changed slightly between the different set-ups was the soil temperature since the interpolated temperature profile always followed the layering. In these simulations, anoxic respiration was not constant but simulated (see App. B). The model spin-up was conducted by running the model through the entire seven-year time series of input data until the peat CH$_4$ concentrations stabilized. The spin-up time we used depended on the peat thickness, being up to 600 cycles in the case of 5 m peat.

We tested four peat depths, 1 m, 2 m, 3 m and 5 m using 0.2 m layer thickness in every case. In addition, we tested two evenly spaced layerings, 0.1 and 0.2 m, as well as one logarithmic layer structure, in a 2 m deep peat column. The logarithmic structure was based on the one used in the land surface model JSBACH (Ekici et al., 2014) and the layer thicknesses from top to bottom were 0.06, 0.13, 0.26, 0.52 and 1.03 m.

**3..34  Comparison of HIMMELI and measured CH$_4$ fluxes in the Siikaneva and Lompolojänkkä site**

In order to demonstrate that HIMMELI outputs realistic fluxes when run with realistic input – which is not so evident if looking only at the mechanistic sensitivity tests – we compared the modelled and measured CH$_4$ fluxes  on two sites, Siikaneva and Lompolojänkkä (Sect. 3.4) using anoxic respiration estimated for the sites as input. The purpose of this comparison also was a general evaluation of  what is the -significance of using HIMMELI compared to using (simulated) anoxic respiration rate directly as the basis of CH$_4$ emission estimations.

**3.3 Peatland sites and data**

**3.4.1 Siikaneva site description**

The eddy covariance flux measurement site is located in Siikaneva in Ruovesi, Southern Finland (61°49´ N, 24°11´ E, 162 m a.s.l.) (Rinne et al., 2007). The site is a boreal oligotrophic fen where the vegetation is dominated by sedges (*C. rostrata*, *C.*

*limosa*, *E. vaginatum*), Rannoch-rush (*Scheuchzeria palustris*) and peat mosses (*Sphagnum balticum*, *S. majus*, *S. papillosum*). Peat depth at the measurement footprint is 2 to 4 m. Annual mean temperature in 1971 to 2000 at a nearby weather station was 3.3° C and precipitation 713 mm (Drebs et al., 2002). Siikaneva is a well-established site following the common standards and requirements for eddy-covariance measurements and its characteristics and representativeness of the data has been

5   analyzed in several papers (Aurela et al., 2007; Rinne et al., 2007).

We drove the model with daily averages of WTD, peat temperature profile, LAI and anoxic respiration rate, and compared the results with daily medians of CH₄ flux data from years 2005…2011. Simulation of LAI and anoxic respiration are described in Appendix B. Peat temperature has been monitored in Siikaneva at five depths: -5 cm, -10 cm, -20 cm, -35 cm and -50 cm.

10  We created the temperature profile by interpolating linearly between the measurements. To obtain temperatures below the -50 cm depth 
[revised manuscript text omitted]
. Simulated NPP (that formed the main part of the anoxic respiration) or the LAI curve may have been biased or the way how the anoxic respiration depended on NPP may have been too straightforward. It now depended directly on the daily NPP and produced $CH_4$ and $CO_2$ immediately, there were no pools of potential $CH_4$ substrates. In reality, as well as in soil carbon models with which HIMMELI could be combined, there is some lag in the process of carbon fixation turning into root exudates and further to $CH_4$.

Most probably both the magnitude and the annual pattern of the emissions can be improved by more realistic simulation of anoxic respiration. However, the model explained the variation in emissions relatively well: the $R^2$ between model and measurement was 0.63 at Siikaneva and 0.70 at Lompolojänkkä.

The simulated $CO_2$ emissions were also at realistic levels both at Siikaneva and Lompolojänkkä. According to Aurela et al. (2007), the mean respiration in Siikaneva in July 2005 was 1.1 to 2.3 µmol m$^{-2}$ s$^{-1}$ and in our simulation, the mean $CO_2$ emission in July 2005 was 2.4 to 2.8 µmol m$^{-2}$ s$^{-1}$ (Fig. 15). At Lompolojänkkä, monthly respiration of July 2006 to 2008 was around 2.5 µmol m$^{-2}$ s$^{-1}$ (Aurela et al., 2009) while the model simulated a $CO_2$ flux of 3.5 µmol m$^{-2}$ s$^{-1}$ (data not shown). The model overestimated slightly the emissions, especially given that it does not include $CO_2$ from autotrophic respiration unlike the observed fluxes, but the result is still reasonable.

Summer 2010 at Siikaneva was interesting since both model and measurements show the highest emission peaks then. The maximum emissions do not coincide exactly on the same days, but they are temporally close. In HIMMELI, the main reason was an exceptionally abrupt temperature rise in the peat water, followed by decreasing gas solubilities and increased ebullition

– as was observed in the temperature transition tests. Summer 2010 was unusually hot in Finland and so the heat can very well be the cause of the observed high emissions also in nature. We do not know whether the effect really can be transmitted via gas solubilities instead of, for instance, increased respiration. Grant and Roulet (2002) compared simulated and measured $CH_4$ emissions at a beaver pond. Their model captured some bubbling events, driven by warming soil that affected both fermentation

5 and methanogenesis rates and gas solubilities. In our case, the simulated input anoxic respiration did not increase noticeably during this high-emission period, but our simulation may underestimate the effect of temperature. Moreover, although the soil temperature profile used to run the model was derived from measurements, it was an approximation as it was created by linear interpolation between measurement points. The temperature change of the lower peat layers may be exaggerated compared with reality. However, the modelled $CH_4$ emission peaks nicely matched with observations.

10 ~~measurements reasonably but it underestimated the observed $CH_4$ emissions on average by approximately 20%. The main reason for this discrepancy was that the anoxic respiration rate calculation (App. B) deviated from that of Susiluoto et al. (2017) and we did not optimize the parameters of HIMMELI specifically for the set-up of this study but used their parameter values as such. Simulation-based input data may also have caused the divergence between the model and measurements in the autumns when generally the modelled $CH_4$ emissions seemed to decline too early. Simulated NPP (that formed the main part~~

15 ~~of the anoxic respiration) or the LAI curve may have been biased or the way how the anoxic respiration depended on NPP may have been too straightforward. It now depended directly on the daily NPP and produced $CH_4$ and $CO_2$ immediately, there were no pools of potential $CH_4$ substrates. 
[revised manuscript text omitted]
. 2004) was estimated following closely the parallel study Susiluoto et al. (2017) in which the HIMMELI parameters were optimized for Siikaneva. We simply simulated the NPP time series for the sites, allocated the NPP vertically along the root distribution (Eq. 4), and removed the fraction that was in aerobic conditions, i.e., above the WTD (based on the measured WTD time series). Susiluoto et al. (2017) simulated production of CH$_4$ substrates as temperature dependent anoxic peat decomposition $V_{pR}$ (mol m$^{-2}$ s$^{-1}$) and decomposition of root exudates in the inundated peat layers. This was computed with a simple respiration model combined with HIMMELI. Since in the present study the focus was on testing the HIMMELI model as such, we did not involve the complete respiration model here but produced input for HIMMELI using the same components as Susiluoto et al. (2017) in a simplified form. The input respiration in our study was simply the sum of anoxic peat decomposition rate and estimated root exudate production $V_{exu}$ (mol m$^{-2}$ s$^{-1}$) in the inundated peat layers, we did not simulate the actual decomposition of the root exudate pool. The soil profile for which the respirationis was computed was 2 m of peat with 0.1 m layers. This NPP was scaled so that the output visually fitted the measured CH$_4$ fluxes at Siikaneva using a scaling factor $f_s$ of 0.4.

$V_{exu}$ was estimated by taking a fixed fraction $f_{exu}$ of the net primary productivity (NPP) of vascular vegetation, i.e., simulated net photosynthesis rate $P_n$ (mol m$^{-2}$ s$^{-1}$) of vascular plants. $f_{exu}$ (Table B1) was obtained from Susiluoto et al. (2017) and tThe 
[revised manuscript text omitted]

[Figure]

[Figure]

**Figure 15. ~~Comparison of the measured and modelled CH₄ fluxes of Siikaneva. The modelled flux is from the simulation with logarithmic layer structure and 2 m of peat, driven with measured temperature profile and WTD and simulated LAI and anoxic respiration. (a) The time series of measurements versus model, (b) correlation between model and measurement, (c) correlation between the anoxic respiration rate given as input to the model and the measured CH₄ fluxes.~~rrelations between (a) modelled and measured CH₄ flux, (b) input anoxic respiration and measured CH₄ flux, (c) observed air temperature and input anoxic respiration, and (d) observed air temperature and modelled CH₄ flux. The data are from the Siikaneva test (Fig. 14 a).**